# CONTEXTUAL BANDITS WITH ONLINE NEURAL REGRESSION

**Rohan Deb, Yikun Ban, Shiliang Zhou, Jingrui He, & Arindam Banerjee**
University of Illinois, Urbana-Champaign
{rd22,yikunb2,szuo3,jingrui,arindamb}@cs.illinois.edu

## ABSTRACT

Recent works have shown a reduction from contextual bandits to online regression under a realizability assumption (Foster and Rakhlin, 2020; Foster and Krishnamurthy, 2021). In this work, we investigate the use of neural networks for such online regression and associated Neural Contextual Bandits (NeuCBs). Using existing results for wide networks, one can readily show a $\mathcal{O}(\sqrt{T})$ regret for online regression with square loss, which via the reduction implies a $\mathcal{O}(\sqrt{K}T^{3/4})$ regret for NeuCBs. Departing from this standard approach, we first show a $\mathcal{O}(\log T)$ regret for online regression with almost convex losses that satisfy QG (Quadratic Growth) condition, a generalization of the PL (Polyak-Łojasiewicz) condition, and that have a unique minima. Although not directly applicable to wide networks since they do not have unique minima, we show that adding a suitable small random perturbation to the network predictions surprisingly makes the loss satisfy QG with unique minima. Based on such a perturbed prediction, we show a $\mathcal{O}(\log T)$ regret for online regression with both squared loss and KL loss, and subsequently convert these respectively to $\tilde{\mathcal{O}}(\sqrt{KT})$ and $\tilde{\mathcal{O}}(\sqrt{KL^*} + K)$ regret for NeuCB, where $L^*$ is the loss of the best policy. Separately, we also show that existing regret bounds for NeuCBs are $\Omega(T)$ or assume i.i.d. contexts, unlike this work. Finally, our experimental results on various datasets demonstrate that our algorithms, especially the one based on KL loss, persistently outperform existing algorithms.

## 1 INTRODUCTION

Contextual Bandits (CBs) provide a powerful framework for sequential decision making problems, where a learner takes decisions over $T$ rounds based on partial feedback from the environment. At each round, the learner is presented with $K$ context vectors to choose from, and a scalar output is generated based on the chosen context. The objective is to minimize[1] the accumulated output in $T$ rounds. Many existing works assume that the expected output at each round depends linearly on the chosen context. This assumption has enabled tractable solutions, such as UCB-based approaches (Chu et al., 2011; Abbasi-Yadkori et al., 2011) and Thompson Sampling (Agrawal and Goyal, 2013). However, in many real-world applications, the output function may not be linear, rendering these methods inadequate. Recent years have seen progress in the use of neural networks for contextual bandit problems (Zhou et al., 2020; Zhang et al., 2021) by leveraging the representation power of overparameterized models, especially wide networks (Allen-Zhu et al., 2019b; Cao and Gu, 2019b; Du et al., 2019; Arora et al., 2019b). These advances focus on learning the output function in the Neural Tangent Kernel (NTK) regime and drawing on results from the kernel bandit literature (Valko et al., 2013).

Separately, (Foster and Rakhlin, 2020) adapted the inverse gap weighting idea from Abe and Long (1999); Abe et al. (2003) and gave an algorithm (SquareCB) that relates the regret of CBs, $\mathrm{Reg}_{\mathrm{CB}}(T)$ to the regret of online regression with square loss $\mathrm{R}_{\mathrm{Sq}}(T)$. The work uses a realizability assumption: the true function generating the outputs belongs to some function class $\mathcal{F}$. In this approach, the learner learns a score for every arm (using online regression) and computes the probability of choosing an arm based on the inverse of the gap in scores leading to a regret bound $\mathrm{Reg}_{\mathrm{CB}}(T) = \mathcal{O}\left(\sqrt{KT\mathrm{R}_{\mathrm{Sq}}(T)}\right)$. Subsequently, Foster and Krishnamurthy (2021) revisited SquareCB and provided a modified algorithm (FastCB), with binary Kullback–Leibler (KL) loss and a re-weighted inverse gap weighting scheme that attains a *first-order* regret bound. A *first-order* regret bound is data-dependent in the

---

[1]We use the loss formulation instead of rewards.

sense that it scales sub-linearly with the loss of the best policy $L^*$ instead of $T$. They showed that if regret of online regression with KL loss is $R_{KL}(T)$ then the regret for the bandit problem can be bounded by $\text{Reg}_{CB}(T) = \mathcal{O}(\sqrt{KL^* R_{KL}(T)} + K R_{KL}(T))$. Further, under the realizability assumption, Simchi-Levi and Xu (2020) showed that an offline regression oracle with $\mathcal{O}(\log T)$ calls can also achieve an optimal regret for CBs. This improves upon the $\mathcal{O}(T)$ calls to an online regression oracle made by SquareCB and FastCB but works only for the stochastic setting, i.e., when the contexts are drawn i.i.d. from a fixed distribution.

In this work we develop novel regret bounds for online regression with neural networks and subsequently use the reduction in Foster and Rakhlin (2020); Foster and Krishnamurthy (2021) to give regret guarantees for NeuCBs. Before we unpack the details, we discuss the research gaps in existing algorithms for NeuCBs to better motivate our contributions in the context of available literature.

## 1.1 RESEARCH GAPS IN NEURAL CONTEXTUAL BANDITS

We discuss some problems and restrictive assumptions with existing NeuCB algorithms. Table 1 summarizes these comparisons. We also discuss why naively extending existing results for wide networks with the CBs to online regression reduction lead to sub-optimal regret bounds. In Section F we further outline some related works in contextual bandits and overparameterized neural models.

**1. Neural UCB ((Zhou et al., 2020)) and Neural TS ((Zhang et al., 2021)):** Both these works focus on learning the loss function in the Neural Tangent Kernel (NTK) regime and drawing on results from the kernel bandit literature (Valko et al., 2013). The regret bound is shown to be $\tilde{\mathcal{O}}(\tilde{d}\sqrt{T})$, where $\tilde{d}$ is the effective dimension of the NTK matrix. When the eigen-values of the kernel decreases polynomially, one can show that $\tilde{d}$ depends logarithmically in $T$ (see Remark 4.4 in (Zhou et al., 2020) and Remark 1 in (Valko et al., 2013)) and therefore the final regret is still $\tilde{\mathcal{O}}(\sqrt{T})$. However in Appendix A we show that under the assumptions in the papers, the regret bounds for both NeuralUCB and NeuralTS is $\Omega(T)$ in worst case.

**2. EE-Net (Ban et al., 2022b):** This work uses an exploitation network for learning the output function and an exploration network to learn the potential gain of exploring at current step. Although EE-Net avoids picking up a $\tilde{d}$ dependence in its regret bound, it has two drawbacks. 1) It assumes that the contexts are chosen i.i.d. from a given distribution, an assumption that generally does not hold for real world CB problems. 2) It needs to store all the previous networks until the current time step and then makes a prediction by randomly picking a network from all the past networks (see lines 32-33 in Algorithm 1 of Ban et al. (2022b)), a strategy that does not scale to real world deployment.

**3. SquareCB (Foster and Rakhlin, 2020) and FastCB (Foster and Krishnamurthy, 2021):** Both these works provide regret bounds for CBs in terms of regret for online regression. Using online gradient descent for online regression (as in this paper) with regret $\mathcal{O}(\sqrt{T})$, SquareCB and FastCB provide $\mathcal{O}(\sqrt{K}T^{3/4})$ and $\mathcal{O}(\sqrt{KL^*}T^{1/4} + K\sqrt{T})$ regret for CBs (see Example 2 in Section 2.3 of Foster and Rakhlin (2020) and Example 5 in Section 4 of Foster and Krishnamurthy (2021) respectively). Existing analysis with neural models that use almost convexity of the loss (see Chen et al. (2021)) show $\mathcal{O}(\sqrt{T})$ regret for online regression, and naively combining it with the SquareCB and FastCB lead to the same sub-optimal regret bounds for NeuCBs.

## 1.2 OUR CONTRIBUTIONS

1. **Lower Bounds:** As outlined in Section 1.1, we formally show that an (oblivious) adversary can always choose a set of context vectors and a reward function at the beginning, such that the regret bounds for NeuralUCB ((Zhou et al., 2020)) and NeuralTS ((Zhang et al., 2021)) becomes $\Omega(T)$. See Appendix A.1 and A.2 for the corresponding theorems and their proofs.

2. **QG Regret:** We provide $\mathcal{O}(\log T) + \epsilon T$ regret for online regression when the loss function satisfies (i) $\epsilon$- almost convexity, (ii) QG condition, and (iii) has unique minima (cf. Assumption 2) as long as the minimum cumulative loss in hindsight (interpolation loss) is $O(\log T)$. This improves over the $\mathcal{O}(\sqrt{T}) + \epsilon T$ bound in Chen et al. (2021) that only exploits $\epsilon$- almost convexity.

3. **Regret for wide networks:** While the QG result is not directly applicable for neural models, since they do not have unique minima, we show adding a suitably small random perturbation to the prediction (10), makes the losses satisfy QG with unique minima. Using such a perturbed neural prediction, we provide regret bounds with the following loss functions:

| Algorithm | Regret | Remarks |
|---|---|---|
| NeuralUCB (Zhou et al., 2020) | $\tilde{\mathcal{O}}(\tilde{d}\sqrt{T})$ | Bound is $\Omega(T)$ in worst case. |
| NeuralTS Zhang et al. (2021) | $\tilde{\mathcal{O}}(\tilde{d}\sqrt{T})$ | Bound is $\Omega(T)$ in worst case. |
| EE-Net (Ban et al., 2022b) | $\tilde{\mathcal{O}}(\sqrt{T})$ | Assumes that the contexts are drawn i.i.d and needs to store all previous networks. |
| NeuSquareCB (This work) | $\tilde{\mathcal{O}}(\sqrt{KT})$ | No dependence on $\tilde{d}$ and holds even when the contexts are chosen adversarially. |
| NeuFastCB (This work) | $\tilde{\mathcal{O}}(\sqrt{L^*K} + K)$ | No dependence on $\tilde{d}$ and holds even when the contexts are chosen adversarially. |

Table 1: Comparison with prior works. $T$ is the horizon length, $L^*$ is the cumulative loss of the best policy, $\tilde{d}$ is the effective dimension of the NTK matrix and $K$ is the number of arms.

    (a) **Squared loss:** We provide $\mathcal{O}(\log T)$ regret for online regression with the perturbed network (Theorem 3.2) and thereafter using the online regression to CB reduction obtain $\tilde{\mathcal{O}}(\sqrt{KT})$ regret for NeuCBs with our algorithm NeuSquareCB (Algorithm 1).

    (b) **Kullback–Leibler (KL) loss:** We further provide an $\mathcal{O}(\log T)$ regret for online regression with KL loss using the perturbed network (Theorem 3.3). To the best of our knowledge, this is the first result that shows PL/QG condition holds for KL loss, and would be of independent interest. Further, using the reduction, we obtain the first data dependent regret bound of $\tilde{\mathcal{O}}(\sqrt{L^*K} + K)$ for NeuCBs with our algorithm NeuFastCB (Algorithm 2).

4. **Empirical Performance:** Finally, in Section 5 we compare our algorithms against baseline algorithms for NeuCBs. Unlike previous works, we feed in contexts to the algorithms in an adversarial manner (see Section 5 for details). Our experiments on various datasets demonstrate that the proposed algorithms (especially NeuFastCB) consistently outperform existing NeuCB algorithms, which themselves have been shown to outperform their linear counterparts such as LinUCB and LinearTS (Zhou et al., 2020; Zhang et al., 2021; Ban et al., 2022b).

We also emphasize that our regret bounds are independent of the effective dimension that appear in kernel bandits (Valko et al., 2013) and some recent neural bandit algorithms (Zhou et al., 2020; Zhang et al., 2021). Further our algorithms are efficient to implement, do not require matrix inversions unlike NeuralUCB (Zhou et al., 2020) and NeuralTS (Zhang et al., 2021), and work even when the contexts are chosen adversarially unlike approaches specific to the i.i.d. setting (Ban et al., 2022b).

## 2   Neural Online Regression: Setting and Formulation

**Problem Formulation:** At round $t \in [T]$, the learner is presented with an input $\mathbf{x}_t \in \mathcal{X} \subset \mathbb{R}^d$ and is required to make real-valued predictions $\hat{y}_t$. Then, the true outcome $y_t \in \mathcal{Y} = [0,1]$ is revealed.

**Assumption 1 (Realizability).** *The conditional expectation of $y_t$ given $\mathbf{x}_t$ is given by some unknown function $h \colon \mathcal{X} \mapsto \mathcal{Y}$, i.e., $\mathbb{E}[y_t|\mathbf{x}_t] = h(\mathbf{x}_t)$. Further, the context vectors satisfy $\|\mathbf{x}_t\| \leq 1 \, \forall t \in [T]$.*

**Neural Architecture:** We consider a feedforward neural network with smooth activations as in Du et al. (2019); Banerjee et al. (2023) and the network output is given by

$$f(\theta_t; \mathbf{x}) := m^{-1/2}\mathbf{v}_t^\top \phi(m^{-1/2}W_t^{(L)}\phi(\cdots \phi(m^{-1/2}W_t^{(1)}\mathbf{x}) \cdots)) , \qquad (1)$$

where $L$ is the number of hidden layers and $m$ is the width of the network. Further, $W_t^{(1)} \in \mathbb{R}^{m \times d}, W_t^{(l)} \in \mathbb{R}^{m \times m}, \forall l \in \{2, \ldots, L\}$ are layer-wise weight matrices with $W_t^{(l)} = [w_{t,i,j}^{(l)}]$, $\mathbf{v}_t \in \mathbb{R}^m$ is the last layer vector and $\phi(\cdot)$ is a lipschitz and smooth (pointwise) activation function.

We define

$$\theta_t := (\text{vec}(W_t^{(1)})^\top, \ldots, \text{vec}(W_t^{(L)})^\top, \mathbf{v}^\top)^\top \qquad (2)$$

as the vector of all parameters in the network. Note that $\theta_t \in \mathbb{R}^p$, where $p = md + (L-1)m^2 + m$ is total number of parameters.

**Regret:** At time $t$ an algorithm picks a $\theta_t \in B$, where $B \subset \mathbb{R}^p$ is some comparator class, and outputs the prediction $f(\theta_t; \mathbf{x}_t)$. Consider a loss function $\ell : \mathcal{Y} \times \mathcal{Y} \mapsto \mathbb{R}$ that measures the error between $f(\theta_t; \mathbf{x}_t)$ and the true output $y_t$. Then the regret of neural online regression with loss $\ell$ is defined as

$$\mathrm{R}(T) := \sum_{t=1}^{T} \ell(y_t, f(\theta_t; \mathbf{x}_t)) - \inf_{\theta \in B} \sum_{t=1}^{T} \ell(y_t, f(\theta; \mathbf{x}_t)) , \tag{3}$$

**Remark 2.1.** Given the definition of regret in (3), one might wonder if we are assuming that the function $h$ in Assumption 1 is somehow $f(\tilde{\theta}; \cdot)$ for some $\tilde{\theta} \in B$. Wide networks in fact can realize any function $h$ on a finite set of $T$ points (see Theorem E.1 in Appendix E).

Using almost convexity of the loss function for wide networks, Chen et al. (2021) show $\mathrm{R}(T) = \mathcal{O}(\sqrt{T})$ regret. Instead, we work with a small random perturbation to the neural model prediction denoted by $\tilde{f}$ (see Section 3) with $\mathbb{E}[\tilde{f}] = f$ and consider the following regret:

$$\tilde{\mathrm{R}}(T) := \sum_{t=1}^{T} \ell(y_t, \tilde{f}(\theta_t; \mathbf{x}_t)) - \inf_{\theta \in B} \sum_{t=1}^{T} \ell(y_t, \tilde{f}(\theta; \mathbf{x}_t)) . \tag{4}$$

As we shortly show in Section 3, the surprising aspect of working with such mildly perturbed $\tilde{f}$ is that we will get $\tilde{R}(T) = O(\log T)$. Further, the cumulative loss with such $\tilde{f}$ will also be competitive against the best non-perturbed $f$ with $\theta \in B$ in hindsight (Remark 3.4).

**Notation:** $n = \mathcal{O}(t)$ (and $\Omega(t)$ respectively) means there exists constant $c > 0$ such that $n \le ct$ (and $n \ge ct$ respectively). Further the notation $n = \Theta(t)$ means there exist constants $c_1, c_2 > 0$ such that $c_1 t \le n \le c_2 t$. The notations $\tilde{\mathcal{O}}(t), \tilde{\Omega}(t), \tilde{\Theta}(t)$ further hide the dependence on logarithmic terms.

## 3   NEURAL ONLINE REGRESSION: REGRET BOUNDS

Our objective in this section is to provide regret bounds for projected Online Gradient Descent (OGD) with the projection operator $\prod_B(\theta) = \mathrm{arginf}_{\theta' \in B} \|\theta' - \theta\|_2$. The iterates are given by

$$\theta_{t+1} = \prod_B \left( \theta_t - \eta_t \nabla \ell(y_t, f(\theta_t; \mathbf{x}_t)) \right). \tag{5}$$

**Definition 3.1 (Quadratic Growth (QG) condition).** *Consider a function $J : \mathbb{R}^p \to \mathbb{R}$ and let the solution set be $\mathcal{J}^* = \{\theta' : \theta' \in \arg\min_\theta J(\theta)\}$. Then $J$ is said to satisfy the QG condition with QG constant $\mu$, if $J(\theta) - J(\theta^*) \ge \frac{\mu}{2} \|\theta - \theta^*\|^2 , \forall \theta \in \mathbb{R}^p \setminus \mathcal{J}^*$, where $\theta^*$ is the projection of $\theta$ onto $\mathcal{J}^*$.*

**Remark 3.1 (PL $\Rightarrow$ QG).** The recent literature has extensively studied the PL condition and how neural losses satisfy the PL condition (Karimi et al., 2016; Liu et al., 2020; 2022). While the Quadratic Growth (QG) condition has not been as widely studied, one can show that the PL condition implies the QG condition (Karimi et al., 2016, Appendix A), i.e., for $\mu > 0$

$$J(\theta) - J(\theta^*) \le \frac{1}{2\mu} \|\nabla J(\theta)\|_2^2 \quad \text{(PL)} \quad \Rightarrow \quad J(\theta) - J(\theta^*) \ge \frac{\mu}{2} \|\theta - \theta^*\|_2^2 \quad \text{(QG)}$$

**Assumption 2.** *Consider a predictor $g(\theta; \mathbf{x})$ and suppose the loss $\ell(y_t, g(\theta; \mathbf{x}_t)), \forall t \in [T]$, satisfies*

*(a) Almost convexity, i.e., there exists $\epsilon > 0$, such that $\forall \theta, \theta' \in B$,*

$$\ell(y_t, g(\theta; \mathbf{x}_t)) \ge \ell(y_t, g(\theta'; \mathbf{x}_t)) + \langle \theta - \theta', \nabla_\theta \ell(y_t, g(\theta'; \mathbf{x}_t)) \rangle - \epsilon. \tag{6}$$

*(b) QG condition i.e., $\exists \mu > 0$, such that $\forall \theta \in B \setminus \Theta_t^*$, where $\Theta_t^* = \{\theta_t | \theta_t \in \mathrm{arginf}_\theta \ell(y_t, g(\theta; \mathbf{x}_t))\}$ and $\theta_t^*$ is the projection of $\theta$ onto $\Theta_t^*$ we have*

$$\ell(y_t, g(\theta; \mathbf{x}_t)) - \ell(y_t, g(\theta_t^*; \mathbf{x}_t)) \ge \frac{\mu}{2} \|\theta - \theta_t^*\|_2^2. \tag{7}$$

*(c) has a unique minima.*

Following (Liu et al., 2020; 2022), we also assume the following for the activation and loss function.

**Assumption 3.** *With $\ell_t := \ell(y_t, \hat{y}_t)$, $\ell_t' := \frac{d\ell_t}{d\hat{y}_t}$, and $\ell_t'' := \frac{d^2\ell_t}{d\hat{y}_t^2}$, we assume that the loss $\ell(y_t, \hat{y}_t)$ is Lipschitz, i.e., $|\ell_t'| \le \lambda$, strongly convex, i.e., $\ell_t'' \ge a$ and smooth, i.e., $\ell_t'' \le b$, for some $\lambda, a, b > 0$.*

**Theorem 3.1** (**Regret Bound under QG condition**). *Under Assumptions 2 and 3 the regret of projected OGD with step size $\eta_t = \frac{4}{\mu t}$, where $\mu$ is the QG constant from Assumption 2(b), satisfies*

$$\tilde{R}(T) \leq \mathcal{O}\left(\frac{\lambda^2}{\mu}\log T\right) + \epsilon T + 2\inf_{\theta \in B}\sum_{t=1}^{T}\ell(y_t, g(\theta; \mathbf{x}_t)). \tag{8}$$

The proof of Theorem 3.1 can be found in Appendix C.1.

**Remark 3.2.** Under Assumption 2(a), Chen et al. (2021) show $R(T) = \mathcal{O}(\sqrt{T}) + \epsilon T$. In contrast our bound improves the first term to $\mathcal{O}(\log T)$ but has an extra term: cumulative loss of the best $\theta$, and uses two additional assumptions (2(b) and 2(c)). Note that the third term vanishes if $g$ interpolates and the interpolation loss is zero (holds for e.g., with over-parameterized networks and square loss). Further for over-parameterized networks Chen et al. (2021) show that $\epsilon = 1/\mathrm{poly}(m)$, and therefore the $\epsilon T$ term is $\mathcal{O}(1)$ for large enough $m$. A similar argument also holds for our model.

Next we discuss if Theorem 3.1 can indeed be used for neural loss functions. For concreteness consider the square loss $\ell_{\mathrm{Sq}}(y_t, f(\theta_t; \mathbf{x}_t)) := \frac{1}{2}(y_t - f(\theta_t; \mathbf{x}_t))^2$. Following Liu et al. (2020); Banerjee et al. (2023), we make the following assumption on the initial parameters of the network.

**Assumption 4.** *We initialize $\theta_0$ with $w_{0,ij}^{(l)} \sim \mathcal{N}(0, \sigma_0^2)$ for $l \in [L]$ where $\sigma_0 = \frac{\sigma_1}{2\left(1+\frac{\sqrt{\log m}}{\sqrt{2m}}\right)}, \sigma_1 > 0$, and $\mathbf{v}_0$ is a random unit vector with $\|\mathbf{v}_0\|_2 = 1$.*

Next we define $K_{\mathrm{NTK}}(\theta) := [\langle \nabla f(\theta; \mathbf{x}_t), \nabla f(\theta; \mathbf{x}_{t'}) \rangle] \in \mathbb{R}^{T \times T}$ to be the so-called Neural Tangent Kernel (NTK) matrix at $\theta$ (Jacot et al., 2018) and make the following assumption at initialization.

**Assumption 5.** $K_{\mathrm{NTK}}(\theta_0)$ *is positive definite, i.e.,* $K_{\mathrm{NTK}}(\theta_0) \succeq \lambda_0 \mathbb{I}$ *for some* $\lambda_0 > 0$.

**Remark 3.3.** The above assumption on the NTK is common in the deep learning literature (Du et al., 2019; Arora et al., 2019b; Cao and Gu, 2019a) and is ensured as long any two context vectors $\mathbf{x}_t$ do not overlap. All existing regret bounds for NeuCBs make this assumption (see Assumption 4.2 in Zhou et al. (2020), Assumption 3.4 in Zhang et al. (2021) and Assumption 5.1 in Ban et al. (2022b)).

Finally we choose the comparator class $B = B_{\rho,\rho_1}^{\mathrm{Frob}}(\theta_0)$, the layer-wise Frobenius ball around the initialization $\theta_0$ with radii $\rho, \rho_1$ (to be chosen as part of analysis) which is defined as

$$B_{\rho,\rho_1}^{\mathrm{Frob}}(\theta_0) := \{\theta \in \mathbb{R}^p \text{ as in (2)} : \|\mathrm{vec}(W^{(l)}) - \mathrm{vec}(W_0^{(l)})\|_2 \leq \rho, l \in [L], \|\mathbf{v} - \mathbf{v}_0\|_2 \leq \rho_1\}. \tag{9}$$

Consider a network $f(\theta; \mathbf{x})$ that satisfies Assumption 5 with $\theta_0$ initialized as in Assumption 4. Then for $B = B_{\rho,\rho_1}^{\mathrm{Frob}}(\theta_0)$ we can show the following: (i) $\ell_{\mathrm{Sq}}$ satisfies Assumption 2(a) with $\epsilon = \mathcal{O}(\mathrm{poly}(L, \rho, \rho_1))/\sqrt{m}$ (Lemma 13) and therefore with $m = \Omega(\mathrm{poly}(T, L, \rho, \rho_1))$, the second term in (8) is $\mathcal{O}(1)$. (ii) $\ell_{\mathrm{Sq}}$ for a wide networks satisfies PL condition (Liu et al., 2022) which implies QG, and therefore Assumption 2(b) is satisfied. (iii) Further the network interpolates (Theorem E.1) which ensures that the third term in (8) is 0, which makes the rhs $\mathcal{O}(\log T)$. However neural models do not have a unique minima and as such we cannot take advantage of Theorem 3.1 as Assumption 2(c) is violated. To mitigate this, in the next subsection we construct a randomized predictor $\tilde{f}$ with $\mathbb{E}[\tilde{f}] = f$ and show that Assumption 2(a), 2(b) and 2(c) hold in high probability.

### 3.1 REGRET BOUNDS FOR SQUARED LOSS

To ensure that the loss has a unique minimizer at every time step, we consider a random network with a small perturbation to the output. In detail, given the input $\mathbf{x}_t$, we define a perturbed network as

$$\tilde{f}(\theta_t, \mathbf{x}_t, \boldsymbol{\varepsilon}) = f(\theta_t; \mathbf{x}_t) + c_p \sum_{j=1}^{p} \frac{(\theta_t - \theta_0)^T e_j \varepsilon_j}{m^{1/4}}, \tag{10}$$

where $f(\theta_t; \mathbf{x}_t)$ is the output of the network as defined in (1), $c_p$ is the perturbation constant to be chosen later, $\{e_j\}_{j=1}^{p}$ are the standard basis vectors and $\boldsymbol{\varepsilon} = (\varepsilon_1, \ldots, \varepsilon_p)^T$ is a random vector where $\varepsilon_j$ is drawn i.i.d from a Rademacher distribution, i.e., $P(\varepsilon_j = +1) = P(\varepsilon_j = -1) = 1/2$.

Note that $\mathbb{E}[\tilde{f}] = f$. Further $\tilde{f}$ ensures that the expected loss $\mathbb{E}_{\boldsymbol{\varepsilon}}\ell_{\mathrm{Sq}}(y_t, \tilde{f}(\theta, \mathbf{x}_t, \boldsymbol{\varepsilon}))$ has a unique minimizer (see Lemma 14). However, since running projected OGD on $\mathbb{E}_{\boldsymbol{\varepsilon}}\ell_{\mathrm{Sq}}(y_t, \tilde{f}(\theta, \mathbf{x}_t, \boldsymbol{\varepsilon}))$ is not

feasible, we next consider an average version of it. Let $\boldsymbol{\varepsilon}_s = (\varepsilon_{s,1}, \varepsilon_{s,2}, \ldots, \varepsilon_{s,p})^T$ where each $\varepsilon_{s,i}$ is i.i.d. Rademacher. Consider $S$ such i.i.d. draws $\{\boldsymbol{\varepsilon}_s\}_{s=1}^S$, and define the predictor

$$\tilde{f}^{(S)}\left(\theta; \mathbf{x}_t, \boldsymbol{\varepsilon}^{(1:S)}\right) = \frac{1}{S}\sum_{s=1}^S \tilde{f}(\theta; \mathbf{x}_t, \boldsymbol{\varepsilon}_s), \tag{11}$$

where $\tilde{f}(\theta; \mathbf{x}_t, \boldsymbol{\varepsilon}_s)$ is as defined in (10). We define the corresponding regret with square loss as

$$\tilde{\mathsf{R}}_{\mathsf{Sq}}(T) = \sum_{t=1}^T \ell_{\mathsf{Sq}}\left(y_t, \tilde{f}^{(S)}\left(\theta_t; \mathbf{x}_t, \boldsymbol{\varepsilon}^{(1:S)}\right)\right) - \min_{\theta \in B_{\rho,\rho_1}^{\mathrm{Frob}}(\theta_0)} \sum_{t=1}^T \ell_{\mathsf{Sq}}\left(y_t, \tilde{f}^{(S)}\left(\theta; \mathbf{x}_t, \boldsymbol{\varepsilon}^{(1:S)}\right)\right). \tag{12}$$

However, instead of running projected OGD with the loss $\ell_{\mathsf{Sq}}\left(y_t, \tilde{f}^{(S)}\left(\theta; \mathbf{x}_t, \boldsymbol{\varepsilon}^{(1:S)}\right)\right)$, we use

$$\mathcal{L}_{\mathsf{Sq}}^{(S)}\left(y_t, \left\{\tilde{f}(\theta; \mathbf{x}_t, \boldsymbol{\varepsilon}_s)\right\}_{s=1}^S\right) := \frac{1}{S}\sum_{s=1}^S \ell_{\mathsf{Sq}}\left(y_t, \tilde{f}(\theta; \mathbf{x}_t, \boldsymbol{\varepsilon}_s)\right) \tag{13}$$

Note that since $\ell_{\mathsf{Sq}}$ is convex in the second argument, using Jensen we have

$$\ell_{\mathsf{Sq}}\left(y_t, \tilde{f}^{(S)}\left(\theta; \mathbf{x}_t, \boldsymbol{\varepsilon}^{(1:S)}\right)\right) = \ell_{\mathsf{Sq}}\left(y_t, \frac{1}{S}\sum_{s=1}^S \tilde{f}(\theta; \mathbf{x}_t, \boldsymbol{\varepsilon}_s)\right) \le \frac{1}{S}\sum_{s=1}^S \ell_{\mathsf{Sq}}\left(y_t, \tilde{f}(\theta; \mathbf{x}_t, \boldsymbol{\varepsilon}_s)\right). \tag{14}$$

Subsequently we will show via (14) that bounding the regret with (13) implies a bound on (12).

**Theorem 3.2** (**Regret Bound for square loss**). *Under Assumption 1, 4 and 5 with appropriate choice of step-size sequence $\{\eta_t\}$, width $m$, and perturbation constant $c_p$ in (10), with probability at least $\left(1 - \frac{C}{T^4}\right)$ for some constant $C > 0$, over the randomness of initialization and $\{\boldsymbol{\varepsilon}\}_{s=1}^S$, the regret in (12) of projected OGD with loss $\mathcal{L}_{\mathsf{Sq}}^{(S)}\left(y_t, \left\{\tilde{f}(\theta; \mathbf{x}_t, \boldsymbol{\varepsilon}_s)\right\}_{s=1}^S\right)$, $S = \Theta(\log m)$ and projection ball $B_{\rho,\rho_1}^{\mathrm{Frob}}(\theta_0)$ with $\rho = \Theta(\sqrt{T}/\lambda_0)$ and $\rho_1 = \Theta(1)$ is given by $\tilde{\mathsf{R}}_{\mathsf{Sq}}(T) = \mathcal{O}(\log T)$.*

*Proof sketch.* The proof of the theorem follows along four key steps as described below. All of these hold with high probability over the randomness of initialization and $\{\boldsymbol{\varepsilon}_s\}_{s=1}^S$. A detailed version of the proof along with all intermediate lemmas and their proofs are in Appendix C.2. Note that we do not use Assumptions 2 and 3 and, but rather explicitly prove that they hold.

1. **Square loss is Lipschitz, strongly convex, and smooth w.r.t. the output:** This step ensures that Assumption 3 is satisfied. Strong convexity and smoothness follow trivially from the definition of the $\ell_{\mathsf{Sq}}$. To show that $\ell_{\mathsf{Sq}}$ is Lipschitz we show that the output $\tilde{f}(\theta; \mathbf{x}, \boldsymbol{\varepsilon})$ is bounded for any $\theta \in B_{\rho,\rho_1}^{\mathrm{Frob}}(\theta_0)$. Also note from Theorem 3.1 that the lipschitz parameter of the loss, $\lambda$ appears in the $\log T$ term and therefore to obtain a $\mathcal{O}(\log T)$ regret we also ensure that $\lambda = \mathcal{O}(1)$.

2. **The average loss in (13) is almost convex and has a unique minimizer:** We show that with $S = \Theta(\log m)$, the average loss in (13) is $\nu$ - Strongly Convex (SC) with $\nu = \mathcal{O}\left(\frac{1}{\sqrt{m}}\right)$ w.r.t. $\theta \in B_{\rho,\rho_1}^{\mathrm{Frob}}(\theta_0), \forall t \in [T]$ which immediately implies Assumption 2(a) and 2(c).

3. **The average loss in (13) satisfies the QG condition:** It is known that square loss with wide networks under Assumption 5 satisfies the PL condition (eg. Liu et al. (2022)) with $\mu = \mathcal{O}(1)$. We show that the average loss in (13) with $S = \Theta(\log m)$, also satisfies the PL condition with $\mu = \mathcal{O}(1)$, which implies that it satisfies the QG condition with same $\mu$.

4. **Bounding the final regret:** Steps 1 and 2 above surprisingly show that with a small output perturbation, square loss satisfies (a) almost convexity, (b) QG, and (c) unique minima as in Assumption 2. Combining with step 3, all the assumptions of Theorem 3.1 are satisfied by $\mathcal{L}_{\mathsf{Sq}}^{(S)}$. Using union bound over the three steps, invoking Theorem 3.1 we get with high probability

$$\sum_{t=1}^T \mathcal{L}_{\mathsf{Sq}}^{(S)}\left(y_t, \left\{\tilde{f}(\theta_t; \mathbf{x}_t, \boldsymbol{\varepsilon}_s)\right\}_{s=1}^S\right) - \inf_{\theta \in B_{\rho,\rho_1}^{\mathrm{Frob}}(\theta_0)} \sum_{t=1}^T \mathcal{L}_{\mathsf{Sq}}^{(S)}\left(y_t, \left\{\tilde{f}(\theta; \mathbf{x}_t, \boldsymbol{\varepsilon}_s)\right\}_{s=1}^S\right)$$

$$\le 2 \inf_{\theta \in B_{\rho,\rho_1}^{\mathrm{Frob}}(\theta_0)} \sum_{t=1}^T \mathcal{L}_{\mathsf{Sq}}^{(S)}\left(y_t, \left\{\tilde{f}(\theta; \mathbf{x}_t, \boldsymbol{\varepsilon}_s)\right\}_{s=1}^S\right) + \mathcal{O}(\log T). \tag{15}$$

---

**Algorithm 1** Neural SquareCB (`NeuSquareCB`); Uses Square loss

---

1: Initialize $\boldsymbol{\theta}_0, \gamma, \{\eta_t\}$
2: **for** $t = 1, 2, ..., T$ **do**
3:     Receive contexts $\mathbf{x}_{t,1}, ..., \mathbf{x}_{t,K}$, and compute $\hat{y}_{t,a} = \tilde{f}^{(S)}\big(\theta; \mathbf{x}_{t,a}, \boldsymbol{\varepsilon}^{(1:S)}\big), \forall a \in [K]$ using *equation* 11
4:     Let $b = \arg\min_a \hat{y}_{t,a}, \ p_{t,a} = \frac{1}{K + \gamma(\hat{y}_{t,b} - \hat{y}_{t,a})}$, and $p_{t,b} = 1 - \sum_{a \neq b} p_{t,a}$
5:     Sample arm $a_t \sim p_t$ and observe output $y_{t,a_t}$
6:     Update $\theta_{t+1} = \prod_{B_{\rho,\rho_1}^{\mathrm{Frob}}(\theta_0)} \Big(\theta_t - \eta_t \nabla \mathcal{L}_{\mathrm{Sq}}^{(S)}\big(y_{t,a_t}, \{\tilde{f}(\theta; \mathbf{x}_{t,a_t}, \boldsymbol{\varepsilon}_s)\}_{s=1}^{S}\big)\Big).$
7: **end for**

---

Finally we show $\inf_{\theta \in B_{\rho,\rho_1}^{\mathrm{Frob}}(\theta_0)} \sum_{t=1}^{T} \mathcal{L}_{\mathrm{Sq}}^{(S)}\big(y_t, \{\tilde{f}(\tilde{\theta}^*; \mathbf{x}_t, \boldsymbol{\varepsilon}_s)\}_{s=1}^{S}\big) = \mathcal{O}(1)$ using the fact that wide networks interpolate (Theorem E.1) which implies $\sum_{t=1}^{T} \mathcal{L}_{\mathrm{Sq}}^{(S)}\big(y_t, \{\tilde{f}(\theta_t; \mathbf{x}_t, \boldsymbol{\varepsilon}_s)\}_{s=1}^{S}\big) = \mathcal{O}(\log T)$ which using (14) and recalling the definition in (13) implies $\tilde{\mathrm{R}}_{\mathrm{Sq}}(T) = \mathcal{O}(\log T)$    □

**Remark 3.4.** Note that $\sum_{t=1}^{T} \mathcal{L}_{\mathrm{Sq}}^{(S)}\big(y_t, \{\tilde{f}(\theta_t; \mathbf{x}_t, \boldsymbol{\varepsilon}_s)\}_{s=1}^{S}\big) = \mathcal{O}(\log T)$ from step-4 above also implies that $\sum_{t=1}^{T} \mathcal{L}_{\mathrm{Sq}}^{(S)}\big(y_t, \{\tilde{f}(\theta_t; \mathbf{x}_t, \boldsymbol{\varepsilon}_s)\}_{s=1}^{S}\big) - \min_{\theta \in B_{\rho,\rho_1}^{\mathrm{Frob}}(\theta_0)} \sum_{t=1}^{T} \ell_{\mathrm{Sq}}\big(y_t, f(\theta, \mathbf{x}_t)\big) = \mathcal{O}(\log T)$ and therefore our predictions are competitive against $f$ as defined in (1) as well.

**Remark 3.5.** Although the average loss in (13) is SC (Lemma 6), we do not use standard results from Shalev-Shwartz (2012); Hazan (2021) to obtain $\mathcal{O}(\log T)$ regret. This is because, the strong convexity constant $\nu = \mathcal{O}(1/\sqrt{m})$, and although OGD ensures $\mathcal{O}(\log T)$ regret for SC functions, the constant hidden by $\mathcal{O}$ scales as $\frac{1}{\nu} = \sqrt{m}$. For large width models, $m >> T$, and therefore this approach does not yield a $\mathcal{O}(\log T)$ bound. The key idea is to introduce bare minimum strong convexity using (10), to ensure unique minima, without letting go of the QG condition with $\mu = \mathcal{O}(1)$.    □

## 3.2 REGRET BOUNDS FOR KL LOSS

Next we consider the binary KL loss, defined as $\ell_{\mathrm{KL}}(y_t, \hat{y}_t) = y_t \ln\left(\frac{y_t}{\sigma(\hat{y}_t)}\right) + (1 - y_t) \ln\left(\frac{1 - y_t}{1 - \sigma(\hat{y}_t)}\right)$, where $\sigma(y) = \frac{1}{1+e^{-y}}$ is the sigmoid function. Following the approach outlined in Section 3.1, we consider a perturbed network as defined in (10). Note that here the output of the neural network is finally passed through a sigmoid. As in (11), we will consider a combined predictor. With slight abuse of notation, we define the prediction and the corresponding regret with $\ell_{\mathrm{KL}}$ respectively as

$$\sigma\big(\tilde{f}^{(S)}\big(\theta; \mathbf{x}_t, \boldsymbol{\varepsilon}^{(1:S)}\big)\big) = \frac{1}{S} \sum_{s=1}^{S} \sigma(\tilde{f}(\theta; \mathbf{x}_t, \boldsymbol{\varepsilon}_s)) \tag{16}$$

$$\tilde{\mathrm{R}}_{\mathrm{KL}}(T) = \sum_{t=1}^{T} \ell_{\mathrm{KL}}\big(y_t, \sigma\big(\tilde{f}^{(S)}\big(\theta_t; \mathbf{x}_t, \boldsymbol{\varepsilon}^{(1:S)}\big)\big)\big) - \min_{\theta \in B_{\rho,\rho_1}^{\mathrm{Frob}}(\theta_0)} \sum_{t=1}^{T} \ell_{\mathrm{KL}}\big(y_t, \sigma\big(\tilde{f}^{(S)}\big(\theta; \mathbf{x}_t, \boldsymbol{\varepsilon}^{(1:S)}\big)\big)\big).$$

**Theorem 3.3** (**Regret Bound for KL Loss**). *Under Assumption 1, 4 and 5 for $y_t \in [z, 1 - z]$, $0 < z < 1$, with appropriate choice of step-size sequence $\{\eta_t\}$, width $m$, and perturbation constant $c_p$, with high probability over the randomness of initialization and $\{\boldsymbol{\varepsilon}\}_{s=1}^{S}$, the regret of projected OGD with loss $\mathcal{L}_{KL}^{(S)}\big(y_t, \{\sigma(\tilde{f}(\theta; \mathbf{x}_t, \boldsymbol{\varepsilon}_s))\}_{s=1}^{S}\big) = \frac{1}{S} \sum_{s=1}^{S} \ell_{KL}\left(y_t, \sigma(\tilde{f}(\theta; \mathbf{x}_t, \boldsymbol{\varepsilon}_s))\right), S = \Theta(\log m)$ and projection ball $B_{\rho,\rho_1}^{\mathrm{Frob}}(\theta_0)$ with $\rho = \Theta(\sqrt{T}/\lambda_0)$ and $\rho_1 = \Theta(1)$ is given by $\mathrm{R}_{KL}(T) = \mathcal{O}(\log T)$.*

The proof of the theorem follows a similar approach as in proof of Theorem 3.2 (See Appendix C.3).

## 4 NEURAL CONTEXTUAL BANDITS: FORMULATION AND REGRET BOUNDS

We consider a contextual bandit problem where a learner needs to make sequential decisions over $T$ time steps. At any round $t \in [T]$, the learner observes the context for $K$ arms $\mathbf{x}_{t,1}, ,..., \mathbf{x}_{t,K} \in \mathbb{R}^d$, where the contexts can be chosen adversarially. The learner chooses an arm $a_t \in [K]$ and then the associated output $y_{t,a_t} \in [0, 1]$ is observed. We make the following assumption on the output.

**Assumption 6.** *The conditional expectation of $y_{t,a}$ given $\mathbf{x}_{t,a}$ is given by $h$: $\mathbb{R}^d \mapsto [0, 1]$, i.e., $\mathbb{E}[y_{t,a} | \mathbf{x}_{t,a}] = h(\mathbf{x}_{t,a})$. Further, the context vectors satisfy $\|\mathbf{x}_{t,a}\| \leq 1, t \in [T], a \in [K]$.*

---

**Algorithm 2** Neural FastCB (NeuFastCB); Uses KL loss

1: Initialize $\boldsymbol{\theta}_0, \gamma, \{\eta_t\}$
2: **for** $t = 1, 2, ..., T$ **do**
3:     Receive contexts $\mathbf{x}_{t,1}, ..., \mathbf{x}_{t,K}$ and compute $\hat{y}_{t,k}, \ \forall k \in [K]$ using (16)
4:     Let $b_t = \underset{k \in [K]}{\arg\min}\, \hat{y}_{t,k}, \ \ p_{t,k} = \frac{\hat{y}_{t,b_t}}{K\hat{y}_{t,b_t} + \gamma(\hat{y}_{t,k} - \hat{y}_{t,b_t})}, k \in [K]$, and $p_{t,b_t} = 1 - \sum_{k \neq b_t} p_{t,k}$.
5:     Sample arm $a_t \sim p_t$ and observe output $y_{t,a_t}$
6:     Update $\theta_{t+1} = \prod_{B_{\rho,\rho_1}^{\mathrm{Frob}}(\theta_0)} \left( \theta_t - \eta_t \nabla \mathcal{L}_{\mathrm{KL}}^{(S)} \left( y_{t,a_t}, \left\{ \tilde{f}(\theta; \mathbf{x}_{t,a_t}, \boldsymbol{\varepsilon}_s) \right\}_{s=1}^S \right) \right)$.
7: **end for**

---

The learner's goal is to minimize the regret of the contextual bandit problem and is defined as the expected difference between the cumulative output of the algorithm and that of the optimal policy:

$$\mathrm{Reg}_{\mathrm{CB}}(T) = \mathbb{E}\Big[ \sum_{t=1}^T \left( y_{t,a_t} - y_{t,a_t^*} \right) \Big], \tag{17}$$

where $a_t^* = \arg\min_{k \in [K]} h(\mathbf{x}_{t,a})$ is the best action minimizing the expected output in round $t$.

NeuSquareCB and NeuFastCB are summarized in **Algorithm** 1 and **Algorithm** 2 respectively. At time $t$, the algorithm computes $\tilde{f}^{(S)}\left( \theta; \mathbf{x}_{t,a}, \boldsymbol{\varepsilon}^{(1:S)} \right), \forall a \in [K]$ using (11) (see line 4). It then computes the probability of selecting an arm using the gap between learned outputs following inverse gap weighting scheme from Abe and Long (1999) (see line 6) and samples an action $a_t$ from this distribution (see line 7). It then receives the true output for the selected arm $y_{t,a_t}$, and updates the parameters of the network using projected online gradient descent. NeuFastCB employs a similar approach, except that it uses KL loss to update the parameters of the network and uses a slightly different weighting scheme to compute the action distribution (Foster and Krishnamurthy, 2021).

**Theorem 4.1** (**Regret bound for** NeuSquareCB). *Under Assumption 6 and 5 with appropriate choice of the parameter $\gamma$, step-size sequence $\{\eta_t\}$ width $m$, and regularization parameter $c_p$, with high probability over the randomness in the initialization and $\{\varepsilon\}_{s=1}^S$ the regret for* NeuSquareCB *with $\rho = \Theta(\sqrt{T}/\lambda_0), \rho_1 = \Theta(1)$ is given by $\mathrm{Reg}_{\mathrm{CB}}(T) \leq \tilde{\mathcal{O}}(\sqrt{KT})$.*

**Theorem 4.2** (**Regret bound for** NeuFastCB). *Under Assumption 6 and 5 with appropriate choice of the parameter $\gamma$, step-size sequence $\{\eta_t\}$ width $m$, and regularization parameter $c_p$, with high probability over the randomness in the initialization and $\{\varepsilon\}_{s=1}^S$, the regret for* NeuFastCB *with $\rho = \Theta(\sqrt{T}/\lambda_0), \rho_1 = \Theta(1)$ is given by $\mathrm{Reg}_{\mathrm{CB}}(T) \leq \tilde{\mathcal{O}}(\sqrt{L^* K} + K)$, where $L^* = \sum_{t=1}^T y_{t,a_t^*}$.*

The proof of the Theorem 4.1 and 4.2 follow using the reduction from Foster and Rakhlin (2020) and (Foster and Krishnamurthy, 2021) respectively, and crucially using our sharp regret bounds for online regression in Section 3. We provide both proofs in Appendix D for completeness.

**Remark 4.1.** Note that since $L^* \leq T$ then $\mathcal{O}(\sqrt{KL^*}) \leq \mathcal{O}(\sqrt{KT})$. Therefore NeuFastCB is expected to perform better in most settings "in practice", especially when $L^*$ is small, i.e., the best policy has low regret. Also note that going by the upper bounds on the regret, especially the dependence on $K$, NeuSquareCB could outperform NeuFastCB only if $L^* = \Theta(T)$ and $K >> T$.

**Remark 4.2.** In the linear setting, (Azoury and Warmuth, 2001) gives $\mathrm{R}_{\mathrm{Sq}}(T) \leq \mathcal{O}(p \log(T/p))$, where $p$ is the feature dimension (Section 2.3, Foster and Rakhlin (2020)). This translates to $\mathrm{Reg}_{\mathrm{CB}}(T) \leq \tilde{\mathcal{O}}(\sqrt{pKT})$. Further with KL loss, using continuous exponential weights gives $\mathbb{R}_{\mathrm{KL}}(T) = \mathcal{O}(p \log T/p)$ which translates to $\mathrm{Reg}_{\mathrm{CB}}(T) \leq \mathcal{O}(\sqrt{L^* Kp \log T/p} + Kp \log T/p)$ (Section 4, Foster and Krishnamurthy (2021)). However, with over-parameterized networks, (with $p >> T$), both bounds are $\Omega(T)$. Therefore it becomes essential to obtain regret bounds that are independent of the number of parameters in the network, which our results do.

## 5 EXPERIMENTS

In this section, we evaluate the performance of NeuSquareCB and NeuFastCB, without output perturbation against some popular NeuCB algorithms. We briefly describe the settings and the baselines considered here. For more details, a scaled-up version of Figure 1 and a discussion on the effect of output perturbation, see Appendix G.

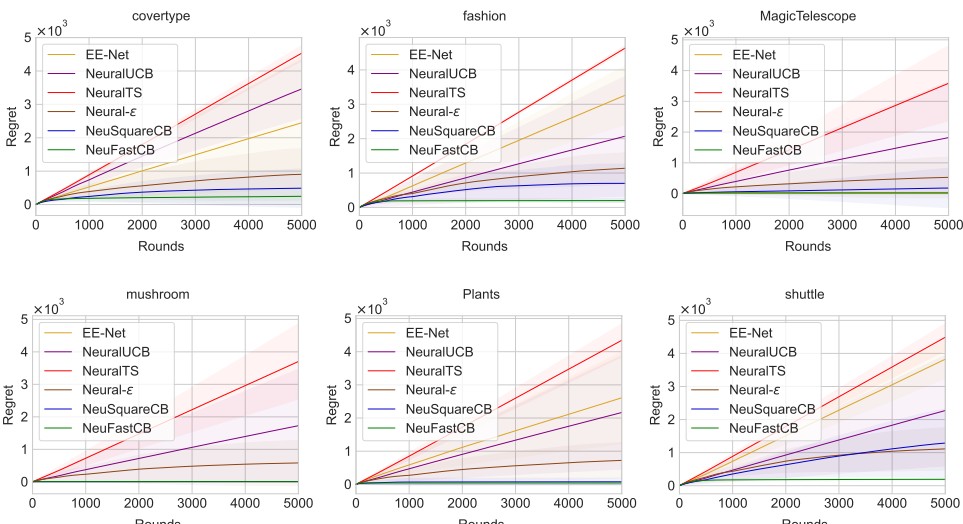

Figure 1: Comparison of cumulative regret of NeuSquareCB and NeuFastCB with baselines on openml datasets (averaged over 20 runs).

**Baselines and Datasets.** For comparison, we choose four NeuCB algorithms: (i) Neural UCB (Zhou et al., 2020) (ii) Neural TS (Zhang et al., 2021), (iii) EE-Net (Ban et al., 2022b) and (iv) Neural-$\epsilon$ greedy. We consider a collection of 6 multiclass classification datasets from the `openml.org` platform: covertype, fashion, MagicTelescope, mushroom, Plants and shuttle. We follow the standard evaluation strategy as in Zhou et al. (2020); Ban et al. (2022b) (see Appendix G for details).

**Adversarial Contexts.** In order to evaluate the performance of the algorithms on adversarially chosen contexts, we feed data points to each of the model in the following manner: for the first 500 rounds, an instance $\mathbf{x}$ is uniformly sampled from each of the classes, transformed into context vectors as described above and presented to the model. We calculate the accuracy for each class by recording the rewards of this class divided by the number of instances drawn from this class. In the subsequent 500 rounds, we increase the probability of sampling instances from the class which had the least accuracy in the previous rounds. We repeat this procedure every 500 rounds.

**Results.** Figure 1 plots the cumulative regret of all the algorithms across different rounds. All experiments were averaged across 20 rounds and the standard deviation is plotted along with the average performance. Although all the algorithms use a neural network to model the potential non-linearity in the reward, the baseline algorithms show erratic performance with a lot of variance. Both our algorithms NeuSquareCB and NeuFastCB show consistent performance across all the datasets. Moreover, NeuFastCB persistently outperforms all baselines for all the datasets.

## 6 CONCLUSION

In this work, we develop novel regret bounds for online regression with neural networks and subsequently give regret guarantees for NeuCBs. We provide a sharp $\mathcal{O}(\log T)$ regret for online regression when the loss satisfies almost convexity, QG condition, and has unique minima. We then propose a network with a small random perturbation, and show that this surprisingly makes the loss satisfy all three conditions. Using these results we obtain $\mathcal{O}(\log T)$ regret bound with both square loss and KL loss and thereafter, convert these bounds to regret bounds for NeuCBs. Separately, we provide lower bound results for Neural UCB (Zhou et al., 2020) and Neural TS (Zhang et al., 2021) and show that even an oblivious adversary can choose a sequence of contexts and a reward function that make their regret bounds $\Omega(T)$. Our algorithms in contrast guarantee $\mathcal{O}(\sqrt{T})$ regret, are efficient to implement, work even for contexts drawn by an adaptive adversary and does not need to store previous networks (unlike (Ban et al., 2022b)). Additionally, our experimental comparisons with the baselines on various datasets further highlight the advantages of our methods and therefore significantly advances the state of the art in NeuCBs from both theoretical and empirical perspectives.

ACKNOWLEDGEMENT

The work was supported in part by grants from the National Science Foundation (NSF) through awards IIS 21-31335, OAC 21-30835, DBI 20-21898, IIS-2002540 as well as a C3.ai research award.

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

## A COMPARISON WITH RECENT NEURAL CONTEXTUAL BANDIT ALGORITHMS

In this section we show that the regret bounds for NeuralUCB (Zhou et al., 2020) and Neural Thompson Sampling (NeuralTS) (Zhang et al., 2021) is $\Omega(T)$ in the worst case. We start with a brief description of the notations used in these works. $\mathbf{H}$ is the Neural Tangent Kernel (NTK) matrix computed from all the context vectors $\mathbf{x}_{t,i}, t \in [T], i \in [K]$, and $h(\mathbf{x})$ is the true reward function given a context $\mathbf{x}$. The cumulative reward vector is defined as $\mathbf{h} = (h(\mathbf{x}_1), \ldots, h(\mathbf{x}_{TK}))^\top$.

With $\lambda$ as the regularization parameter in the loss, the effective dimension $\tilde{d}$ of the Neural Tangent Kernel $\mathbf{H}$ on the contexts $\{\mathbf{x}_t\}_{t=1}^{TK}$ is defined as:

$$\tilde{d} = \frac{\log \det(\mathbf{I} + \mathbf{H}/\lambda)}{\log(1 + T/\lambda)}$$

Further it is assumed that $\mathbf{H} \succeq \lambda_0 \mathbb{I}$ (see Assumption 4.2 in (Zhou et al., 2020) and Assumption 3.4 in (Zhang et al., 2021)).

### A.1 $\Omega(T)$ REGRET FOR NEURALUCB

The bound on the regret for NeuralUCB (Zhou et al., 2020) is given by (ignoring constants):

$$\text{Reg}_{\text{CB}}(T) \leq \sqrt{T}\sqrt{\tilde{d}\log\left(1 + \frac{TK}{\lambda}\right)}\left(\sqrt{\tilde{d}\log\left(1 + \frac{TK}{\lambda}\right)} + \sqrt{\lambda}S\right)$$

$$= \sqrt{T}\left(\tilde{d}\log\left(1 + \frac{TK}{\lambda}\right) + S\sqrt{\tilde{d}\log\left(1 + \frac{TK}{\lambda}\right)\lambda}\right) := \text{B}_{\text{NUCB}}(T, \lambda),$$

where, $\lambda$ is the regularization constant and $S \geq \sqrt{\mathbf{h}^T H^{-1} \mathbf{h}}$ with $\mathbf{h} = (h(\mathbf{x}_1), \ldots, h(\mathbf{x}_{TK}))^T$. We provide two $\Omega(T)$ regret bounds for NeuralUCB.

The first result creates an instance that an oblivious adversary can choose before the algorithm begins such that the regret bound is $\Omega(T)$. The second result provides an $\Omega(T)$ bound for any reward function and set of context vectors as long as $\dfrac{\kappa}{\|\mathbf{h}\|}$ is $\Theta(1)$.

**Theorem A.1.** *There exists a reward vector $\mathbf{h}$ such that the regret bound for NeuralUCB is lower bounded as*

$$\text{B}_{\text{NUCB}}(\lambda, T) \geq \frac{1}{\sqrt{2}}\sqrt{K}T.$$

*Proof.* Consider the eigen-decomposition of $\mathbf{H} = U\Sigma_{\mathbf{H}}U^T$, where columns of $U \in \mathbb{R}^{TK \times TK}$ are the normalized eigen-vectors $\{u_i\}_{i=1}^{TK}$ of $\mathbf{H}$ and $\Sigma_{\mathbf{H}}$ is a diagonal matrix containing the eigenvalues $\{\lambda_i(\mathbf{H})\}_{i=1}^{TK}$ of $H$. Now,

$$S \geq \sqrt{\mathbf{h}^T\mathbf{H}^{-1}\mathbf{h}} = \sqrt{\mathbf{h}^T U\Sigma_{\mathbf{H}}^{-1}U^T\mathbf{h}} = \sqrt{\sum_{i=1}^{TK}\frac{1}{\lambda_i(\mathbf{H})}(u_i^T h)^2} \geq \xi\sqrt{\sum_{i=1}^{TK}\frac{1}{\lambda_i(\mathbf{H})}}, \qquad (18)$$

$$\text{where } \xi = \min_i(u_i^T\mathbf{h})$$

Further observe that we can rewrite the effective dimension as follows:

$$\tilde{d} = \frac{\log\det(\mathbf{I} + \mathbf{H}/\lambda)}{\log(1 + TK/\lambda)} = \frac{\log\left(\prod_{i=1}^{TK}\left(1 + \frac{\lambda_i(\mathbf{H})}{\lambda}\right)\right)}{\log(1 + TK/\lambda)} = \frac{\sum_{i=1}^{TK}\left(1 + \frac{\lambda_i(\mathbf{H})}{\lambda}\right)}{\log(1 + TK/\lambda)}$$

Using these we can lower bound $B_{\text{NUCB}}(T, \lambda)$ as follows:

$$
\begin{aligned}
B_{\text{NUCB}}(T, \lambda) &\geq \sqrt{T}\Bigg( \sum_{i=1}^{TK} \frac{\log\left(1 + \frac{\lambda_i(\mathbf{H})}{\lambda}\right)}{\log\left(1 + \frac{TK}{\lambda}\right)} \log\left(1 + \frac{TK}{\lambda}\right) \\
&\quad + \xi\sqrt{\sum_{i=1}^{TK} \frac{\lambda}{\lambda_i(\mathbf{H})} \sum_{i=1}^{TK} \frac{\log(1 + \frac{\lambda_i(\mathbf{H})}{\lambda})}{\log\left(1 + \frac{TK}{\lambda}\right)} \log\left(1 + \frac{TK}{\lambda}\right)}\Bigg) \\
&= \sqrt{T}\Bigg( \sum_{i=1}^{TK} \log\left(1 + \frac{\lambda_i(\mathbf{H})}{\lambda}\right) + \xi\sqrt{\sum_{i=1}^{TK} \frac{\lambda}{\lambda_i(\mathbf{H})} \sum_{i=1}^{TK} \log\left(1 + \frac{\lambda_i(\mathbf{H})}{\lambda}\right)}\Bigg) \\
&\geq \sqrt{T}\Bigg( \sum_{i=1}^{TK} \log\left(1 + \frac{\lambda_i(\mathbf{H})}{\lambda}\right) + \xi\sqrt{\sum_{i=1}^{TK} \frac{\lambda}{\lambda_i(\mathbf{H})} \log\left(1 + \frac{\lambda_i(\mathbf{H})}{\lambda}\right)}\Bigg) \\
&= \sqrt{T}\Bigg( \sum_{i=1}^{TK} \log\left(1 + \frac{1}{y_i}\right) + \xi\sqrt{\sum_{i=1}^{TK} y_i \log\left(1 + \frac{1}{y_i}\right)}\Bigg),
\end{aligned}
$$

where $y_i = \frac{\lambda}{\lambda_i(\mathbf{H})}$, $i \in [TK]$. Using this we can further bound $B_{\text{NUCB}}(T, \lambda)$ as follows:

$$
\begin{aligned}
B_{\text{NUCB}}(T, \lambda) &\geq \sqrt{T}\left( \sum_{i=1}^{TK} \log\left(1 + \frac{1}{y_i}\right) + \frac{\xi}{\sqrt{TK}} \sum_{i=1}^{TK} \sqrt{y_i \log\left(1 + \frac{1}{y_i}\right)}\right) \\
&\geq \frac{1}{\sqrt{K}}\left( \sum_{i=1}^{TK} \log\left(1 + \frac{1}{y_i}\right) + \xi y_i \log\left(1 + \frac{1}{y_i}\right)\right) \\
&\geq \frac{1}{\sqrt{K}}\left( \sum_{i=1}^{TK} \log\left(1 + \frac{1}{y_i}\right)(1 + \xi y_i)\right) \\
&\geq \frac{1}{\sqrt{K}}\left( \sum_{i=1}^{TK} \frac{\frac{1}{y_i}}{1 + \frac{1}{y_i}}(1 + \xi y_i)\right) \\
&= \frac{1}{\sqrt{K}}\left( \sum_{i=1}^{TK} \frac{1 + \xi y_i}{1 + y_i}\right) \\
&\geq T\sqrt{K}\xi.
\end{aligned}
$$

Recall that $\xi = \min_i (u_i^T \mathbf{h})^2$. Now, consider an $\mathbf{h}$ that makes a $\pi/4$ angle with all the eigen-vectors $u_i$, $i \in [TK]$ and therefore $\xi = \frac{1}{\sqrt{2}}$. Note that for the positive definite assumption of NTK to hold, all the contexts need to be distinct and therefore an oblivious adversary can always choose such an $h$. In such a case, the regret bound for NeuralUCB is $B_{\text{NUCB}}(T, \lambda) = \Omega(T)$. $\qquad\square$

**Remark A.1.** Note that the $\Omega(T)$ regret holds for any $\mathbf{h}$ whose dot product with all the eigen vectors of $\mathbf{H}$ is lower bounded by a constant.

**Theorem A.2.** *For any cumulative reward vector $\mathbf{h}$, with $\kappa$ as the condition number of the NTK matrix, the regret bound for NeuralUCB is*

$$
B_{\text{NUCB}}(\lambda, T) \geq \frac{\|\mathbf{h}\|_2}{\sqrt{\kappa}} \sqrt{K}T.
$$

*Proof.* Using the same notation as in the previous section, we can lower bound $S$ and $\tilde{d}$ as follows:

$$S \geq \sqrt{\lambda_{\min}(\mathbf{H}^{-1})\|\mathbf{h}\|_2^2} = \frac{1}{\sqrt{\lambda_{\max}(\mathbf{H})}}\|\mathbf{h}\|_2$$

$$\tilde{d} = \frac{\log\det(\mathbf{I}+\mathbf{H}/\lambda)}{\log(1+TK/\lambda)} \geq \frac{\log\left(\lambda_{\min}\left(\mathbf{I}+\mathbf{H}/\lambda\right)^{TK}\right)}{\log\left(1+TK/\lambda\right)} \geq TK\frac{\log(1+\frac{\lambda_0}{\lambda})}{\log\left(1+\frac{TK}{\lambda}\right)}$$

Using these we can lower bound $\mathrm{B_{NUCB}}(T,\lambda)$ as follows:

$$\mathrm{B_{NUCB}}(T,\lambda) \geq \sqrt{T}\Bigg(TK\frac{\log(1+\frac{\lambda_0}{\lambda})}{\log\left(1+\frac{TK}{\lambda}\right)}\log\left(1+\frac{TK}{\lambda}\right)$$

$$+ \|\mathbf{h}\|_2\sqrt{TK\frac{\lambda}{\lambda_{\max}(H)}\frac{\log(1+\frac{\lambda_0}{\lambda})}{\log\left(1+\frac{TK}{\lambda}\right)}\log\left(1+\frac{TK}{\lambda}\right)}\Bigg)$$

$$= T\sqrt{K}\left(\sqrt{TK}\log\left(1+\frac{\lambda_0}{\lambda}\right) + \frac{\|\mathbf{h}\|_2}{\sqrt{\kappa}}\sqrt{\frac{\lambda}{\lambda_0}\log\left(1+\frac{\lambda_0}{\lambda}\right)}\right)$$

$$= T\sqrt{K}\left(\sqrt{TK}\log\left(1+\frac{1}{y}\right) + \frac{\|\mathbf{h}\|_2}{\sqrt{\kappa}}\sqrt{y\log\left(1+\frac{1}{y}\right)}\right),$$

where $y = \frac{\lambda}{\lambda_0}$ and $\kappa = \frac{\lambda_{\max(H)}}{\lambda_{\min(H)}}$ is the condition number of of the NTK. Since $y\log(1+\frac{1}{y}) \leq 1$ and for $T, K \geq 1$, we have

$$\mathrm{B_{NUCB}}(T,\lambda) \geq T\sqrt{K}\left(\log\left(1+\frac{1}{y}\right)\left(1+\frac{y\|\mathbf{h}\|_2}{\sqrt{\kappa}}\right)\right) \tag{19}$$

$$\geq T\sqrt{K}\left(\frac{1+\frac{\|\mathbf{h}\|_2}{\sqrt{\kappa}}y}{1+y}\right) \tag{20}$$

$$\geq T\sqrt{K}\frac{\|\mathbf{h}\|_2}{\sqrt{\kappa}} \tag{21}$$

If $\frac{\kappa}{\|\mathbf{h}\|}$ is $\Theta(1)$ then $\mathrm{B_{NUCB}}(T,\lambda)$ is $\Omega(T)$. $\qquad\square$

## A.2 $\Omega(T)$ REGRET FOR NEURAL THOMPSON SAMPLING

The bound on the regret for Neural Thompson sampling (Zhang et al., 2021) is given by (ignoring constants):

$$\mathrm{Reg_{CB}}(T) \leq \sqrt{T}(1+\sqrt{\log T + \log K})\left(S + \sqrt{\tilde{d}\log(1+TK/\lambda)}\right)\sqrt{\lambda\tilde{d}\log(1+TK)}$$

$$:= B(T)$$

We present two lower bounds on $\mathrm{B_{NTS}}(T)$ below. As in NeuralUCB, the first result creates an instance that an oblivious adversary can choose before the algorithm begins such that the regret bound is $\Omega(T)$ and the second result provides an $\Omega(T)$ bound for any reward function and set of context vectors as long as $\lambda_0$ is $\Theta(1)$.

**Theorem A.3.** *There exists a reward vector $\mathbf{h}$ such that the regret bound for NeuralUCB is lower bounded as*

$$\mathrm{B_{NUCB}}(\lambda,T) \geq \frac{1}{2\sqrt{2}}\sqrt{K}T.$$

*Proof.* For Neural Thompson sampling the regularization parameter $\lambda$ is chosen to be $\lambda = 1+1/T$ (see Theorem 3.5 in Zhang et al. (2021)) and therefore $1 \leq \lambda \leq 2$ for any $T \geq 1$. Therefore we can

lower bound the effective dimension as,

$$\tilde{d} = \frac{\log \det(\mathbf{I} + \mathbf{H}/\lambda)}{\log(1 + TK/\lambda)} = \frac{\log \left( \prod_{i=1}^{TK} \left( 1 + \frac{\lambda_i(\mathbf{H})}{\lambda} \right) \right)}{\log(1 + TK/\lambda)}$$

$$= \frac{\sum_{i=1}^{TK} \log \left( 1 + \frac{\lambda_i(\mathbf{H})}{\lambda} \right)}{\log(1 + TK/\lambda)} \geq \frac{\sum_{i=1}^{TK} \log \left( 1 + \frac{\lambda_i(\mathbf{H})}{2} \right)}{\log(1 + TK)}$$

$$\mathrm{B_{NTS}}(T) \geq \sqrt{T} \left( \sqrt{\mathbf{h}^T \mathbf{H}^{-1} \mathbf{h}} \sqrt{\sum_{i=1}^{TK} \log \left( 1 + \frac{\lambda_i(\mathbf{H})}{2} \right)} + \sum_{i=1}^{TK} \log \left( 1 + \frac{\lambda_i(\mathbf{H})}{2} \right) \right)$$

$$\geq \sqrt{T} \left( \xi \sqrt{\sum_{i=1}^{TK} \frac{1}{\lambda_i(\mathbf{H})} \sum_{i=1}^{TK} \log \left( 1 + \frac{\lambda_i(\mathbf{H})}{2} \right)} + \sum_{i=1}^{TK} \log \left( 1 + \frac{\lambda_i(\mathbf{H})}{2} \right) \right)$$

where recall from *equation* 18 that

$$S \geq \xi \sqrt{\sum_{i=1}^{TK} \frac{1}{\lambda_i(\mathbf{H})}},$$

$$\text{where } \xi = \min_i (u_i^T \mathbf{h})$$

Therefore

$$\mathrm{B_{NTS}}(T) \geq \sqrt{T} \left( \xi \sqrt{\sum_{i=1}^{TK} y_i \log \left( 1 + \frac{1}{2y_i} \right)} + \sum_{i=1}^{TK} \log \left( 1 + \frac{1}{2y_i} \right) \right)$$

$$\geq \sqrt{T} \left( \xi \frac{1}{\sqrt{TK}} \sum_{i=1}^{TK} \sqrt{y_i \log \left( 1 + \frac{1}{2y_i} \right)} + \sum_{i=1}^{TK} \log \left( 1 + \frac{1}{2y_i} \right) \right)$$

$$\geq \sqrt{T} \left( \frac{1}{\sqrt{TK}} \sum_{i=1}^{TK} \xi y_i \log \left( 1 + \frac{1}{2y_i} \right) + \log \left( 1 + \frac{1}{2y_i} \right) \right)$$

$$= \sqrt{T} \left( \frac{1}{\sqrt{TK}} \sum_{i=1}^{TK} \log \left( 1 + \frac{1}{2y_i} \right) (1 + \xi y_i) \right)$$

$$\geq \sqrt{T} \left( \frac{1}{\sqrt{TK}} \sum_{i=1}^{TK} \frac{1/2y_i}{1 + 1/2y_i} (1 + \xi y_i) \right)$$

$$= \sqrt{T} \left( \frac{1}{\sqrt{TK}} \sum_{i=1}^{TK} \frac{1 + \xi y_i}{1 + 2y_i} \right)$$

$$\geq \sqrt{T} \frac{1}{\sqrt{TK}} TK \frac{\xi}{2}$$

$$= T\sqrt{K} \frac{\xi}{2}.$$

As in the proof of Theorem A.1, for $\mathbf{h}$ making an angle of $\pi/4$ with all eigen-vectors of $\mathbf{H}$, $\xi \geq \frac{1}{\sqrt{2}}$, which proves the claim.

**Theorem A.4.** *For any cumulative reward vector* $\mathbf{h}$*, the regret bound for Neural Thompson sampling is*

$$\mathrm{B_{NTS}}(\lambda, T) \geq T\sqrt{T} K \frac{\lambda_0}{2 + \lambda_0}$$

Recall, $1 \leq \lambda \leq 2$ for any $T \geq 1$, and therefore we can lower bound the effective dimension as,

$$\tilde{d} = \frac{\log \det(I + \mathbf{H}/\lambda)}{\log(1 + TK/\lambda)} \geq TK \frac{\log(1 + \lambda_0/\lambda)}{\log(1 + TK/\lambda)} \geq TK \frac{\log(1 + \lambda_0/\lambda)}{\log(1 + TK)}$$

Therefore, using $1 \leq \lambda \leq 2$ and $S \geq \sqrt{\mathbf{h}^T \mathbf{H}^{-1} \mathbf{h}}$,

$$\begin{aligned}
\mathrm{B_{NTS}}(T) &\geq \sqrt{T} \left( \sqrt{\mathbf{h}^T \mathbf{H}^{-1} \mathbf{h}} + \sqrt{TK \log(1 + \lambda_0/2)} \right) \sqrt{TK \log(1 + \lambda_0/2)} \\
&\geq T\sqrt{T} K \log(1 + \lambda_0/2) \\
&\geq T\sqrt{T} K \frac{\lambda_0/2}{1 + \lambda_0/2} \\
&\geq T\sqrt{T} K \frac{\lambda_0}{2 + \lambda_0}
\end{aligned}$$

If $\lambda_0 = \Theta(1)$, $\mathrm{B_{NTS}}(T)$ is $\Omega(T)$.

$\square$

## B  BACKGROUND AND PRELIMINARIES FOR TECHNICAL ANALYSIS

Before we proceed with the proof of the claims in Section 3, we state a few recent results from Banerjee et al. (2023) that we will use throughout our proofs. We assume the loss to be the squared loss throughout this subsection.

**Lemma 1** (**Hessian Spectral Norm Bound**, Theorem 4.1 and Lemma 4.1 in Banerjee et al. (2023) ). *Under Assumptions 3 and 4, for any $\mathbf{x} \in \mathcal{X}$, $\theta \in B_{\rho,\rho_1}^{\mathrm{Spec}}(\theta_0)$, with probability at least $(1 - \frac{2(L+1)}{m})$, we have*

$$\left\| \nabla_\theta^2 f(\theta; \mathbf{x}) \right\|_2 \leq \frac{c_H}{\sqrt{m}} \quad \text{and} \quad \|\nabla_\theta f(\theta; \mathbf{x})\|_2 \leq \varrho \,, \tag{22}$$

*where,*

$$c_H = L(L^2 \gamma^{2L} + L\gamma^L + 1) \cdot (1 + \rho_1) \cdot \psi_H \cdot \max_{l \in [L]} \gamma^{L-l} + L\gamma^L \max_{l \in [L]} h(l) \,,$$

$$\gamma = \sigma_1 + \frac{\rho}{\sqrt{m}}, \quad h(l) = \gamma^{l-1} + |\phi(0)| \sum_{i=1}^{l-1} \gamma^{i-1},$$

$$\psi_H = \max_{1 \leq l_1 < l_2 \leq L} \left\{ \beta_\phi h(l_1)^2 \,, \ h(l_1) \left( \frac{\beta_\phi}{2}(\gamma^2 + h(l_2)^2) + 1 \right) \,, \ \beta_\phi \gamma^2 h(l_1) h(l_2) \right\} \,,$$

$$\varrho^2 = (h(L+1))^2 + \frac{1}{m}(1 + \rho_1)^2 \sum_{l=1}^{L+1} (h(l))^2 \gamma^{2(L-l)}.$$

**Lemma 2** (**Loss bounds**, Lemma 4.2 in Banerjee et al. (2023)). *Under Assumptions 3 and 4, for $\gamma = \sigma_1 + \frac{\rho}{\sqrt{m}}$, each of the following inequalities hold with probability at least $\left( 1 - \frac{2(L+1)}{m} \right)$: $\ell(\theta_0) \leq c_{0,\sigma_1}$ and $\ell(\theta) \leq c_{\rho_1,\gamma}$ for $\theta \in B_{\rho,\rho_1}^{\mathrm{Frob}}(\theta_0)$, where $c_{a,b} = 2 \sum_{i=1}^N y_i^2 + 2(1 + a)^2 |g(b)|^2$ and $g(a) = a^L + |\phi(0)| \sum_{i=1}^L a^i$ for any $a, b \in \mathbb{R}$.*

**Lemma 3** (**Loss gradient bound**, Corollary 4.1 in Banerjee et al. (2023)). *Under Assumptions 3 and 4, for $\theta \in B_{\rho,\rho_1}^{\mathrm{Frob}}(\theta_0)$, with probability at least $\left( 1 - \frac{2(L+1)}{m} \right)$, we have $\|\nabla_\theta \ell(\theta)\|_2 \leq 2\sqrt{\ell(\theta)}\varrho \leq 2\sqrt{c_{\rho_1,\gamma}}\varrho$, with $\varrho$ as in Lemma 1 and $c_{\rho_1,\gamma}$ as in Lemma 2.*

**Lemma 4** (**Local Smoothness**, Theorem 5.2 in Banerjee et al. (2023)). *Under Assumptions 3 and 4, with probability at least $(1 - \frac{2(L+1)}{m})$, $\forall \theta, \theta' \in B_{\rho,\rho_1}^{\mathrm{Frob}}(\theta_0)$,*

$$\ell(\theta') \leq \ell(\theta) + \langle \theta' - \theta, \nabla_\theta \hat{\ell}(\theta) \rangle + \frac{\beta}{2} \|\theta' - \theta\|_2^2 \,, \quad \text{with} \quad \beta = b\varrho^2 + \frac{c_H \sqrt{c_{\rho_1,\gamma}}}{\sqrt{m}} \,, \tag{23}$$

*with $c_H$ as in Lemma 1, $\varrho$ as in Lemma 1, and $c_{\rho_1,\gamma}$ as in Lemma 2. Consequently, $\hat{\ell}$ is locally $\beta$-smooth.*

## C   PROOF OF CLAIMS FOR NEURAL ONLINE REGRESSION (SECTION 3)

### C.1   REGRET BOUND UNDER QG CONDITION (PROOF OF THEOREM 3.1)

**Theorem 3.1** (**Regret Bound under QG condition**). *Under Assumptions 2 and 3 the regret of projected OGD with step size $\eta_t = \frac{4}{\mu t}$, where $\mu$ is the QG constant from Assumption 2(b), satisfies*

$$\tilde{R}(T) \leq \mathcal{O}\left(\frac{\lambda^2}{\mu} \log T\right) + \epsilon T + 2 \inf_{\theta \in B} \sum_{t=1}^{T} \ell(y_t, g(\theta; \mathbf{x}_t)). \tag{8}$$

*Proof.* Take any $\theta' \in B$. By Assumption 2(a) (almost convexity) we have

$$\ell(y_t, g(\theta_t; \mathbf{x}_t)) - \ell(y_t, g(\theta'; \mathbf{x}_t)) \leq \langle \theta_t - \theta', \nabla \ell(y_t, g(\theta_t; \mathbf{x}_t)) \rangle + \epsilon$$

Note that for any $\theta \in B$, we have

$$\left\| \prod_B (\theta'') - \theta \right\|_2 \leq \|\theta'' - \theta\|_2.$$

As a result, for $\theta' \in B$, we have

$$\|\theta_{t+1} - \theta'\|_2^2 - \|\theta_t - \theta'\|_2^2 \leq \|\theta_t - \eta_t \nabla \ell(y_t, g(\theta_t; \mathbf{x}_t)) - \theta'\|_2^2 - \|\theta_t - \theta'\|_2^2$$

$$= -2\eta_t \langle \theta_t - \theta', \nabla \ell(y_t, g(\theta_t; \mathbf{x}_t)) \rangle + \eta_t^2 \left\| \nabla \ell(y_t, g(\theta_t; \mathbf{x}_t)) \right\|_2^2$$

$$\leq -2\eta_t \left( \ell(y_t, g(\theta_t; \mathbf{x}_t)) - \ell(y_t, g(\theta'; \mathbf{x}_t)) - \epsilon \right) + \eta_t^2 \lambda^2 \varrho^2 .$$

Rearranging sides and dividing by $2\eta_t$ we get

$$\ell(y_t, g(\theta_t; \mathbf{x}_t)) - \ell(y_t, g(\theta'; \mathbf{x}_t)) \leq \frac{\|\theta_t - \theta'\|_2^2 - \|\theta_{t+1} - \theta'\|_2^2}{2\eta_t} + \frac{\eta_t}{2} \lambda^2 \varrho^2 + \epsilon . \tag{24}$$

Let $\eta_t = \frac{4}{\mu t}$. Then, summing (24) over $t = 1, \ldots, T$, we have

$$\tilde{R}(T) = \sum_{t=1}^{T} \ell(y_t, g(\theta_t; \mathbf{x}_t)) - \sum_{t=1}^{T} \ell(y_t, g(\theta'; \mathbf{x}_t))$$

$$\leq \sum_{t=1}^{T} \|\theta_t - \theta'\|_2^2 \left( \frac{1}{2\eta_t} - \frac{1}{2\eta_{t-1}} \right) + \frac{\lambda^2 \varrho^2}{2} \sum_{t=1}^{T} \eta_t + \epsilon$$

$$\leq \frac{\mu}{8} \sum_{t=1}^{T} \|\theta_t - \theta'\|_2^2 + \mathcal{O}(\lambda^2 \log T) + \frac{c_0 T}{\sqrt{m}}$$

$$\leq \frac{\mu}{4} \sum_{t=1}^{T} \|\theta_t - \theta_t^*\|_2^2 + \frac{\mu}{4} \sum_{t=1}^{T} \|\theta' - \theta_t^*\|_2^2 + \mathcal{O}(\lambda^2 \log T) + \epsilon \tag{25}$$

where $\theta_t^* \in \arg\inf_\theta \ell(y_t, g(\theta; \mathbf{x}_t))$. By Assumption 2(b), the loss satisfies the QG condition and by Assumption 2(c), it has unique minimizer, so that for $t \in [T]$

$$\ell(y_t, g(\theta_t; \mathbf{x}_t)) - \ell(y_t, g(\theta_t^*; \mathbf{x}_t)) \geq \frac{\mu}{2} \|\theta_t - \theta_t^*\|_2^2 , \tag{26}$$

$$\ell(y_t, g(\theta'; \mathbf{x}_t)) - \ell(y_t, g(\theta_t^*; \mathbf{x}_t)) \geq \frac{\mu}{2} \|\theta' - \theta_t^*\|_2^2 . \tag{27}$$

Then, we can lower bound the regret as

$$\tilde{R}(T) = \sum_{t=1}^{T} \ell(y_t, g(\theta_t; \mathbf{x}_t)) - \sum_{t=1}^{T} \ell(y_t, g(\theta'; \mathbf{x}_t))$$

$$= \sum_{t=1}^{T} \left( \ell(y_t, g(\theta_t; \mathbf{x}_t)) - \ell(y_t, g(\theta_t^*; \mathbf{x}_t)) \right) - \sum_{t=1}^{T} \left( \ell(y_t, g(\theta'; \mathbf{x}_t)) - \ell(y_t, g(\theta_t^*; \mathbf{x}_t)) \right)$$

$$\geq \frac{\mu}{2} \sum_{t=1}^{T} \|\theta_t - \theta_t^*\|_2^2 - \sum_{t=1}^{T} \ell(y_t, g(\theta'; \mathbf{x}_t)) , \tag{28}$$

from (26) Multiplying (28) by $-\frac{1}{2}$ and adding to (25), we get

$$
\begin{aligned}
\frac{1}{2}\tilde{R}(T) &= \sum_{t=1}^{T}\ell(y_t, g(\theta_t; \mathbf{x}_t)) - \sum_{t=1}^{T}\ell(y_t, g(\theta'; \mathbf{x}_t)) \\
&\leq \frac{\mu}{4}\sum_{t=1}^{T}\|\theta' - \theta_t^*\|_2^2 + \mathcal{O}(\lambda^2\log T) + \epsilon T + \frac{\sum_{t=1}^{T}\ell(y_t, g(\theta'; \mathbf{x}_t))}{2} \\
&\leq \frac{1}{2}\sum_{t=1}^{T}(\ell(y_t, g(\theta'; \mathbf{x}_t)) - \ell(y_t, g(\theta_t^*; \mathbf{x}_t))) + \mathcal{O}(\lambda^2\log T) + \epsilon T + \frac{\sum_{t=1}^{T}\ell(y_t, g(\theta'; \mathbf{x}_t))}{2}, \\
&\leq \sum_{t=1}^{T}\ell(y_t, g(\theta'; \mathbf{x}_t)) + \mathcal{O}(\lambda^2\log T) + \epsilon T .
\end{aligned}
$$

Since $\theta' \in B$ was arbitrary, taking an infimum completes the proof. $\qquad\square$

## C.2 Regret Bound for Square Loss (Proof of Theorem 3.2)

**Theorem 3.2** (**Regret Bound for square loss**). *Under Assumption 1, 4 and 5 with appropriate choice of step-size sequence $\{\eta_t\}$, width $m$, and perturbation constant $c_p$ in (10), with probability at least $\left(1 - \frac{C}{T^4}\right)$ for some constant $C > 0$, over the randomness of initialization and $\{\varepsilon\}_{s=1}^{S}$, the regret in (12) of projected OGD with loss $\mathcal{L}_{Sq}^{(S)}\left(y_t, \{\tilde{f}(\theta; \mathbf{x}_t, \varepsilon_s)\}_{s=1}^{S}\right)$, $S = \Theta(\log m)$ and projection ball $B_{\rho, \rho_1}^{Frob}(\theta_0)$ with $\rho = \Theta(\sqrt{T}/\lambda_0)$ and $\rho_1 = \Theta(1)$ is given by $\tilde{R}_{Sq}(T) = \mathcal{O}(\log T)$.*

*Proof.* The proof follows along the following four steps.

1. **Square loss is Lipschitz, strongly convex, and smooth w.r.t. the output:**

   This step ensures that Assumption 3 (lipschitz, strong convexity and smoothness) is satisfied. We show that the loss function $\ell_{Sq}$ is lipschitz, strongly convex and smooth with respect to the output $\tilde{f}(\theta; \mathbf{x}, \varepsilon)$ inside $B_{\rho, \rho_1}^{Frob}(\theta_0)$ with high probability over the randomness of initialization and $\{\varepsilon\}_{s=1}^{S}$. We will denote $\lambda_{Sq}, a_{Sq}$ and $b_{Sq}$ as the lipschitz, strong convexity and smoothness parameters respectively as defined in Assumption 3.

   **Lemma 5.** *For $\theta \in B_{\rho, \rho_1}^{Frob}(\theta_0)$, with probability $\left(1 - \frac{2(L+1)+1}{m}\right)$ over the randomness of initialization and $\{\varepsilon\}_{s=1}^{S}$, the loss $\ell_{Sq}\left(y_t, \tilde{f}(\theta; \mathbf{x}_t, \varepsilon)\right)$ is lipschitz, strongly convex and smooth with respect to its output $\tilde{f}(\theta; \mathbf{x}_t, \varepsilon)$. Further the corresponding parameters for square loss, $\lambda_{Sq}, a_{Sq}$ and $b_{Sq}$ are $\mathcal{O}(1)$.*

   Strong convexity and smoothness follow trivially from the definition of the $\ell_{Sq}$. To show that $\ell_{Sq}$ is Lipschitz we show that the output of the neural network $\tilde{f}(\theta; \mathbf{x}, \varepsilon)$ is bounded for any $\theta \in B_{\rho, \rho_1}^{Frob}(\theta_0)$. Also note from Theorem 3.1 that $\lambda^2$ appears in the $\log T$ term and therefore to obtain a $\mathcal{O}(\log T)$ regret we must ensure that $\lambda_{Sq} = \mathcal{O}(1)$, which Lemma 5 does. Note that using a union bound over $t \in [T]$ and Lemma 5 it follows that with probability $\left(1 - \frac{2(L+1)+1}{m}\right)$, $\mathcal{L}_{Sq}^{(S)}\left(y_t, \{\sigma(\tilde{f}(\theta; \mathbf{x}_t, \varepsilon_s))\}_{s=1}^{S}\right)$ is also lipschitz, strongly convex and smooth with respect to each of the outputs $\tilde{f}(\theta; \mathbf{x}_t, \varepsilon_s), s \in [S]$.

2. **The average loss in (13) is almost convex and has a unique minimizer:**

   We show that with $S = \Theta(\log m)$, the average loss in (13) is $\nu$ - Strongly Convex (SC) with $\nu = \mathcal{O}\left(\frac{1}{\sqrt{m}}\right)$ w.r.t. $\theta \in B_{\rho, \rho_1}^{Frob}(\theta_0)$, $\forall t \in [T]$ which immediately implies Assumption 2(a) (almost convexity) and 2(c) (unique minima).

**Lemma 6.** *Under Assumption 5 and $c_p = \sqrt{8\lambda_{Sq}C_H}$, with probability $\left(1 - \frac{2T(L+2)}{m}\right)$ over the randomness of the initialization and $\{\varepsilon_s\}_{s=1}^{S}$, $\mathcal{L}_{Sq}^{(S)}\left(y_t, \{\tilde{f}(\theta; \mathbf{x}_t, \varepsilon_s)\}_{s=1}^{S}\right)$ is $\nu$-strongly convex with respect to $\theta \in B_{\rho,\rho_1}^{\mathrm{Frob}}(\theta_0)$, where $\nu = \mathcal{O}\left(\frac{1}{\sqrt{m}}\right)$.*

3. **The average loss in (13) satisfies the QG condition:**

It is known that square loss with wide networks under Assumption 5 (positive definite NTK) satisfies the PL condition (eg. Liu et al. (2022)) with $\mu = \mathcal{O}(1)$. We show that the average loss in (13) with $S = \Theta(\log m)$, also satisfies the PL condition with $\mu = \mathcal{O}(1)$, which implies that it satisfies the QG condition with same $\mu$ with high probability over the randomness of initialization $\varepsilon_s$.

**Lemma 7.** *Under Assumption 5 with probability $\left(1 - \frac{2T(L+1)C}{m}\right)$, for some absolute constant $C > 0$, $\mathcal{L}_{Sq}^{(S)}\left(y_t, \{\tilde{f}(\theta; \mathbf{x}_t, \varepsilon_s)\}_{s=1}^{S}\right)$ satisfies the QG condition over the randomness of the initialization and $\{\varepsilon_s\}_{s=1}^{S}$, with QG constant $\mu = \mathcal{O}(1)$.*

4. **Bounding the final regret.**

Steps 1 and 2 above surprisingly show that with a small output perturbation, square loss satisfies (a) almost convexity, (b) QG, and (c) unique minima as in Assumption 2. Combining with step 3, we have that all the assumptions of Theorem 3.1 are satisfied by the average loss. Using union bound over the three steps, invoking Theorem 3.1 we get with probability $\left(1 - \frac{T(L+1)C}{m}\right)$ for some absolute constant $C > 0$, with $\tilde{\theta}^* = \min_{\theta \in B_{\rho,\rho_1}^{\mathrm{Frob}}(\theta_0)} \sum_{t=1}^{T} \mathcal{L}_{Sq}^{(S)}\left(y_t, \{\tilde{f}(\theta; \mathbf{x}_t, \varepsilon_s)\}_{s=1}^{S}\right)$,

$$\sum_{t=1}^{T} \mathcal{L}_{Sq}^{(S)}\left(y_t, \{\tilde{f}(\theta_t; \mathbf{x}_t, \varepsilon_s)\}_{s=1}^{S}\right) - \mathcal{L}_{Sq}^{(S)}\left(y_t, \{\tilde{f}(\tilde{\theta}^*; \mathbf{x}_t, \varepsilon_s)\}_{s=1}^{S}\right) \tag{29}$$

$$\leq 2\sum_{t=1}^{T} \mathcal{L}_{Sq}^{(S)}\left(y_t, \{\tilde{f}(\tilde{\theta}^*; \mathbf{x}_t, \varepsilon_s)\}_{s=1}^{S}\right) + \mathcal{O}(\log T). \tag{30}$$

Next we bound $\sum_{t=1}^{T} \mathcal{L}_{Sq}^{(S)}\left(y_t, \{\tilde{f}(\tilde{\theta}^*; \mathbf{x}_t, \varepsilon_s)\}_{s=1}^{S}\right)$ using the following lemma.

**Lemma 8.** *Under Assumption 1 and 5 with probability $\left(1 - \frac{2T(L+1)}{m}\right)$ over the randomness of the initialization and $\{\varepsilon_s\}_{s=1}^{S}$ we have $\sum_{t=1}^{T} \mathcal{L}_{Sq}^{(S)}\left(y_t, \{\tilde{f}(\tilde{\theta}^*; \mathbf{x}_t, \varepsilon_s)\}_{s=1}^{S}\right) = \mathcal{O}(1)$.*

Using Lemma 8 and (29) we have $\sum_{t=1}^{T} \mathcal{L}_{Sq}^{(S)}\left(y_t, \{\tilde{f}(\theta_t; \mathbf{x}_t, \varepsilon_s)\}_{s=1}^{S}\right) = \mathcal{O}(\log T)$ which using (14) and recalling the definition in (13) implies $\tilde{R}_{Sq}(T) = \mathcal{O}(\log T)$

$\square$

Next we prove all the intermediate lemmas in the above steps.

**Lemma 5.** *For $\theta \in B_{\rho,\rho_1}^{\mathrm{Frob}}(\theta_0)$, with probability $\left(1 - \frac{2(L+1)+1}{m}\right)$ over the randomness of initialization and $\{\varepsilon\}_{s=1}^{S}$, the loss $\ell_{Sq}\left(y_t, \tilde{f}(\theta; \mathbf{x}_t, \varepsilon)\right)$ is lipschitz, strongly convex and smooth with respect to its output $\tilde{f}(\theta; \mathbf{x}_t, \varepsilon)$. Further the corresponding parameters for square loss, $\lambda_{Sq}, a_{Sq}$ and $b_{Sq}$ are $\mathcal{O}(1)$.*

*Proof.* We begin by showing that the output of both regularized and un-regularized network defined in (10) and (1) respectively is bounded with high probability over the randomness of the initialization and in expectation over the randomness of $\varepsilon$. Consider any $\theta \in B_{\rho,\rho_1}^{\mathrm{Frob}}(\theta_0)$ and $\mathbf{x} \in \mathcal{X}$. Then, the

output of the network in expectation w.r.t. $\varepsilon_i$'s can be bounded as follows.

$$\mathbb{E}_{\boldsymbol{\varepsilon}}|\tilde{f}(\theta;\mathbf{x}_t,\boldsymbol{\varepsilon})|^2 \leq 2|f(\theta;\mathbf{x})|^2 + 2c_{\text{reg}}^2 \mathbb{E}_{\boldsymbol{\varepsilon}} \left( \sum_{j=1}^{p} \frac{(\theta-\theta_0)^T e_j \varepsilon_j}{m^{1/4}} \right)^2$$

$$= 2|f(\theta;\mathbf{x})|^2 + 2c_{\text{reg}}^2 \left( \sum_{i=1}^{p}\sum_{j=1}^{p} \frac{(\theta-\theta_0)^T e_i e_j^T (\theta-\theta_0)}{\sqrt{m}} \mathbb{E}_{\boldsymbol{\varepsilon}}[\varepsilon_i\varepsilon_j] \right)$$

$$\overset{(a)}{\leq} 2|f(\theta;\mathbf{x})|^2 + 2c_{\text{reg}}^2 \sum_{j=1}^{p} \frac{(\theta-\theta_0)_i^2}{\sqrt{m}}$$

$$= \frac{2}{m}|\mathbf{v}^\top \alpha^{(L)}(\mathbf{x})|^2 + \frac{2c_{\text{reg}}^2}{\sqrt{m}}\|\theta-\theta_0\|^2$$

$$\overset{(b)}{\leq} \frac{2}{m}\|\mathbf{v}\|^2\|\alpha^{(L)}(\mathbf{x})\|^2 + \frac{2}{\sqrt{m}}c_{\text{reg}}^2(L\rho+\rho_1)^2$$

$$\overset{(c)}{\leq} \frac{2}{m}(1+\rho_1)^2\left(\gamma^L + |\phi(0)|\sum_{i=1}^{L}\gamma^{i-1}\right)^2 m + \frac{2}{\sqrt{m}}c_{\text{reg}}^2(L\rho+\rho_1)^2$$

where $\gamma = \sigma_1 + \frac{\rho}{\sqrt{m}}$. Here $(a)$ follows from the fact that $\mathbb{E}[\varepsilon_i\varepsilon_j] = 0$ when $i \neq j$, and $\mathbb{E}[\varepsilon_i\varepsilon_j] = 1$ when $i = j$, $(b)$ follows from $\|\theta-\theta_0\|_2 \leq L\rho + \rho_1$, and $(c)$ holds with probability $\left(1 - \frac{2(L+1)}{m}\right)$ and follows from the fact that $\|\mathbf{v}\| \leq \|\mathbf{v}_0\| + \|\mathbf{v}-\mathbf{v}_0\| \leq 1 + \rho_1$ and thereafter using the arguments in proof of Lemma 4.2 from Banerjee et al. (2023). Finally with a union bound over $t \in [T]$ and using Jensen we get with probability $\left(1 - \frac{2T(L+1)}{m}\right)$

$$\mathbb{E}|\tilde{f}(\theta;\mathbf{x}_t,\boldsymbol{\varepsilon})| \leq \sqrt{\mathbb{E}|\tilde{f}(\theta;\mathbf{x}_t,\boldsymbol{\varepsilon})|^2}$$

$$\leq \sqrt{2(1+\rho_1)^2\left(\gamma^L + |\phi(0)|\sum_{i=1}^{L}\gamma^{i-1}\right)^2 + \frac{2}{\sqrt{m}}c_{\text{reg}}^2(L\rho+\rho_1)^2}. \qquad (31)$$

Now consider $\boldsymbol{\varepsilon} = (\varepsilon_1,\ldots,\varepsilon_{j-1},\varepsilon_j,\varepsilon_{j+1}\ldots,\varepsilon_p)$ and $\boldsymbol{\varepsilon}' = (\varepsilon_1,\ldots,\varepsilon_{j-1},\varepsilon'_j,\varepsilon_{j+1}\ldots,\varepsilon_p)$ that differ only at the $j$-th variable where $\varepsilon_j$ is an independent copy of $\varepsilon'_j$. Now,

$$|\tilde{f}(\theta;\mathbf{x}_t,\boldsymbol{\varepsilon}) - \tilde{f}(\theta;\mathbf{x}_t,\boldsymbol{\varepsilon}')| = \frac{c_{\text{reg}}}{m^{1/4}}|(\theta-\theta_0)^T v_j \varepsilon_j - (\theta-\theta_0)^T v_j \varepsilon'_j|$$

$$\leq \frac{2c_{\text{reg}}}{m^{1/4}}|(\theta-\theta_0)_j|$$

By McDiarmid's inequality we have with probability $(1-\delta)$ over the randomness of $\{\varepsilon_i\}_{i=1}^{p}$

$$\left||\tilde{f}(\theta;\mathbf{x}_t,\boldsymbol{\varepsilon})| - \mathbb{E}|\tilde{f}(\theta;\mathbf{x}_t,\boldsymbol{\varepsilon})|\right| \leq \frac{\sqrt{2}c_{\text{reg}}}{m^{1/4}}\sqrt{\sum_{j=1}^{p}(\theta-\theta_0)_j^2 \ln(1/\delta)}$$

$$= \frac{\sqrt{2}c_{\text{reg}}}{m^{1/4}}\|\theta-\theta_0\|_2\sqrt{\ln(1/\delta)}$$

$$\leq \frac{\sqrt{2}c_{\text{reg}}}{m^{1/4}}(L\rho+\rho_1)\sqrt{\ln(1/\delta)}$$

Taking a union bound over $t \in [T]$, with probability $1-\delta$ over the randomness of $\{\varepsilon_i\}_{i=1}^{p}$ we have

$$\left||\tilde{f}(\theta;\mathbf{x}_t,\boldsymbol{\varepsilon})| - \mathbb{E}|\tilde{f}(\theta;\mathbf{x}_t,\boldsymbol{\varepsilon})|\right| \leq \frac{\sqrt{2}c_{\text{reg}}}{m^{1/4}}(L\rho+\rho_1)\sqrt{\ln(T/\delta)}$$

Choosing $\delta = \frac{1}{m}$ we get with probability $\left(1 - \frac{1}{m}\right)$ over the randomness of $\{\varepsilon_i\}_{i=1}^{p}$

$$\left||\tilde{f}(\theta;\mathbf{x}_t,\boldsymbol{\varepsilon})| - \mathbb{E}|\tilde{f}(\theta;\mathbf{x}_t,\boldsymbol{\varepsilon})|\right| \leq \frac{\sqrt{2}c_{\text{reg}}}{m^{1/4}}(L\rho+\rho_1)\sqrt{\ln(mT)}$$

Combining with (31), we have with probability at least $\left(1 - \frac{2T(L+2)+1}{m}\right)$ over the randomness of the initialization and $\{\varepsilon_i\}_{i=1}^p$ we have

$$|\tilde{f}(\theta; \mathbf{x}_t, \boldsymbol{\varepsilon})| \le \sqrt{2(1+\rho_1)^2 \left(\gamma^L + |\phi(0)| \sum_{i=1}^L \gamma^{i-1}\right)^2 + \frac{2}{\sqrt{m}} c_{\text{reg}}^2 (L\rho + \rho_1)^2}$$
$$+ \frac{\sqrt{2} c_{\text{reg}}}{m^{1/4}} (L\rho + \rho_1) \sqrt{\ln(mT)} \tag{32}$$

Finally taking a union bound over $\{\boldsymbol{\varepsilon}_s\}_{s=1}^S$ for $S = \mathcal{O}(\log m)$ and using the fact that $m = \Omega(T^5 L/\lambda_0^6)$ we get with probability at least $\left(1 - \frac{2T(L+2)+1}{m}\right)$ over the randomness of the initialization and $\{\boldsymbol{\varepsilon}\}_{s=1}^S$, $\forall t \in [T]$, $\mathcal{L}_{\text{Sq}}^{(S)}\left(y_t, \{\tilde{f}(\theta; \mathbf{x}_t, \boldsymbol{\varepsilon}_s)\}_{s=1}^S\right)$ is lipschitz with $\lambda_{\text{Sq}} = \Theta(1)$.

Further $\ell_{\text{Sq}}'' = 1$ and therefore the loss is strongly convex and smooth with $a_{\text{Sq}} = b_{\text{Sq}} = 1$ a.s. over the randomness of $\{\boldsymbol{\varepsilon}_s\}_{s=1}^S$, which completes the proof. $\square$

**Lemma 6.** *Under Assumption 5 and $c_p = \sqrt{8\lambda_{Sq} C_H}$, with probability $\left(1 - \frac{2T(L+2)}{m}\right)$ over the randomness of the initialization and $\{\boldsymbol{\varepsilon}_s\}_{s=1}^S$, $\mathcal{L}_{Sq}^{(S)}\left(y_t, \{\tilde{f}(\theta; \mathbf{x}_t, \boldsymbol{\varepsilon}_s)\}_{s=1}^S\right)$ is $\nu$-strongly convex with respect to $\theta \in B_{\rho,\rho_1}^{\text{Frob}}(\theta_0)$, where $\nu = \mathcal{O}\left(\frac{1}{\sqrt{m}}\right)$.*

*Proof.* From (10) we have a.s.

$$\nabla_\theta \tilde{f}(\theta, \mathbf{x}_t, \boldsymbol{\varepsilon}) = \nabla_\theta f(\theta; \mathbf{x}) + c_{\text{reg}} \sum_{i=1}^p \frac{e_i \varepsilon_i}{m^{1/4}}, \tag{33}$$

$$\nabla_\theta^2 \tilde{f}(\theta, \mathbf{x}_t, \boldsymbol{\varepsilon}) = \nabla_\theta^2 f(\theta; \mathbf{x}). \tag{34}$$

Next, with $\ell_t' = (\tilde{f}(\theta, \mathbf{x}_t, \boldsymbol{\varepsilon}) - y_t)$

$$\nabla_\theta \ell_{\text{Sq}}\big(y_t, \tilde{f}(\theta, \mathbf{x}_t, \boldsymbol{\varepsilon})\big) = \ell_t' \nabla_\theta \tilde{f}(\theta, \mathbf{x}_t, \boldsymbol{\varepsilon})$$
$$\nabla_\theta^2 \ell_{\text{Sq}}\big(y_t, \tilde{f}(\theta, \mathbf{x}_t, \boldsymbol{\varepsilon})\big) = \nabla_\theta \tilde{f}(\theta, \mathbf{x}_t, \boldsymbol{\varepsilon}) \nabla_\theta \tilde{f}(\theta, \mathbf{x}_t, \boldsymbol{\varepsilon})^T + \ell_t' \nabla_\theta^2 \tilde{f}(\theta, \mathbf{x}_t, \boldsymbol{\varepsilon})$$

where we have used the fact that $\ell_t'' = 1$, and $\nabla_\theta^2 \tilde{f}(\theta, \mathbf{x}_t, \boldsymbol{\varepsilon}) = \nabla_\theta^2 f(\theta; \mathbf{x}_t)$ from (34).

Consider $u \in \mathcal{S}^{p-1}$, the unit ball in $\mathbb{R}^p$. Then

$$u^T \nabla_\theta^2 \mathcal{L}_{\text{Sq}}^{(S)}\left(y_t, \{\tilde{f}(\theta; \mathbf{x}_t, \boldsymbol{\varepsilon}_s)\}_{s=1}^S\right) u = \frac{1}{S} \sum_{s=1}^S u^T \Bigg[ \nabla_\theta f(\theta; \mathbf{x}_t) \nabla_\theta f(\theta; \mathbf{x}_t)^T$$

$$+ c_{\text{reg}} \sum_{i=1}^p \frac{v_i \nabla_\theta f(\theta; \mathbf{x}_t)^T}{m^{1/4}} \varepsilon_{s,i} + c_{\text{reg}} \sum_{i=1}^p \frac{\nabla_\theta f(\theta; \mathbf{x}_t) v_i^T}{m^{1/4}} \varepsilon_{s,i} + c_{\text{reg}}^2 \sum_{i=1}^p \sum_{j=1}^p \frac{v_i v_j^T}{\sqrt{m}} \varepsilon_{s,i} \varepsilon_{s,j} \Bigg] u$$

$$+ \ell_{t,i}' u^T \nabla_\theta^2 f(\theta; \mathbf{x}_t) u$$

$$= \langle \nabla_\theta f(\theta; \mathbf{x}_t), u \rangle^2 + 2 \langle \nabla_\theta f(\theta; \mathbf{x}_t), u \rangle \frac{c_{\text{reg}}}{m^{1/4}} \frac{1}{S} \sum_{s=1}^S \underbrace{\sum_{i=1}^p u_i \varepsilon_{s,i}}_{\Gamma_s}$$

$$+ \frac{c_{\text{reg}}^2}{\sqrt{m}} \frac{1}{S} \sum_{s=1}^S \underbrace{\sum_{i=1}^p \sum_{j=1}^p u_i u_j \varepsilon_{s,i} \varepsilon_{s,j}}_{\Gamma_s^2} + \ell_{t,i}' u^T \nabla_\theta^2 f(\theta; \mathbf{x}_t) u \tag{35}$$

Notice that $\forall s \in [S]$, $\Gamma_s = \langle u, \varepsilon_s \rangle$ is a weighted sum of Rademacher random variables and therefore is $\|u\|_2 = 1$ sub-gaussian. Using Hoeffding's inequality, for a given $t \in [T]$,

$$P\left\{\frac{1}{S}\sum_{s=1}^{S}\Gamma_s \leq -\omega_1\right\} \leq e^{-S^2\omega_1^2/2}.$$

Taking a union bound we get $t \in [T]$

$$P\left\{\frac{1}{S}\sum_{s=1}^{S}\Gamma_s \leq -\omega_1\right\} \leq Te^{-S^2\omega_1^2/2}.$$

Further if $\Gamma_s$ is $\sigma_s$ sub-gaussian, then $\Gamma_s^2$ is $(\nu^2, \alpha)$ sub-exponential with $\nu = 4\sqrt{2}\sigma_s^2, \alpha = 4\sigma_s^2$ (see Honorio and Jaakkola (2014, Appendix B)) and therefore $\Gamma_s^2$ is $(4\sqrt{2}, 4)$ sub-exponential. Using Bernstein's inequality for a given $t \in [T]$,

$$P\left\{\frac{1}{S}\sum_{s=1}^{S}\Gamma_s^2 - \mathbb{E}\Gamma_s^2 \leq -\omega_2\right\} \leq e^{-\frac{1}{2}\min\left(\frac{S^2\omega_2^2}{32}, \frac{S\omega_2}{4}\right)}$$

Taking a union bound we get for any $t \in [T]$

$$P\left\{\frac{1}{S}\sum_{s=1}^{S}\Gamma_s^2 - \mathbb{E}\Gamma_s^2 \leq -\omega_2\right\} \leq Te^{-\frac{1}{2}\min\left(\frac{S^2\omega_2^2}{32}, \frac{S\omega_2}{4}\right)}$$

Now choosing $\omega_1 = \omega_2 = \frac{1}{2}$, we get for any $t \in [T]$

$$P\left\{\frac{1}{S}\sum_{s=1}^{S}\Gamma_s \leq -\frac{1}{2}\right\} \leq Te^{-S^2/8}.$$

$$P\left\{\frac{1}{S}\sum_{s=1}^{S}\Gamma_s^2 - \mathbb{E}\Gamma_s^2 \leq -\frac{1}{2}\right\} \leq Te^{-\frac{1}{2}\min\left(\frac{S^2}{128}, \frac{S}{8}\right)}$$

$$\leq Te^{-S/8}, \quad \forall S \geq 16$$

Observing that $\mathbb{E}\Gamma_s = 0$, $\mathbb{E}\Gamma_s^2 = 1$, combining with (35) and recalling that with probability $\left(1 - \frac{2(L+1)T}{m}\right)$ over the randomness of initialization, $\|\nabla_\theta f(\theta; \mathbf{x}_t)\|_2 \leq \frac{c_H}{\sqrt{m}}$ we get with probability at least $1 - T(e^{-S^2/8} + e^{-S/8}) - \frac{2(L+1)T}{m}$

$$u^T\nabla_\theta^2\mathcal{L}_{\mathrm{Sq}}^{(S)}\left(y_t, \{\tilde{f}(\theta; \mathbf{x}_t, \varepsilon_s)\}_{s=1}^{S}\right)u \geq \underbrace{\langle\nabla_\theta f(\theta; \mathbf{x}_t), u\rangle^2 - \langle\nabla_\theta f(\theta; \mathbf{x}_t), u\rangle\frac{c_{\mathrm{reg}}}{m^{1/4}}}_{I} + \frac{c_{\mathrm{reg}}^2}{2\sqrt{m}} - \frac{\lambda_{\mathrm{Sq}}C_H}{\sqrt{m}}.$$

Term $I$ is minimized for $\langle\nabla_\theta f(\theta; \mathbf{x}_t), u\rangle = \frac{c_{\mathrm{reg}}}{2m^{1/4}}$ and the minimum value is $-\frac{c_{\mathrm{reg}}^2}{4\sqrt{m}}$. Substituting this back to the above equation we get

$$u^T\nabla_\theta^2\mathcal{L}_{\mathrm{Sq}}^{(S)}\left(y_t, \{\tilde{f}(\theta; \mathbf{x}_t, \varepsilon_s)\}_{s=1}^{S}\right)u \geq -\frac{c_{\mathrm{reg}}^2}{4\sqrt{m}} + \frac{c_{\mathrm{reg}}^2}{2\sqrt{m}} - \frac{\lambda_{\mathrm{Sq}}C_H}{\sqrt{m}}$$

$$= \frac{c_{\mathrm{reg}}^2}{4\sqrt{m}} - \frac{\lambda_{\mathrm{Sq}}C_H}{\sqrt{m}}$$

$$= \frac{\lambda_{\mathrm{Sq}}C_H}{\sqrt{m}}$$

where the last equality follows because $c_{\mathrm{reg}}^2 = 8\lambda_{\mathrm{Sq}}C_H$. Since $S^2/8 \geq S/8$ for $S \geq 1$, we have for $S \geq 16$ with probability $1 - 2Te^{-S/8} - \frac{2(L+1)T}{m}$

$$u^T\nabla_\theta^2\mathcal{L}_{\mathrm{Sq}}^{(S)}\left(y_t, \{\tilde{f}(\theta; \mathbf{x}_t, \varepsilon_s)\}_{s=1}^{S}\right)u \geq \frac{\lambda_{\mathrm{Sq}}C_H}{\sqrt{m}} > 0.$$

Choosing $S = \max\{8 \log m, 16\}$, we have with probability $\left(1 - \frac{2T(L+2)}{m}\right)$ over the randomness of initialization and $\{\varepsilon_s\}_{s=1}^{S}$ we have

$$u^T \nabla_\theta^2 \mathcal{L}_{\text{Sq}}^{(S)}\left(y_t, \{\tilde{f}(\theta; \mathbf{x}_t, \varepsilon_s)\}_{s=1}^{S}\right) u \geq \frac{\lambda_{\text{Sq}} C_H}{\sqrt{m}} > 0.$$

which completes the proof. $\qquad\square$

**Lemma 7.** *Under Assumption 5 with probability $\left(1 - \frac{2T(L+1)C}{m}\right)$, for some absolute constant $C > 0$, $\mathcal{L}_{Sq}^{(S)}\left(y_t, \{\tilde{f}(\theta; \mathbf{x}_t, \varepsilon_s)\}_{s=1}^{S}\right)$ satisfies the QG condition over the randomness of the initialization and $\{\varepsilon_s\}_{s=1}^{S}$, with QG constant $\mu = \mathcal{O}(1)$.*

*Proof.* We have

$$\left\|\nabla \mathcal{L}_{\text{Sq}}^{(S)}\left(y_t, \{\tilde{f}(\theta; \mathbf{x}_t, \varepsilon_s)\}_{s=1}^{S}\right)\right\|_2^2 = \left\|\frac{1}{S}\sum_{s=1}^{S} \nabla \mathcal{L}_{\text{Sq}}\left(y_t, \tilde{f}(\theta; \mathbf{x}_t, \varepsilon_s)\right)\right\|_2^2$$

$$= \frac{1}{S^2}\sum_{s=1}^{S}\sum_{s'=1}^{S}(\tilde{f}(\theta; \mathbf{x}_t, \varepsilon_s) - y_t)(\tilde{f}(\theta; \mathbf{x}_t, \varepsilon_s) - y_t)\left\langle \nabla \tilde{f}(\theta; \mathbf{x}_t, \varepsilon_s), \nabla \tilde{f}(\theta; \mathbf{x}_t, \varepsilon_{s'})\right\rangle$$

$$= \frac{1}{S^2}(F(\theta; \mathbf{x}_t) - y_t \mathbb{1}_S)^T \tilde{K}(\theta, \mathbf{x}_t)(F(\theta; \mathbf{x}_t) - y_t \mathbb{1}_S) \tag{36}$$

where $F(\theta, \mathbf{x}_t) : \mathbb{R}^{p \times d} \to \mathbb{R}^S$ such that $(F(\theta, \mathbf{x}_t))_s = \tilde{f}(\theta; \mathbf{x}_t, \varepsilon_s)$, $\mathbb{1}_S$ is an $S$-dimensional vector of $1's$ and $\tilde{K}(\theta, \mathbf{x}_t) = \left[\langle \nabla_\theta \tilde{f}(\theta; \mathbf{x}_t, \varepsilon_s), \nabla_\theta \tilde{f}(\theta; \mathbf{x}_t, \varepsilon'_s)\rangle\right]$. $\qquad\square$

Now,

$$\tilde{K}(\theta, \mathbf{x}_t)_{s,s'} = \left\langle \nabla f(\theta; \mathbf{x}_t) + \frac{c_{\text{reg}}}{m^{1/4}}\sum_{j=1}^{p} e_j \varepsilon_{s,j}, f(\theta; \mathbf{x}_t) + \frac{c_{\text{reg}}}{m^{1/4}}\sum_{j=1}^{p} e_j \varepsilon_{s',j}\right\rangle$$

$$= \left\langle \nabla f(\theta; \mathbf{x}_t) + \frac{c_{\text{reg}}}{m^{1/4}}\bar{\varepsilon}_s, f(\theta; \mathbf{x}_t) + \frac{c_{\text{reg}}}{m^{1/4}}\bar{\varepsilon}_{s'}\right\rangle$$

$$= \left\langle \nabla f(\theta; \mathbf{x}_t), \nabla f(\theta; \mathbf{x}_t)\right\rangle + \frac{c_{\text{reg}}}{m^{1/4}}\langle \bar{\varepsilon}_s, \nabla f(\theta; \mathbf{x}_t)\rangle$$

$$+ \frac{c_{\text{reg}}}{m^{1/4}}\langle \bar{\varepsilon}_{s'}, \nabla f(\theta; \mathbf{x}_t)\rangle + \frac{c_{\text{reg}}^2}{\sqrt{m}}\langle \bar{\varepsilon}_s, \bar{\varepsilon}_{s'}\rangle$$

where $\bar{\varepsilon}_s \in \mathbb{R}^S$ with $(\bar{\varepsilon}_s)_j = \varepsilon_{s,j}$. Therefore,

$$\tilde{K}(\theta, \mathbf{x}_t) = \left\langle \nabla f(\theta; \mathbf{x}_t), \nabla f(\theta; \mathbf{x}_t)\right\rangle \mathbb{1}_S \mathbb{1}_S^T + \frac{c_{\text{reg}}^2}{\sqrt{m}}\varepsilon_M^T \varepsilon_M$$

$$+ \frac{c_{\text{reg}}}{m^{1/4}}\begin{pmatrix} \nabla f(\theta; \mathbf{x}_t)^T \bar{\varepsilon}_1 & \nabla f(\theta; \mathbf{x}_t)^T \bar{\varepsilon}_2 & \cdots & \nabla f(\theta; \mathbf{x}_t)^T \bar{\varepsilon}_S \\ \nabla f(\theta; \mathbf{x}_t)^T \bar{\varepsilon}_1 & \nabla f(\theta; \mathbf{x}_t)^T \bar{\varepsilon}_2 & \cdots & \nabla f(\theta; \mathbf{x}_t)^T \bar{\varepsilon}_S \\ & & \vdots & \\ \nabla f(\theta; \mathbf{x}_t)^T \bar{\varepsilon}_1 & \nabla f(\theta; \mathbf{x}_t)^T \bar{\varepsilon}_2 & \cdots & \nabla f(\theta; \mathbf{x}_t)^T \bar{\varepsilon}_S \end{pmatrix}$$

$$+ \frac{c_{\text{reg}}}{m^{1/4}}\begin{pmatrix} \nabla f(\theta; \mathbf{x}_t)^T \bar{\varepsilon}_1 & \nabla f(\theta; \mathbf{x}_t)^T \bar{\varepsilon}_1 & \cdots & \nabla f(\theta; \mathbf{x}_t)^T \bar{\varepsilon}_1 \\ \nabla f(\theta; \mathbf{x}_t)^T \bar{\varepsilon}_2 & \nabla f(\theta; \mathbf{x}_t)^T \bar{\varepsilon}_2 & \cdots & \nabla f(\theta; \mathbf{x}_t)^T \bar{\varepsilon}_2 \\ \vdots & & & \vdots \\ \nabla f(\theta; \mathbf{x}_t)^T \bar{\varepsilon}_S & \nabla f(\theta; \mathbf{x}_t)^T \bar{\varepsilon}_S & \cdots & \nabla f(\theta; \mathbf{x}_t)^T \bar{\varepsilon}_S \end{pmatrix}$$

where $\varepsilon_M$ is an $p \times S$ matrix whose columns are $\bar{\varepsilon}_s$. Next we give a uniform bound on the smallest eigenvalue of $\tilde{K}(\theta, \mathbf{x}_t)$. Consider $u \in \mathcal{S}^{S-1}$, unit sphere in $\mathbb{R}^S$. We have

$$u^T \tilde{K}(\theta, \mathbf{x}_t)u = \underbrace{\langle \nabla f(\theta; \mathbf{x}_t), \nabla f(\theta; \mathbf{x}_t) \rangle u^T \mathbb{1}_S \mathbb{1}_S^T u}_{I} + \underbrace{\frac{c_{\text{reg}}^2}{\sqrt{m}} u^T \varepsilon_M^T \varepsilon_M u}_{II}$$
$$+ \underbrace{\frac{c_{\text{reg}}}{m^{1/4}} \sum_{s=1}^{S} \sum_{s'=1}^{S} u_s u_{s'} \left( \nabla f(\theta; \mathbf{x}_t)^T \bar{\varepsilon}_s + \nabla f(\theta; \mathbf{x}_t)^T \bar{\varepsilon}_{s'} \right)}_{III}$$

Consider term $I$. We can lower bound it as follows

$$\langle \nabla f(\theta; \mathbf{x}_t), \nabla f(\theta; \mathbf{x}_t) \rangle u^T \mathbb{1}_S \mathbb{1}_S^T u \geq \langle \nabla f(\theta; \mathbf{x}_t), \nabla f(\theta; \mathbf{x}_t) \rangle \geq \frac{\lambda_0}{2}.$$

where the first inequality follows because $\lambda_{\min}(\mathbb{1}_S \mathbb{1}_S^T) \geq 0$ and the second inequality because for any $\theta \in B_{\rho, \rho_1}^{\text{Frob}}(\theta_0)$ with probability $\left(1 - \frac{2(L+1)}{m}\right)$ over the initialization

$$\langle \nabla f(\theta; \mathbf{x}_t), \nabla f(\theta; \mathbf{x}_t) \rangle \geq \lambda_{\min}(K_{\text{NTK}}(\theta)) \geq \lambda_0 - \frac{2c_H \varrho T}{\sqrt{m}} L(\rho + \rho_1) \geq \frac{\lambda_0}{2} \geq 0, \qquad (37)$$

where the last inequality holds by choosing $m \geq \frac{16c_H^2 \varrho T^2 (L\rho + \rho_1)^2}{\lambda_0^2} = \Omega(T^3 / \lambda_0^4)$. To see why (37) holds observe that using the exact same argument as in (55) and (52) we have

$$\|K_{\text{NTK}}(\theta) - K_{\text{NTK}}(\theta_0)\|_2 \leq 2\sqrt{T}\varrho \frac{c_H \sqrt{T}}{\sqrt{m}} \|\theta - \theta_0\|_2$$

Using the fact that $\theta \in B_{\rho, \rho_1}^{\text{Frob}}(\theta_0)$ we have

$$\|K_{\text{NTK}}(\theta) - K_{\text{NTK}}(\theta_0)\|_2 \leq \frac{2c_H \varrho T}{\sqrt{m}} L(\rho + \rho_1)$$

and therefore

$$\lambda_{\min}(K_{\text{NTK}}(\theta)) \geq \lambda_{\min}(K_{\text{NTK}}(\theta_0)) - \|K_{\text{NTK}}(\theta) - K_{\text{NTK}}(\theta_0)\|_2$$
$$\geq \lambda_0 - \frac{2c_H \varrho T}{\sqrt{m}} L(\rho + \rho_1).$$

Next consider term $II$. Since every entry of the $p \times S$ matrix $\varepsilon_M$ is i.i.d Rademacher, using Lemma 5.24, Vershynin (2011), it follows that the rows are independent sub-gaussian, with the sub-gaussian norm of the any row $\|(\varepsilon_M)_i\|_{\psi_2} \leq C_i$, where $C_i$ is an absolute constant. Further using Theorem 5.39 from Vershynin (2011) we have with probability at least $1 - 2\exp(-c_M t^2)$

$$\lambda_{\min}(\varepsilon_M^T \varepsilon_M) \geq \sqrt{p} - C_M \sqrt{S} - t$$

where $C_M$ and $c_M$ depend on the maximum of sub-gaussian norm of the rows of $\varepsilon_M$, i.e., $\max_i \|(\varepsilon_M)_i\|_{\psi_2}$, which is an absolute constant. Choosing $t = \sqrt{S}$, for some absolute constant $c$, with probability at least $\left(1 - \frac{c}{m}\right)$, uniformly for any $u \in \mathcal{S}^{S-1}$ we have

$$\lambda_{\min}(\varepsilon_M^T \varepsilon_M) \geq \sqrt{p} - (C_M + 1)\sqrt{S}$$

Noting that $S = \max\{16, 8\log m\}$ and $p = md + (L-1)m^2 + m$ we have

$$\lambda_{\min}(\varepsilon_M^T \varepsilon_M) \geq \frac{m + \sqrt{m}}{2} - (C_M + 1)\sqrt{\max\{16, 8\log m\}}.$$

Using $m \geq 64(C_M + 1)^2$, $\sqrt{m\log m} \geq 8\sqrt{2}(C_M + 1)$, with probability at least $\left(1 - \frac{c}{m}\right)$, uniformly for any $u \in \mathcal{S}^{S-1}$ we have

$$u^T(\varepsilon_M^T \varepsilon_M)u \geq m/2$$

Finally consider term $III$. We have

$$\frac{c_{\text{reg}}}{m^{1/4}} \sum_{s=1}^{S} \sum_{s'=1}^{S} u_s u_{s'} \left( \nabla f(\theta; \mathbf{x}_t)^T \bar{\varepsilon}_s + \nabla f(\theta; \mathbf{x}_t)^T \bar{\varepsilon}_{s'} \right) \geq \frac{2\sqrt{S} c_{\text{reg}}}{m^{1/4}} \sum_{s=1}^{S} u_s \nabla f(\theta; \mathbf{x}_t)^T \varepsilon_s.$$

Since each $\varepsilon_{s,j}$ is 1 sub-gaussian, therefore $\nabla f(\theta; \mathbf{x}_t)^T \varepsilon_s$ is $\|\nabla f(\theta; \mathbf{x}_t)\|_2 \, (\leq \varrho)$ sub-gaussian with zero mean. Using Hoeffding we have with probability at least $1 - \exp(-S^2 t^2)$

$$\frac{2\sqrt{S} c_{\text{reg}}}{m^{1/4}} \sum_{s=1}^{S} u_s \nabla f(\theta; \mathbf{x}_t)^T \varepsilon_s \geq -\frac{2\sqrt{S} c_{\text{reg}}}{m^{1/4}} St$$

Choosing $t = \frac{1}{8}$ and noting that $S = \max\{16, 8\log m\}$, we get with probability at least $\left(1 - \frac{1}{m^2}\right)$

$$\frac{2\sqrt{S} c_{\text{reg}}}{m^{1/4}} \sum_{s=1}^{S} u_s \nabla f(\theta; \mathbf{x}_t)^T \varepsilon_s \geq \frac{-16 c_{\text{reg}}}{m^{1/4}} (\log m)^{3/2} \geq -1$$

where the last inequality used $m^{1/4} \geq 16 c_{\text{reg}}$ and $m^{1/4} (\log m)^{-3/2} \geq 4\sqrt{2}$. Finally to get a uniform bound over $\mathcal{S}^{S-1}$, we can use a standard $\epsilon$-net argument with $\epsilon = 1/4$ and metric entropy $S \log 9$ (see eg. proof of Theorem 5.39 in Vershynin (2011)). Using $m(\log m)^{-1} \geq 8 \log 9$, with probability at least $\left(1 - \frac{1}{m}\right)$ uniformly for any $u \in \mathcal{S}^{S-1}$ we have

$$\frac{2\sqrt{S} c_{\text{reg}}}{m^{1/4}} \sum_{s=1}^{S} u_s \nabla f(\theta; \mathbf{x}_t)^T \varepsilon_s \geq -1$$

Combining all the terms and using $m \geq 4$, we get with probability $\left(1 - \frac{2(L+1)C}{m}\right)$ for some absolute constant $C > 0$

$$\frac{1}{S^2} (F(\theta; \mathbf{x}_t) - y_t \mathbb{1}_S)^T \tilde{K}(\theta, \mathbf{x}_t)(F(\theta; \mathbf{x}_t) - y_t \mathbb{1}_S) \geq \frac{m}{4S} \frac{1}{S} \left\| (F(\theta; \mathbf{x}_t) - y_t \mathbb{1}_S) \right\|_2$$

$$= \frac{m}{64 \log m} \frac{1}{S} \sum_{s=1}^{S} \mathcal{L}_{\text{Sq}}\left( y_t, \tilde{f}(\theta; \mathbf{x}_t, \varepsilon_s) \right)$$

$$\geq 2\mu \, \mathcal{L}_{\text{Sq}}^{(S)}\left( y_t, \{ \tilde{f}(\theta; \mathbf{x}_t, \varepsilon_s) \}_{s=1}^{S} \right)$$

where $\mu = 128$ where the last inequality follows from $m(\log m)^{-1} \geq 1$. Combining with (36) and using the fact that PL implies QG (see Remark 3.1) along with a union bound over $t \in [T]$ completes the proof.

**Lemma 8.** *Under Assumption 1 and 5 with probability* $\left(1 - \frac{2T(L+1)}{m}\right)$ *over the randomness of the initialization and* $\{\varepsilon_s\}_{s=1}^{S}$ *we have* $\sum_{t=1}^{T} \mathcal{L}_{Sq}^{(S)}\left( y_t, \{ \tilde{f}(\tilde{\theta}^*; \mathbf{x}_t, \varepsilon_s) \}_{s=1}^{S} \right) = \mathcal{O}(1)$.

*Proof.* Consider any $\theta \in B_{\rho,\rho_1}^{\text{Frob}}(\theta_0)$. We have

$$\mathcal{L}_{\text{Sq}}^{(S)}\left( y_t, \{ \tilde{f}(\theta; \mathbf{x}_t, \varepsilon_s) \}_{s=1}^{S} \right) = \frac{1}{S} \sum_{s=1}^{S} \mathcal{L}_{\text{Sq}}\left( y_t, \tilde{f}(\theta; \mathbf{x}_t, \varepsilon_s) \right) = \frac{1}{2S} \sum_{s=1}^{S} \left( y_t - \tilde{f}(\theta; \mathbf{x}_t, \varepsilon_s) \right)^2$$

$$= \frac{1}{2S} \sum_{s=1}^{S} \left( y_t - f(\theta_t; \mathbf{x}_t) - c_{\text{reg}} \sum_{j=1}^{p} \frac{(\theta_t - \theta_0)^T e_j \varepsilon_{s,j}}{m^{1/4}} \right)^2$$

$$= \frac{1}{2} \left[ (y_t - f(\theta_t; \mathbf{x}_t))^2 - 2c_{\text{reg}}(y_t - f(\theta_t; \mathbf{x}_t)) \underbrace{\frac{1}{S} \sum_{s=1}^{S} \left( \sum_{j=1}^{p} \frac{(\theta - \theta_0)^T e_j \varepsilon_{s,j}}{m^{1/4}} \right)}_{I} \right.$$

$$\left. + c_{\text{reg}}^2 \underbrace{\frac{1}{S} \sum_{s=1}^{S} \left( \sum_{j=1}^{p} \frac{(\theta - \theta_0)^T e_j \varepsilon_{s,j}}{m^{1/4}} \right)^2}_{II} \right] \tag{38}$$

Consider term $I$.

$$\frac{1}{S}\sum_{s=1}^{S}\Big(\sum_{j=1}^{p}\frac{(\theta-\theta_0)^T e_j \varepsilon_{s,j}}{m^{1/4}}\Big) = \frac{1}{m^{1/4}}\frac{1}{S}\sum_{s=1}^{S}\underbrace{\Big(\sum_{j=1}^{p}(\theta-\theta_0)_j \varepsilon_{s,j}\Big)}_{\Gamma_s}$$

Since $\varepsilon_{s,j}$ is 1 sub-gaussian, $\Gamma_s$ is $\|\theta-\theta_0\|_2^2$ sub-gaussian. Using Hoeffding's inequality

$$P\bigg\{\frac{1}{S}\sum_{s=1}^{S}\Gamma_s \geq \omega_1\bigg\} \leq e^{-S^2\omega_1^2/2\|\theta-\theta_0\|_2}.$$

Taking a union bound we get $t \in [T]$

$$P\bigg\{\frac{1}{S}\sum_{s=1}^{S}\Gamma_s \geq \omega_1\bigg\} \leq T e^{-S^2\omega_1^2/2\|\theta-\theta_0\|_2}.$$

Next consider term $II$.

$$\frac{1}{S}\sum_{s=1}^{S}\Big(\sum_{j=1}^{p}\frac{(\theta-\theta_0)^T e_j \varepsilon_{s,j}}{m^{1/4}}\Big)^2 = \frac{1}{\sqrt{m}}\frac{1}{S}\sum_{s=1}^{S}\underbrace{\sum_{i=1}^{p}\sum_{j=1}^{p}(\theta-\theta_0)_i(\theta-\theta_0)_j \varepsilon_{s,i}\varepsilon_{s,j}}_{\Gamma_s^2}$$

Further if $\Gamma_s$ is $\sigma_s$ sub-gaussian, then $\Gamma_s^2$ is $(\nu^2,\alpha)$ sub-exponential with $\nu = 4\sqrt{2}\sigma_s^2, \alpha = 4\sigma_s^2$ (see Honorio and Jaakkola (2014, Appendix B)) and therefore $\Gamma_s^2$ is $\big(4\sqrt{2}\|\theta-\theta_0\|_2^2, 4\|\theta-\theta_0\|_2^2\big)$ sub-exponential. Using Bernstein's inequality for a given $t \in [T]$,

$$P\bigg\{\frac{1}{S}\sum_{s=1}^{S}\Gamma_s^2 - \mathbb{E}\Gamma_s^2 \geq \omega_2\bigg\} \leq e^{-\frac{1}{2}\min\Big(\frac{S^2\omega_2^2}{32\|\theta-\theta_0\|_2^4}, \frac{S\omega_2}{4\|\theta-\theta_0\|_2^2}\Big)}$$

Taking a union bound we get for any $t \in [T]$

$$P\bigg\{\frac{1}{S}\sum_{s=1}^{S}\Gamma_s^2 - \mathbb{E}\Gamma_s^2 \geq \omega_2\bigg\} \leq T e^{-\frac{1}{2}\min\Big(\frac{S^2\omega_2^2}{32\|\theta-\theta_0\|_2^4}, \frac{S\omega_2}{4\|\theta-\theta_0\|_2^2}\Big)}$$

Now choosing $\omega_1 = \|\theta-\theta_0\|_2, \omega_2 = \|\theta-\theta_0\|_2^2$, we get for any $t \in [T]$

$$P\bigg\{\frac{1}{S}\sum_{s=1}^{S}\Gamma_s \geq \|\theta-\theta_0\|_2\bigg\} \leq T e^{-S^2/2}.$$

$$P\bigg\{\frac{1}{S}\sum_{s=1}^{S}\Gamma_s^2 - \mathbb{E}\Gamma_s^2 \geq \frac{1}{2}\|\theta-\theta_0\|_2^2\bigg\} \leq T e^{-\frac{1}{2}\min\Big(\frac{S^2}{128},\frac{S}{8}\Big)}$$

$$\leq T e^{-S/8}, \quad \forall S \geq 16$$

Observing that $\mathbb{E}\Gamma_s = 0$, $\mathbb{E}\Gamma_s^2 = \frac{c_{\text{reg}}^2}{\sqrt{m}}\|\theta-\theta_0\|_2^2$, using the fact that $\lambda_{\text{Sq}} = |y_t - f(\theta;\mathbf{x}_t)| = \Theta(1)$ with probability $\big(1 - \frac{2T(L+1)}{m}\big)$ (see proof of Lemma 5) and finally combining with (38) we get with probability at least $1 - T(e^{-S^2/2} + e^{-S/8}) - \frac{2T(L+1)}{m}$

$$\mathcal{L}_{\text{Sq}}^{(S)}\Big(y_t, \{\tilde{f}(\theta;\mathbf{x}_t,\varepsilon_s)\}_{s=1}^{S}\Big) \leq \mathcal{L}_{\text{Sq}}\big(y_t, f(\theta;\mathbf{x}_t)\big) + \frac{c_{\text{reg}}^2}{2\sqrt{m}}\|\theta-\theta_0\|_2^2 + \frac{2c_{\text{reg}}\lambda_{\text{Sq}}}{m^{1/4}}\|\theta-\theta_0\|_2$$

Summing over $t$ and taking a $\min_{\theta \in B_{\rho,\rho_1}^{\text{Frob}}(\theta_0)}$ over the left hand side we get for any $\theta \in B_{\rho,\rho_1}^{\text{Frob}}(\theta_0)$

$$\min_{\theta \in B_{\rho,\rho_1}^{\text{Frob}}(\theta_0)}\sum_{t=1}^{T}\mathcal{L}_{\text{Sq}}^{(S)}\Big(y_t, \{\tilde{f}(\theta;\mathbf{x}_t,\varepsilon_s)\}_{s=1}^{S}\Big) \leq \sum_{t=1}^{T}\mathcal{L}_{\text{Sq}}\big(y_t, f(\theta;\mathbf{x}_t)\big) + \sum_{t=1}^{T}\frac{c_{\text{reg}}^2}{2\sqrt{m}}\|\theta-\theta_0\|_2^2 + \frac{2c_{\text{reg}}\lambda_{\text{Sq}}}{m^{1/4}}\|\theta-\theta_0\|_2$$

From Theorem E.1 we know there exists $\bar{\theta} \in B_{\rho,\rho_1}^{\text{Frob}}(\theta_0)$ such that with probability at least $(1 - \frac{2(L+1)}{m})$ over the randomness of initialization we have $f(\bar{\theta}, \mathbf{x}_t) = y_t$ for any set of $y_t \in [0, 1], t \in [T]$ which implies $\mathcal{L}_{\text{Sq}}\left(y_t, f(\bar{\theta}; \mathbf{x}_t)\right) = 0, \forall t \in [T]$. Therefore

$$\sum_{t=1}^{T} \mathcal{L}_{\text{Sq}}^{(S)}\left(y_t, \left\{\tilde{f}(\theta^*; \mathbf{x}_t, \boldsymbol{\varepsilon}_s)\right\}_{s=1}^{S}\right) \leq 0 + \sum_{t=1}^{T} \frac{c_{\text{reg}}^2}{2\sqrt{m}} \|\theta' - \theta_0\|_2^2 + \frac{2c_{\text{reg}}\lambda_{\text{Sq}}}{m^{1/4}} \|\theta' - \theta_0\|_2$$

$$\overset{(a)}{\leq} \frac{c_{\text{reg}}^2}{2\sqrt{m}}(L\rho + \rho_1)^2 + \frac{2c_{\text{reg}}\lambda_{\text{Sq}}}{m^{1/4}}(L\rho + \rho_1) \overset{(b)}{=} \mathcal{O}(1) \quad (39)$$

where $(a)$ follows because $\bar{\theta}' \in B_{\rho,\rho_1}^{\text{Frob}}(\theta_0)$ implies $\theta' \in B_{L\rho+\rho_1}^{\text{Euc}}(\theta_0)$ and $(b)$ follows by choosing $m = \Omega(T^5 L/\lambda_0^6)$. □

### C.3 PROOF OF THEOREM 3.3

**Theorem 3.3** (**Regret Bound for KL Loss**). *Under Assumption 1, 4 and 5 for $y_t \in [z, 1 - z]$, $0 < z < 1$, with appropriate choice of step-size sequence $\{\eta_t\}$, width $m$, and perturbation constant $c_p$, with high probability over the randomness of initialization and $\{\boldsymbol{\varepsilon}\}_{s=1}^{S}$, the regret of projected OGD with loss $\mathcal{L}_{KL}^{(S)}\left(y_t, \{\sigma(\tilde{f}(\theta; \mathbf{x}_t, \boldsymbol{\varepsilon}_s))\}_{s=1}^{S}\right) = \frac{1}{S}\sum_{s=1}^{S} \ell_{KL}\left(y_t, \sigma(\tilde{f}(\theta; \mathbf{x}_t, \boldsymbol{\varepsilon}_s))\right), S = \Theta(\log m)$ and projection ball $B_{\rho,\rho_1}^{\text{Frob}}(\theta_0)$ with $\rho = \Theta(\sqrt{T}/\lambda_0)$ and $\rho_1 = \Theta(1)$ is given by $R_{KL}(T) = \mathcal{O}(\log T)$.*

*Proof.* The proof of the claim follows along similar lines as in the previous subsection. It consists of the following four steps:

1. **Binary KL loss is lipschitz, strongly convex and smooth w.r.t. the output.**

   We show that $\ell_{\text{KL}}(y_t, \hat{y}_t)$ is lipschitz, strongly convex and smooth with respect to the output $\hat{y}_t = \tilde{f}(\theta; \mathbf{x}_t, \boldsymbol{\varepsilon})$ inside $B_{\rho,\rho_1}^{\text{Frob}}(\theta_0)$ almost surely over the randomness of initialization and $\boldsymbol{\varepsilon}$. We will use $\lambda_{\text{KL}}, a_{\text{KL}}$ and $b_{\text{KL}}$ respectively to denote the lipschitz, strong convexity and smoothness parameter for $\ell_{\text{KL}}(y_t, \hat{y}_t)$ (c.f. Assumption 3(lipschitz, strongly convex and smooth)).

   **Lemma 9.** *For $\theta \in B_{\rho,\rho_1}^{\text{Frob}}(\theta_0)$, the loss $\ell_{KL}\left(y_t, \tilde{f}(\theta; \mathbf{x}_t, \boldsymbol{\varepsilon})\right)$ is lipschitz, strongly convex and smooth with respect to the output $\tilde{f}(\theta; \mathbf{x}_t, \boldsymbol{\varepsilon})$ a.s. over the randomness of initialization and $\boldsymbol{\varepsilon}$. Further the parameters $\lambda_{KL}, a_{KL}$ and $b_{KL}$ are $\mathcal{O}(1)$.*

   Note that Lemma 9 implies that $\mathcal{L}_{\text{KL}}^{(S)}\left(y_t, \{\sigma(\tilde{f}(\theta; \mathbf{x}_t, \boldsymbol{\varepsilon}_s))\}_{s=1}^{S}\right) = \frac{1}{S}\sum_{s=1}^{S} \ell_{\text{KL}}\left(y_t, \sigma(\tilde{f}(\theta; \mathbf{x}_t, \boldsymbol{\varepsilon}_s))\right)$ is also lipschitz, strongly convex and smooth a.s. with respect to each of the outputs $\tilde{f}(\theta; \mathbf{x}_t, \boldsymbol{\varepsilon}_s)$.

2. **The average loss at $t \in [T]$, $\mathcal{L}_{\text{KL}}^{(S)}\left(y_t, \{\sigma(\tilde{f}(\theta; \mathbf{x}_t, \boldsymbol{\varepsilon}_s))\}_{s=1}^{S}\right)$ is almost convext and has a unique minimizer w.r.t. $\theta$.**

   We show that the random perturbation to the output in (10) assures that $\mathcal{L}_{\text{KL}}^{(S)}\left(y_t, \{\sigma(\tilde{f}(\theta; \mathbf{x}_t, \boldsymbol{\varepsilon}_s))\}_{s=1}^{S}\right)$ is strongly convex with respect to $\theta \in B_{\rho,\rho_1}^{\text{Frob}}(\theta_0)$ at every $t \in [T]$ (with a very small $\mathcal{O}\left(\frac{1}{\sqrt{m}}\right)$ strong convexity parameter), which implies that it satisfies 2(a) and 2(c).

   **Lemma 10.** *Under Assumption 5 and $c_p = \sqrt{8\lambda_{Sq}C_H}$, with probability $\left(1 - \frac{2T(L+2)}{m}\right)$ over the randomness of the initialization and $\{\boldsymbol{\varepsilon}_s\}_{s=1}^{S}$, $\mathcal{L}_{KL}^{(S)}\left(y_t, \{\sigma(\tilde{f}(\theta; \mathbf{x}_t, \boldsymbol{\varepsilon}_s))\}_{s=1}^{S}\right)$ is $\nu$-strongly convex with respect to $\theta \in B_{\rho,\rho_1}^{\text{Frob}}(\theta_0)$, where $\nu = \mathcal{O}\left(\frac{1}{\sqrt{m}}\right)$.*

3. **The average loss at $t \in [T]$, $\mathcal{L}_{\text{KL}}^{(S)}\left(y_t, \{\sigma(\tilde{f}(\theta; \mathbf{x}_t, \boldsymbol{\varepsilon}_s))\}_{s=1}^{S}\right)$ satisfies the QG condition.**

We show that the loss $\mathcal{L}_{\mathrm{KL}}^{(S)}\big(y_t, \{\sigma(\tilde{f}(\theta; \mathbf{x}_t, \varepsilon_s))\}_{s=1}^S\big)$ satisfies the QG condition (with QG constant $\mu = \mathcal{O}(1)$) with high probability over the randomness of initialization and $\{\varepsilon_s\}_{s=1}^S$.

**Lemma 11.** *Under Assumption 5 with probability $\big(1 - \frac{2T(L+1)C}{m}\big)$, for some absolute constant $C > 0$, $\mathcal{L}_{KL}^{(S)}\big(y_t, \{\tilde{f}(\theta; \mathbf{x}_t, \varepsilon_s)\}_{s=1}^S\big)$ satisfies the QG condition over the randomness of the initialization and $\{\varepsilon_s\}_{s=1}^S$, with constant $\mu = \mathcal{O}(1)$.*

## 4. Bounding the final regret.

The above three steps ensure that all the assumptions of Theorem 3.1 are satisfied by the loss function $\mathcal{L}_{\mathrm{KL}}^{(S)}\big(y_t, \{\sigma(\tilde{f}(\theta; \mathbf{x}_t, \varepsilon_s))\}_{s=1}^S\big)$. Taking a union over the events in the above three steps, invoking Theorem 3.1, we get with probability $\big(1 - \frac{TLC}{m}\big)$ for some absolute constant $C > 0$,

$$\sum_{t=1}^T \mathcal{L}_{\mathrm{KL}}^{(S)}\Big(y_t, \{\sigma(\tilde{f}(\theta; \mathbf{x}_t, \varepsilon_s))\}_{s=1}^S\Big) - \mathcal{L}_{\mathrm{KL}}^{(S)}\Big(y_t, \{\sigma(\tilde{f}(\theta^*; \mathbf{x}_t, \varepsilon_s))\}_{s=1}^S\Big)$$

$$\leq 2 \sum_{t=1}^T \mathcal{L}_{\mathrm{KL}}^{(S)}\Big(y_t, \{\sigma(\tilde{f}(\tilde{\theta}^*; \mathbf{x}_t, \varepsilon_s))\}_{s=1}^S\Big) + \mathcal{O}(\log T). \tag{40}$$

$$\text{where} \quad \tilde{\theta}^* = \min_{\theta \in B_{\rho,\rho_1}^{\mathrm{Frob}}(\theta_0)} \sum_{t=1}^T \mathcal{L}_{\mathrm{KL}}^{(S)}\Big(y_t, \{\sigma(\tilde{f}(\theta; \mathbf{x}_t, \varepsilon_s))\}_{s=1}^S\Big)$$

Next we bound $\sum_{t=1}^T \mathcal{L}_{\mathrm{KL}}^{(S)}\big(y_t, \{\sigma(\tilde{f}(\theta^*; \mathbf{x}_t, \varepsilon_s))\}_{s=1}^S\big)$ using the following lemma.

**Lemma 12.** *Under Assumption 1 and 5 with probability $\big(1 - \frac{TLC}{m}\big)$ for some absolute constant $C > 0$ over the randomness of the initialization we have*

$$\sum_{t=1}^T \mathcal{L}_{KL}^{(S)}\Big(y_t, \{\sigma(\tilde{f}(\theta^*; \mathbf{x}_t, \varepsilon_s))\}_{s=1}^S\Big) = \mathcal{O}(1).$$

Using Lemma 12 and (40) we have $\sum_{t=1}^T \mathcal{L}_{\mathrm{KL}}^{(S)}\big(y_t, \{\sigma(\tilde{f}(\theta; \mathbf{x}_t, \varepsilon_s))\}_{s=1}^S\big) \leq \mathcal{O}(\log T)$. which implies $\mathrm{R}_{\mathrm{KL}}(T) \leq \mathcal{O}(\log T)$.

$\square$

Next we prove Lemmas 9,10,11 and 12.

**Lemma 9.** *For $\theta \in B_{\rho,\rho_1}^{\mathrm{Frob}}(\theta_0)$, the loss $\ell_{KL}\big(y_t, \tilde{f}(\theta; \mathbf{x}_t, \varepsilon)\big)$ is lipschitz, strongly convex and smooth with respect to the output $\tilde{f}(\theta; \mathbf{x}_t, \varepsilon)$ a.s. over the randomness of initialization and $\varepsilon$. Further the parameters $\lambda_{KL}, a_{KL}$ and $b_{KL}$ are $\mathcal{O}(1)$.*

*Proof.* Consider $\theta \in B_{\rho,\rho_1}^{\mathrm{Spec}}(\theta_0)$. Now,

$$\ell'_{\mathrm{KL}} := \frac{d\ell_{\mathrm{KL}}(y_t, \tilde{f}(\theta, \mathbf{x}_t, \varepsilon))}{d\tilde{f}(\theta, \mathbf{x}_t, \varepsilon)}$$

$$= -\left(\frac{y_t \sigma(\tilde{f}(\theta, \mathbf{x}_t, \varepsilon))(1 - \sigma(\tilde{f}(\theta, \mathbf{x}_t, \varepsilon)))}{\sigma(\tilde{f}(\theta, \mathbf{x}_t, \varepsilon))} - \frac{(1 - y_t)\sigma(\tilde{f}(\theta, \mathbf{x}_t, \varepsilon))(1 - \sigma(\tilde{f}(\theta, \mathbf{x}_t, \varepsilon)))}{1 - \sigma(\tilde{f}(\theta, \mathbf{x}_t, \varepsilon))}\right)$$

$$= -y_t + y_t \sigma(\tilde{f}(\theta, \mathbf{x}_t, \varepsilon)) + \sigma(\tilde{f}(\theta, \mathbf{x}_t, \varepsilon)) - y_t \sigma(\tilde{f}(\theta, \mathbf{x}_t, \varepsilon))$$

$$= \sigma(\tilde{f}(\theta, \mathbf{x}_t, \varepsilon)) - y_t.$$

It follows that $\lambda_{\mathrm{KL}} \leq 2$ a.s. since $\sigma(\tilde{f}(\theta, \mathbf{x}_t, \varepsilon)), y_t \in [0, 1]$. Further $0 \leq \ell''_{\mathrm{KL}} = \sigma(\tilde{f}(\theta, \mathbf{x}_t, \varepsilon))\big(1 - \sigma(\tilde{f}(\theta, \mathbf{x}_t, \varepsilon))\big) \leq 1/2$ a.s. which completes the proof. $\square$

**Lemma 10.** *Under Assumption 5 and $c_p = \sqrt{8\lambda_{Sq}C_H}$, with probability $\left(1 - \frac{2T(L+2)}{m}\right)$ over the randomness of the initialization and $\{\varepsilon_s\}_{s=1}^S$, $\mathcal{L}_{KL}^{(S)}\left(y_t, \{\sigma(\tilde{f}(\theta; \mathbf{x}_t, \varepsilon_s))\}_{s=1}^S\right)$ is $\nu$-strongly convex with respect to $\theta \in B_{\rho,\rho_1}^{\mathrm{Frob}}(\theta_0)$, where $\nu = \mathcal{O}\left(\frac{1}{\sqrt{m}}\right)$.*

*Proof.* In Lemma 5 we showed that $|\tilde{f}(\theta; \mathbf{x}_t, \varepsilon)|$ is bounded with high probability over the randomness of initialization and $\varepsilon$. Specifically from (32)) we have with probability at least $\left(1 - \frac{2T(L+2)+1}{m}\right)$ over the randomness of the initialization and $\varepsilon$

$$
|\tilde{f}(\theta; \mathbf{x}_t, \varepsilon)| \le \sqrt{2(1+\rho_1)^2 \left(\gamma^L + |\phi(0)| \sum_{i=1}^L \gamma^{i-1}\right)^2 + \frac{2}{\sqrt{m}} c_{\mathrm{reg}}^2 (L\rho + \rho_1)^2}
$$
$$
+ \frac{\sqrt{2}c_{\mathrm{reg}}}{m^{1/4}}(L\rho + \rho_1)\sqrt{\ln(mN)}
$$
$$
:= \tilde{f}_{\max}(\theta; \mathbf{x}_t, \varepsilon) \tag{41}
$$

Since $m = \Omega(T^5/\lambda_0^6)$ it follows that $\tilde{f}_{\max}(\theta; \mathbf{x}_t, \varepsilon)$ is $\mathcal{O}(1)$. Now $\forall t \in [T]$, with

$$
q := \sigma(\tilde{f}_{\max}(\theta; \mathbf{x}_t, \varepsilon)) \tag{42}
$$

with probability at least $\left(1 - \frac{2T(L+2)+1}{m}\right)$ we have $\sigma(\tilde{f}(\theta, \mathbf{x}_t, \varepsilon)) \in [q, (1-q)]$.

Next we show that $\mathcal{L}_{KL}$ is strongly convex. Recall that

$$
\mathcal{L}_{KL}^{(S)}\left(y_t, \{\sigma(\tilde{f}(\theta; \mathbf{x}_t, \varepsilon_s))\}_{s=1}^S\right) = \frac{1}{S}\sum_{s=1}^S \ell_{KL}\left(y_t, \sigma(\tilde{f}(\theta; \mathbf{x}_t, \varepsilon_s))\right),
$$

and therefore we have

$$
\nabla_\theta \mathcal{L}_{KL}^{(S)}\left(y_t, \{\sigma(\tilde{f}(\theta; \mathbf{x}_t, \varepsilon_s))\}_{s=1}^S\right) = \frac{1}{S}\sum_{s=1}^S \frac{y_t \sigma'(\tilde{f}(\theta, \mathbf{x}_t, \varepsilon_s))\nabla_\theta \tilde{f}(\theta, \mathbf{x}_t, \varepsilon_s)}{\sigma(\tilde{f}(\theta, \mathbf{x}_t, \varepsilon_s))}
$$
$$
- \frac{(1-y_t)\sigma'(\tilde{f}(\theta, \mathbf{x}_t, \varepsilon_s))\nabla_\theta \tilde{f}(\theta, \mathbf{x}_t, \varepsilon_s)}{1 - \sigma(\tilde{f}(\theta, \mathbf{x}_t, \varepsilon_s))}
$$
$$
= \frac{1}{S}\sum_{s=1}^S \Big(y_t\big(1 - \sigma(\tilde{f}(\theta, \mathbf{x}_t, \varepsilon_s))\big)) - (1-y_t)\sigma(\tilde{f}(\theta, \mathbf{x}_t, \varepsilon_s))\Big)\nabla_\theta \tilde{f}(\theta, \mathbf{x}_t, \varepsilon_s)
$$
$$
= \frac{1}{S}\sum_{s=1}^S \Big(\sigma(\tilde{f}(\theta, \mathbf{x}_t, \varepsilon_s)) - y_t\Big)\nabla_\theta \tilde{f}(\theta, \mathbf{x}_t, \varepsilon_s) \tag{43}
$$

where we have used the fact that $\sigma'(\cdot) = \sigma(\cdot)(1 - \sigma(\cdot))$. Further the hessian of the loss is given by,

$$
\nabla_\theta^2 \mathcal{L}_{KL}^{(S)}\left(y_t, \{\sigma(\tilde{f}(\theta; \mathbf{x}_t, \varepsilon_s))\}_{s=1}^S\right) = \frac{1}{S}\sum_{s=1}^S \sigma'(\tilde{f}(\theta, \mathbf{x}_t, \varepsilon_s))\nabla_\theta \tilde{f}(\theta, \mathbf{x}_t, \varepsilon_s)\nabla_\theta \tilde{f}(\theta, \mathbf{x}_t, \varepsilon_s)^T
$$
$$
+ (\sigma(\tilde{f}(\theta, \mathbf{x}_t, \varepsilon_s)) - y_t)\nabla_\theta^2 \tilde{f}(\theta, \mathbf{x}_t, \varepsilon_s)
$$
$$
\ge \frac{1}{S}\sum_{s=1}^S q(1-q)\nabla_\theta \tilde{f}(\theta, \mathbf{x}_t, \varepsilon_s)\nabla_\theta \tilde{f}(\theta, \mathbf{x}_t, \varepsilon_s)^T - \lambda_{KL}\nabla_\theta^2 f(\theta; \mathbf{x}_t)
$$

Consider $u \in \mathcal{S}^{p-1}$, the unit ball in $\mathbb{R}^p$. We want to show that $u^T \nabla_\theta^2 \mathcal{L}_{KL}^{(S)}\left(y_t, \{\sigma(\tilde{f}(\theta; \mathbf{x}_t, \varepsilon_s))\}_{s=1}^S\right)u > 0$. Now as in the proof of Lemma 6 we can show that with probability $\left(1 - \frac{2T(L+2)}{m}\right)$

$$
u^T \nabla_\theta^2 \mathcal{L}_{KL}^{(S)}\left(y_t, \{\sigma(\tilde{f}(\theta; \mathbf{x}_t, \varepsilon_s))\}_{s=1}^S\right)u \ge q(1-q)\frac{c_p^2}{4\sqrt{m}} - \frac{\lambda_{KL}C_H}{\sqrt{m}}
$$
$$
= \frac{\lambda_{KL}C_H}{\sqrt{m}},
$$

where the last equality follows because $c_p^2 = \frac{8\lambda_{\mathrm{KL}} C_H}{q(1-q)} = \mathcal{O}(1)$. $\qquad\qquad\qquad\qquad\qquad\qquad\qquad\qquad\qquad\qquad$ $\square$

**Lemma 11.** *Under Assumption 5 with probability $\left(1 - \frac{2T(L+1)C}{m}\right)$, for some absolute constant $C > 0$,*
$\mathcal{L}_{KL}^{(S)}\left(y_t, \{\tilde{f}(\theta; \mathbf{x}_t, \boldsymbol{\varepsilon}_s)\}_{s=1}^S\right)$ *satisfies the QG condition over the randomness of the initialization and* $\{\boldsymbol{\varepsilon}_s\}_{s=1}^S$, *with constant $\mu = \mathcal{O}(1)$.*

*Proof.* Recall from (43) that we have,

$$\nabla_\theta \mathcal{L}_{\mathrm{KL}}^{(S)}\left(y_t, \{\sigma(\tilde{f}(\theta; \mathbf{x}_t, \boldsymbol{\varepsilon}_s))\}_{s=1}^S\right) = \frac{1}{S}\sum_{s=1}^S \left(\sigma(\tilde{f}(\theta, \mathbf{x}_t, \boldsymbol{\varepsilon}_s)) - y_t\right)\nabla_\theta \tilde{f}(\theta, \mathbf{x}_t, \boldsymbol{\varepsilon}_s),$$

and therefore,

$$\left\|\nabla_\theta \mathcal{L}_{\mathrm{KL}}^{(S)}\left(y_t, \{\sigma(\tilde{f}(\theta; \mathbf{x}_t, \boldsymbol{\varepsilon}_s))\}_{s=1}^S\right)\right\|_2^2 = \left\|\frac{1}{S}\sum_{s=1}^S \left(\sigma(\tilde{f}(\theta, \mathbf{x}_t, \boldsymbol{\varepsilon}_s)) - y_t\right)\nabla_\theta \tilde{f}(\theta, \mathbf{x}_t, \boldsymbol{\varepsilon}_s)\right\|_2^2$$

$$= \frac{1}{S^2}\sum_{s=1}^S\sum_{s'=1}^S \left(\sigma(\tilde{f}(\theta; \mathbf{x}_t, \boldsymbol{\varepsilon}_s)) - y_t\right)\left(\sigma(\tilde{f}(\theta; \mathbf{x}_t, \boldsymbol{\varepsilon}_s)) - y_t\right)\left\langle\nabla\tilde{f}(\theta; \mathbf{x}_t, \boldsymbol{\varepsilon}_s), \nabla\tilde{f}(\theta; \mathbf{x}_t, \boldsymbol{\varepsilon}_{s'})\right\rangle$$

$$= \frac{1}{S^2}(F(\theta; \mathbf{x}_t) - y_t\mathbb{1}_S)^T \tilde{K}(\theta, \mathbf{x}_t)(F(\theta; \mathbf{x}_t) - y_t\mathbb{1}_S)$$

where $F(\theta, \mathbf{x}_t) : \mathbb{R}^{p\times d} \to \mathbb{R}^S$ such that $(F(\theta, \mathbf{x}_t))_s = \sigma\left(\tilde{f}(\theta; \mathbf{x}_t, \boldsymbol{\varepsilon}_s)\right)$, $\mathbb{1}_S$ is an $S$-dimensional vector of $1's$ and $\tilde{K}(\theta, \mathbf{x}_t) = \left[\langle\nabla_\theta\tilde{f}(\theta; \mathbf{x}_t, \boldsymbol{\varepsilon}_s), \nabla_\theta\tilde{f}(\theta; \mathbf{x}_t, \boldsymbol{\varepsilon}'_s)\rangle\right]$. As in the proof of Lemma 7 with probability $\left(1 - \frac{2T(L+1)C}{m}\right)$ for some absolute constant $C > 0$, we have

$$\frac{1}{S^2}(F(\theta; \mathbf{x}_t) - y_t\mathbb{1}_S)^T \tilde{K}(\theta, \mathbf{x}_t)(F(\theta; \mathbf{x}_t) - y_t\mathbb{1}_S) \geq 2\mu\, \mathcal{L}_{\mathrm{Sq}}^{(S)}\left(y_t, \{\sigma\left(\tilde{f}(\theta; \mathbf{x}_t, \boldsymbol{\varepsilon}_s)\right)\}_{s=1}^S\right), \quad (44)$$

with $\mu = 128$ and

$$\mathcal{L}_{\mathrm{Sq}}^{(S)}\left(y_t, \{\sigma\left(\tilde{f}(\theta; \mathbf{x}_t, \boldsymbol{\varepsilon}_s)\right)\}_{s=1}^S\right) := \frac{1}{S}\sum_{s=1}^S \ell_{\mathrm{Sq}}\left(y_t, \sigma(\tilde{f}(\theta, \mathbf{x}_t, \boldsymbol{\varepsilon}_s))\right).$$

Now using reverse Pinsker's inequality (see eg. Sason (2015), eq (10)) we have

$$D_{\mathrm{KL}}\left(y_t\|\sigma\left(\tilde{f}(\theta, \mathbf{x}_t, \boldsymbol{\varepsilon})\right)\right) \leq \frac{\ell_{\mathrm{Sq}}\left(y_t, \sigma(\tilde{f}(\theta, \mathbf{x}_t, \boldsymbol{\varepsilon}_s))\right)}{2q}$$

where $D_{\mathrm{KL}}(p\|q)$ is the KL divergence between $p$ and $q$ and $\sigma(\tilde{f}(\theta, \mathbf{x}_t, \boldsymbol{\varepsilon})) \in [q, (1-q)]$ holds with probability at least $\left(1 - \frac{2T(L+2)+1}{m}\right)$. Using this we have with probability at least $\left(1 - \frac{TLC}{m}\right)$ for some absolute constant $C > 0$,

$$\mathcal{L}_{\mathrm{KL}}^{(S)}\left(y_t, \{\sigma(\tilde{f}(\theta; \mathbf{x}_t, \boldsymbol{\varepsilon}_s))\}_{s=1}^S\right) - \mathcal{L}_{\mathrm{KL}}^{(S)}\left(y_t, \{\sigma(\tilde{f}(\theta^*; \mathbf{x}_t, \boldsymbol{\varepsilon}_s))\}_{s=1}^S\right)$$

$$= \frac{1}{S}\sum_{s=1}^S D_{\mathrm{KL}}\left(y_t\|\sigma\left(\tilde{f}(\theta, \mathbf{x}_t, \boldsymbol{\varepsilon}_s)\right)\right) - D_{\mathrm{KL}}\left(y_t\|\sigma\left(\tilde{f}(\theta^*, \mathbf{x}_t, \boldsymbol{\varepsilon}_s)\right)\right)$$

$$\leq \frac{1}{S}\sum_{s=1}^S \frac{\ell_{\mathrm{Sq}}\left(y_t, \sigma(\tilde{f}(\theta, \mathbf{x}_t, \boldsymbol{\varepsilon}_s))\right)}{2q} = \frac{1}{2q}\mathcal{L}_{\mathrm{Sq}}^{(S)}\left(y_t, \{\sigma\left(\tilde{f}(\theta; \mathbf{x}_t, \boldsymbol{\varepsilon}_s)\right)\}_{s=1}^S\right).$$

Combining with (44) we get with probability at least $\left(1 - \frac{TLC}{m}\right)$ over the randomness of the initialization and $\{\boldsymbol{\varepsilon}_s\}_{s=1}^S$

$$\left\|\nabla_\theta \mathcal{L}_{\mathrm{KL}}^{(S)}\left(y_t, \{\sigma(\tilde{f}(\theta; \mathbf{x}_t, \boldsymbol{\varepsilon}_s))\}_{s=1}^S\right)\right\|_2^2$$

$$\geq \mu'\left(\mathcal{L}_{\mathrm{KL}}^{(S)}\left(y_t, \{\sigma(\tilde{f}(\theta; \mathbf{x}_t, \boldsymbol{\varepsilon}_s))\}_{s=1}^S\right) - \mathcal{L}_{\mathrm{KL}}^{(S)}\left(y_t, \{\sigma(\tilde{f}(\theta^*; \mathbf{x}_t, \boldsymbol{\varepsilon}_s))\}_{s=1}^S\right)\right)$$

Therefore $\mathcal{L}_{\mathrm{KL}}^{(S)}\left(y_t, \{\sigma(\tilde{f}(\theta; \mathbf{x}_t, \boldsymbol{\varepsilon}_s))\}_{s=1}^S\right)$ satisfies the QG condition with $\mu' = \mathcal{O}(1)$. $\qquad$ $\square$

**Lemma 12.** *Under Assumption 1 and 5 with probability $\left(1 - \frac{TLC}{m}\right)$ for some absolute constant $C > 0$ over the randomness of the initialization we have*

$$\sum_{t=1}^{T} \mathcal{L}_{KL}^{(S)}\left(y_t, \{\sigma(\tilde{f}(\theta^*; \mathbf{x}_t, \boldsymbol{\varepsilon}_s))\}_{s=1}^{S}\right) = \mathcal{O}(1).$$

*Proof.* Recall from the proof of Lemma 11 that using reverse Pinsker's inequality we have with probability at least $\left(1 - \frac{TLC}{m}\right)$ for some absolute constant $C > 0$,

$$\mathcal{L}_{KL}^{(S)}\left(y_t, \{\sigma(\tilde{f}(\theta; \mathbf{x}_t, \boldsymbol{\varepsilon}_s))\}_{s=1}^{S}\right) \leq \frac{1}{2q}\mathcal{L}_{Sq}^{(S)}\left(y_t, \{\sigma(\tilde{f}(\theta; \mathbf{x}_t, \boldsymbol{\varepsilon}_s))\}_{s=1}^{S}\right).$$

Therefore

$$\sum_{t=1}^{T} \mathcal{L}_{KL}^{(S)}\left(y_t, \{\sigma(\tilde{f}(\theta; \mathbf{x}_t, \boldsymbol{\varepsilon}_s))\}_{s=1}^{S}\right) \leq \frac{1}{2q}\sum_{t=1}^{T}\mathcal{L}_{Sq}^{(S)}\left(y_t, \{\sigma(\tilde{f}(\theta; \mathbf{x}_t, \boldsymbol{\varepsilon}_s))\}_{s=1}^{S}\right) = \mathcal{O}(1)$$

where the last part follows from 39 in the proof of Lemma 8.

Finally we can use Theorem E.1 to conclude that there exists $\bar{\bar{\theta}} \in B_{\rho,\rho_1}^{\text{Frob}}(\theta_0)$ such that with probability at least $\left(1 - \frac{2T(L+1)}{m}\right)$ over the randomness of initialization we have $\sigma(f(\bar{\theta}, \mathbf{x}_t)) = y_t$ for any set of $y_t \in [q, 1-q]$, $\forall t \in [T]$ where $q$ is as defined in (42). Without loss of generality assume $z = q$ (otherwise the predictor can be changed to $\hat{y}_t = \sigma(k_z f(\theta, \mathbf{x}_t))$ for some constant $k_z$ that depends on $z$) and therefore we have with probability at least $\left(1 - \frac{TLC}{m}\right)$ for some absolute constant $C > 0$ $\min_{\theta \in B_{\rho,\rho_1}^{\text{Frob}}(\theta_0)} \sum_{t=1}^{T} \ell_{KL}\left(y_t, \sigma(f(\theta; \mathbf{x}_t))\right) = 0$. $\square$

### C.4 AUXILIARY LEMMAS

With slight abuse of notation, we have defined $\mathbf{z}_t = (\mathbf{x}_t, y_t)$ and $\mathcal{L}(\theta; \mathbf{z}_t) = \ell(y_t, f(\theta; \mathbf{x}_t))$.

**Lemma 13 (Almost Convexity of Loss).** *Under the conditions of Lemma 1, with a high probability, $\forall \theta' \in B_{\rho,\rho_1}^{\text{Frob}}(\theta_0)$,*

$$\mathcal{L}(\theta'; \mathbf{z}_t) \geq \mathcal{L}(\theta_t; \mathbf{z}_t) + \langle \theta' - \theta_t, \nabla_\theta \mathcal{L}(\theta_t; \mathbf{z}_t)\rangle + a\langle \theta' - \theta_t, \nabla f(\theta''; \mathbf{z}_t)\rangle^2 - \epsilon_t, \quad (45)$$

$$where \quad \epsilon_t = \frac{4\left(4a\varrho(L\rho + \rho_1) + \lambda\right)c_H(L\rho + \rho_1)^2}{\sqrt{m}},$$

*for any $\theta'' \in B_{\rho,\rho_1}^{\text{Frob}}(\theta_0)$, where $c_H$ and $\varrho$ are as in Theorem 1 and $\lambda, a$ are as in Assumption 3 (lipschitz, strongly convex, smooth).*

*Proof.* Consider any $\theta' \in B_{\rho,\rho_1}^{\text{Frob}}(\theta_0)$. By the second order Taylor expansion around $\theta_t$, we have

$$\mathcal{L}(\theta'; \mathbf{z}_t) = \mathcal{L}(\theta_t; \mathbf{z}_t) + \langle \theta' - \theta_t, \nabla_\theta \mathcal{L}(\theta_t; \mathbf{z}_t)\rangle + \frac{1}{2}(\theta' - \theta_t)^\top \frac{\partial^2 \mathcal{L}(\tilde{\theta}_t; \mathbf{z}_t)}{\partial \theta^2}(\theta' - \theta_t),$$

where $\tilde{\theta}_t = \xi\theta' + (1-\xi)\theta_t$ for some $\xi \in [0, 1]$. Since $\theta_t, \theta' \in B_{\rho,\rho_1}^{\text{Frob}}(\theta_0)$, we have

$$\begin{aligned}
\left\|\text{vec}(\tilde{W}_t^{(l)}) - \text{vec}(W_0^{(l)})\right\|_2 &= \left\|\xi\text{vec}(W^{'(l)}) + (1-\xi)\text{vec}(W_t^{(l)}) - \text{vec}(W_0^{(l)})\right\|_2 \\
&\leq \xi\left\|\text{vec}(W^{'(l)}) - \text{vec}(W_0^{(l)})\right\|_2 + (1-\xi)\left\|\text{vec}(W_t^{(l)}) - \text{vec}(W_0^{(l)})\right\|_2 \\
&\leq \xi\rho + (1-\xi)\rho = \rho,
\end{aligned}$$

$$\begin{aligned}
\|\tilde{\mathbf{v}}_t - \mathbf{v}_0\|_2 &\leq \|\xi\mathbf{v}' + (1-\xi)\mathbf{v}_t - \mathbf{v}_0\|_2 \leq \xi\|v' - \mathbf{v}_0\|_2 + (1-\xi)\|\mathbf{v}_t - \mathbf{v}_0\|_2 \\
&\leq \xi\rho_1 + (1-\xi)\rho_1 = \rho_1
\end{aligned}$$

and therefore $\tilde{\theta}_t \in B_{\rho,\rho_1}^{\text{Frob}}(\theta_0)$ which implies $\tilde{\theta}_t \in B_{L\rho+\rho_1}^{\text{Euc}}(\theta_0)$. Focusing on the quadratic form in the Taylor expansion, we have

$$(\theta' - \theta_t)^\top \frac{\partial^2 \mathcal{L}(\tilde{\theta}_t; \mathbf{z}_t)}{\partial \theta^2}(\theta' - \theta_t) = (\theta' - \theta_t)^\top \frac{1}{n_t}\sum_{i=1}^{n_t}\left[\tilde{\ell}_{t,i}'' \frac{\partial f(\tilde{\theta}_t; \mathbf{x}_t)}{\partial \theta}\frac{\partial f(\tilde{\theta}_t; \mathbf{x}_t)}{\partial \theta}^\top + \tilde{\ell}_{t,i}'\frac{\partial^2 f(\tilde{\theta}_t; \mathbf{x}_t)}{\partial \theta^2}\right](\theta' - \theta_t)$$

$$= \frac{1}{n_t}\sum_{i=1}^{n_t}\underbrace{\tilde{\ell}_{t,i}''\left\langle \theta' - \theta_t, \frac{\partial f(\tilde{\theta}_t; \mathbf{x}_t)}{\partial \theta}\right\rangle^2}_{I_1} + \underbrace{\tilde{\ell}_{t,i}'(\theta' - \theta_t)^\top\frac{\partial^2 f(\tilde{\theta}_t; \mathbf{x}_t)}{\partial \theta^2}(\theta' - \theta_t)}_{I_2},$$

$\tilde{\ell}_{t,i}$ corresponds to the loss $\ell(y_{t,i}, f(\tilde{\theta}; \mathbf{x}_t))$, and $\tilde{\ell}'_{t,i}, \tilde{\ell}''_{t,i}$ denote the corresponding first and second derivatives w.r.t. $\tilde{y}_{t,i} = f(\tilde{\theta}; \mathbf{x}_t)$. For analyzing $I_1$, choose any $\theta'' \in B^{\text{Frob}}_{\rho, \rho_1}(\theta_0)$. Now, note that

$$
I_1 = \ell''_{t,i} \left\langle \theta' - \theta_t, \frac{\partial f(\tilde{\theta}_t; \mathbf{x}_t)}{\partial \theta} \right\rangle^2
$$

$$
\geq a \left\langle \theta' - \theta_t, \frac{\partial f(\theta''; \mathbf{x}_t)}{\partial \theta} + \left( \frac{\partial f(\tilde{\theta}_t; \mathbf{x}_t)}{\partial \theta} - \frac{\partial f(\theta''; \mathbf{x}_t)}{\partial \theta} \right) \right\rangle^2
$$

$$
= a \left\langle \theta' - \theta_t, \frac{\partial f(\theta''; \mathbf{x}_t)}{\partial \theta} \right\rangle^2 + a \left\langle \theta' - \theta_t, \frac{\partial f(\tilde{\theta}_t; \mathbf{x}_t)}{\partial \theta} - \frac{\partial f(\theta''; \mathbf{x}_t)}{\partial \theta} \right\rangle^2
$$

$$
+ 2a \left\langle \theta' - \theta_t, \frac{\partial f(\theta''; \mathbf{x}_t)}{\partial \theta} \right\rangle \left\langle \theta' - \theta_t, \frac{\partial f(\tilde{\theta}_t; \mathbf{x}_t)}{\partial \theta} - \frac{\partial f(\theta''; \mathbf{x}_t)}{\partial \theta} \right\rangle
$$

$$
\geq a \left\langle \theta' - \theta_t, \frac{\partial f(\theta''; \mathbf{x}_t)}{\partial \theta} \right\rangle^2 - 2a \left\| \frac{\partial f(\theta''; \mathbf{x}_t)}{\partial \theta} \right\|_2 \left\| \frac{\partial f(\tilde{\theta}_t; \mathbf{x}_t)}{\partial \theta} - \frac{\partial f(\theta''; \mathbf{x}_t)}{\partial \theta} \right\|_2 \| \theta' - \theta_t \|_2^2
$$

$$
\overset{(a)}{\geq} a \langle \theta' - \theta_t, \nabla f(\theta''; \mathbf{x}_t) \rangle^2 - 2a\varrho \frac{c_H}{\sqrt{m}} \| \tilde{\theta}_t - \theta'' \|_2 \| \theta' - \theta_t \|_2^2
$$

$$
\overset{(b)}{\geq} a \langle \theta' - \theta_t, \nabla f(\theta''; \mathbf{x}_t) \rangle^2 - \frac{16 a \varrho c_H (L\rho + \rho_1)^3}{\sqrt{m}} \ ,
$$

where (a) follows from Proposition 1 since $\tilde{\theta}_t \in B_\rho(\theta_0)$ and since $\| \nabla f(\theta''; \mathbf{x}_i) \|_2 \leq \varrho$, and (b) follows since $\| \tilde{\theta}_t - \theta'' \|_2, \| \theta' - \theta_t \|_2, \| \tilde{\theta}_t - \theta'' \|_2 \leq 2(L\rho + \rho_1)$ by triangle inequality because $\tilde{\theta}, \theta', \theta_t \in B^{\text{Euc}}_{L\rho + \rho_1}(\theta_0)$.

For analyzing $I_2$, with $Q_{t,i} = (\theta' - \theta_t)^\top \frac{\partial^2 f(\tilde{\theta}_t; \mathbf{x}_i)}{\partial \theta^2} (\theta' - \theta_t)$, we have

$$
|Q_{t,i}| = \left| (\theta' - \theta_t)^\top \frac{\partial^2 f(\tilde{\theta}_t; \mathbf{x}_i)}{\partial \theta^2} (\theta' - \theta_t) \right| \leq \| \theta' - \theta_t \|_2^2 \left\| \frac{\partial^2 f(\tilde{\theta}_t; \mathbf{x}_i)}{\partial \theta^2} \right\|_2 \leq \frac{c_H \| \theta' - \theta_t \|_2^2}{\sqrt{m}} \leq \frac{4 c_H (L\rho + \rho_1)^2}{\sqrt{m}} \ ,
$$

since $\| \theta' - \theta_t \|_2 \leq 2(L\rho + \rho_1)$ by triangle inequality because $\theta', \theta_t \in B^{\text{Euc}}_{L\rho + \rho_1}(\theta_0)$. Further, since $|\tilde{\ell}'_i| \leq \lambda$ by Assumption 3 (lipschitz, strongly convex, smooth), we have

$$
I_2 = \tilde{\ell}'_i Q_{t,i} \geq -|\tilde{\ell}'_i||Q_{t,i}| \geq -\frac{4(L\rho + \rho_1)^2 c_H \lambda}{\sqrt{m}} \ .
$$

Putting the lower bounds on $I_1$ and $I_2$ back, we have

$$
(\theta' - \theta_t)^\top \frac{\partial^2 \mathcal{L}(\tilde{\theta}_t)}{\partial \theta^2} (\theta' - \theta_t) \geq a \langle \theta' - \theta_t, \nabla f(\theta''; \mathbf{x}_t) \rangle^2 - \frac{4\big(4 a \varrho (L\rho + \rho_1) + \lambda\big) c_H (L\rho + \rho_1)^2}{\sqrt{m}} \ .
$$

That completes the proof. $\qquad \square$

**Lemma 14.** *Under Assumption 5 and $c_{reg} = \sqrt{8 \lambda_{Sq} C_H}$, with probability $\left( 1 - \frac{2T(L+1)}{m} \right)$ over the randomness of the initialization, the expected loss $\mathbb{E}_{\boldsymbol{\varepsilon}} \ell_{Sq}\big(y_t, \tilde{f}(\theta, \mathbf{x}_t, \boldsymbol{\varepsilon})\big)$ is $\nu$-strongly convex with respect to $\theta \in B^{\text{Frob}}_{\rho, \rho_1}(\theta_0)$, where $\nu = \mathcal{O}\left( \frac{1}{\sqrt{m}} \right).$*

*Proof.* From (10) we have a.s.

$$
\nabla_\theta \tilde{f}(\theta, \mathbf{x}_t, \boldsymbol{\varepsilon}) = \nabla_\theta f(\theta; \mathbf{x}) + c_{\text{reg}} \sum_{i=1}^p \frac{e_i \varepsilon_i}{m^{1/4}} \ , \tag{46}
$$

$$
\nabla^2_\theta \tilde{f}(\theta, \mathbf{x}_t, \boldsymbol{\varepsilon}) = \nabla^2_\theta f(\theta; \mathbf{x}). \tag{47}
$$

Next, with $\ell_t' = (\tilde{f}(\theta, \mathbf{x}_t, \varepsilon) - y_t)$

$$\nabla_\theta \ell_{\text{Sq}}\big(y_t, \tilde{f}(\theta, \mathbf{x}_t, \varepsilon)\big) = \frac{1}{n_t} \sum_{i=1}^{n_t} \ell_t' \nabla_\theta \tilde{f}(\theta, \mathbf{x}_t, \varepsilon)$$

$$\nabla_\theta^2 \ell_{\text{Sq}}\big(y_t, \tilde{f}(\theta, \mathbf{x}_t, \varepsilon)\big) = \frac{1}{n_t} \sum_{i=1}^{n_t} \nabla_\theta \tilde{f}(\theta, \mathbf{x}_t, \varepsilon) \nabla_\theta \tilde{f}(\theta, \mathbf{x}_t, \varepsilon)^T + \ell_t' \nabla_\theta^2 \tilde{f}(\theta, \mathbf{x}_t, \varepsilon)$$

where we have used the fact that $\ell_t'' = 1$, and $\nabla_\theta^2 \tilde{f}(\theta, \mathbf{x}_t, \varepsilon) = \nabla_\theta^2 f(\theta; \mathbf{x}_t)$ from (47). Taking expectation with respect to $\varepsilon_i$ and using (46) we get

$$\nabla_\theta^2 \mathbb{E}_\varepsilon \ell_{\text{Sq}}\big(y_t, \tilde{f}(\theta, \mathbf{x}_t, \varepsilon)\big) = \frac{1}{n_t} \sum_{i=1}^{n_t} \mathbb{E}_\varepsilon \nabla_\theta \tilde{f}(\theta, \mathbf{x}_t, \varepsilon) \nabla_\theta \tilde{f}(\theta, \mathbf{x}_t, \varepsilon)^T + \ell_{t,i}' \nabla_\theta^2 f(\theta; \mathbf{x}_t)$$

$$= \frac{1}{n_t} \sum_{i=1}^{n_t} \mathbb{E}_\varepsilon \left( \nabla_\theta f(\theta; \mathbf{x}_t) + c_{\text{reg}} \sum_{i=1}^{p} \frac{e_i}{m^{1/4}} \varepsilon_i \right) \left( \nabla_\theta f(\theta; \mathbf{x}_t) + c_{\text{reg}} \sum_{i=1}^{p} \frac{e_i}{m^{1/4}} \varepsilon_i \right)^T + \ell_{t,i}' \nabla_\theta^2 f(\theta; \mathbf{x}_t)$$

$$= \frac{1}{n_t} \sum_{i=1}^{n_t} \mathbb{E}_\varepsilon \left[ \nabla_\theta f(\theta; \mathbf{x}_t) \nabla_\theta f(\theta; \mathbf{x}_t)^T + c_{\text{reg}} \sum_{i=1}^{p} \frac{e_i \nabla_\theta f(\theta; \mathbf{x}_t)^T}{m^{1/4}} \varepsilon_i \right.$$

$$\left. + c_{\text{reg}} \sum_{i=1}^{p} \frac{\nabla_\theta f(\theta; \mathbf{x}_t) e_i^T}{m^{1/4}} \varepsilon_i + c_{\text{reg}}^2 \sum_{i=1}^{p} \sum_{j=1}^{p} \frac{e_i e_j^T}{\sqrt{m}} \varepsilon_i \varepsilon_j \right] + \ell_{t,i}' \nabla_\theta^2 f(\theta; \mathbf{x}_t)$$

$$= \frac{1}{n_t} \sum_{i=1}^{n_t} \nabla_\theta f(\theta; \mathbf{x}_t) \nabla_\theta f(\theta; \mathbf{x}_t)^T + \frac{c_{\text{reg}}^2}{\sqrt{m}} I + \ell_{t,i}' \nabla_\theta^2 f(\theta; \mathbf{x}_t)$$

where the last equality follows from the fact that $\mathbb{E}[\varepsilon_i] = 0$ and $\mathbb{E}[\varepsilon_i \varepsilon_j] = 0$ for $i \neq j$ and $\mathbb{E}[\varepsilon_i \varepsilon_j] = 1$ for $i = j$ and therefore $\sum_{i=1}^{p} e_i e_i^T = I$, the identity matrix. Now consider any $u \in \mathbb{R}^p, \|u\| = 1$. We want to show that $u^T \nabla_\theta^2 \mathbb{E}_\varepsilon \ell_{\text{Sq}}\big(y_t, \tilde{f}(\theta, \mathbf{x}_t, \varepsilon)\big) u > 0$.

$$u^T \nabla_\theta^2 \mathbb{E}_\varepsilon \ell_{\text{Sq}}\big(y_t, \tilde{f}(\theta, \mathbf{x}_t, \varepsilon)\big) u = \frac{1}{n_t} \sum_{i=1}^{n_t} \langle u, \nabla_\theta f(\theta; \mathbf{x}_t) \rangle^2 + \frac{c_{\text{reg}}^2}{\sqrt{m}} \|u\|^2 + \ell_{t,i}' u^T \nabla_\theta^2 f(\theta; \mathbf{x}_t) u$$

$$\overset{(a)}{\geq} \frac{c_{\text{reg}}^2}{\sqrt{m}} - \frac{\lambda_{\text{Sq}} C_H}{\sqrt{m}}$$

$$\overset{(b)}{=} \frac{\lambda_{\text{Sq}} C_H}{\sqrt{m}}$$

where $(a)$ uses the fact that $|\ell_t'| \leq \lambda_{\text{Sq}}$ and that $\|\nabla_\theta^2 f(\theta; \mathbf{x}_t)\|_2 \leq \frac{C_H}{\sqrt{m}}$ holds with probability $\left(1 - \frac{2(L+1)}{m}\right)$ from Proposition 1. Further $(b)$ uses the fact that $c_{\text{reg}}^2 = 2\lambda_{\text{Sq}} C_H$. Finally with a union bound over $i_t \in [n_t], t \in [T]$ and noting that $u$ was arbitrary we conclude that with probability $\left(1 - \frac{2T(L+1)}{m}\right)$ over the randomness of initialization $\mathbb{E}_\varepsilon \ell_{\text{Sq}}\big(y_t, \tilde{f}(\theta, \mathbf{x}_t, \varepsilon)\big)$ is $\nu$-strongly convex with $\nu = \frac{\lambda_{\text{Sq}} C_H}{\sqrt{m}}$. $\qquad\square$

# D  PROOF OF RESULTS FOR CONTEXTUAL BANDITS (SECTION 4)

**Theorem 4.1** (**Regret bound for** NeuSquareCB). *Under Assumption 6 and 5 with appropriate choice of the parameter $\gamma$, step-size sequence $\{\eta_t\}$ width $m$, and regularization parameter $c_p$, with high probability over the randomness in the initialization and $\{\varepsilon\}_{s=1}^S$ the regret for* NeuSquareCB *with $\rho = \Theta(\sqrt{T}/\lambda_0), \rho_1 = \Theta(1)$ is given by* $\text{Reg}_{\text{CB}}(T) \leq \tilde{\mathcal{O}}(\sqrt{KT})$.

*Proof.* Choosing $\delta = \frac{1}{T}$ and $\gamma = \sqrt{KT/(\text{R}_{\text{Sq}}(T)) + \log(2T)}$ in Theorem 1 of Foster and Rakhlin (2020) we get:

$$\mathbb{E}\big[\text{Reg}_{\text{CB}}(T)\big] \leq 4\sqrt{KT\text{R}_{\text{Sq}}(T)} + 8\sqrt{KT\ln(2T)} + 1$$

Using Theorem 4.1 we have with probability at least $\left(1 - \frac{2LC}{m}\right)$ for some absolute constant $C > 0$,

$$\mathbb{E}\big[\text{Reg}_{\text{CB}}(T)\big] \leq \tilde{\mathcal{O}}(\sqrt{KT\log T}) + 8\sqrt{KT\ln(2T)} + 1$$
$$\leq \tilde{\mathcal{O}}(\sqrt{KT})$$

$\square$

which completes the proof.

**Theorem 4.2** (**Regret bound for** NeuFastCB). *Under Assumption 6 and 5 with appropriate choice of the parameter $\gamma$, step-size sequence $\{\eta_t\}$ width $m$, and regularization parameter $c_p$, with high probability over the randomness in the initialization and $\{\varepsilon\}_{s=1}^S$, the regret for* NeuFastCB *with $\rho = \Theta(\sqrt{T}/\lambda_0), \rho_1 = \Theta(1)$ is given by* $\text{Reg}_{\text{CB}}(T) \leq \tilde{\mathcal{O}}(\sqrt{L^*K} + K)$, where $L^* = \sum_{t=1}^T y_{t,a_t^*}$.

*Proof.* Choosing $\gamma = \max(\sqrt{KL^*/3\text{R}_{\text{KL}}(T)}, 10K)$ and using Theorem 1 from Foster and Krishnamurthy (2021) we get

$$\mathbb{E}\big[\text{Reg}_{\text{CB}}(T)\big] = 40\sqrt{L^*K\text{R}_{\text{KL}}(T)} + 600K\ \text{R}_{\text{KL}}(T)$$

Using Theorem 4.2 we have with probability at least $\left(1 - \frac{2LC}{m}\right)$ for some absolute constant $C > 0$,

$$\mathbb{E}\big[\text{Reg}_{\text{CB}}(T)\big] \leq \mathcal{O}(\sqrt{L^*K\log T}) + 600K\mathcal{O}(\log T)$$
$$\leq \tilde{\mathcal{O}}(\sqrt{KL^*} + K)$$

which completes the proof.

$\square$

**Remark D.1.** A keen reader might notice that the reduction in (Foster and Rakhlin, 2020) requires us to control $\text{R}_{\text{Sq}}(T) = \sum_{t=1}^T \ell_{\text{Sq}}(y_t, \hat{y}_t) - \sum_{t=1}^T \ell_{\text{Sq}}(y_t, h(\mathbf{x}_t))$, but our regret guarantee is for $\tilde{\text{R}}_{\text{Sq}}(T)$ in (12). However, note that in step-4 of the proof of Theorem 3.2 we argued that $\sum_{t=1}^T \mathcal{L}_{\text{Sq}}^{(S)}\left(y_t, \{\tilde{f}(\tilde{\theta}^*; \mathbf{x}_t, \varepsilon_s)\}_{s=1}^S\right) = \mathcal{O}(1)$ and therefore our regret bound implies $\sum_{t=1}^T \ell_{\text{Sq}}(y_t, \hat{y}_t) \leq \mathcal{O}(\log T)$ with $\hat{y}_T = \tilde{f}^{(S)}\left(\theta_t; \mathbf{x}_t, \varepsilon^{(1:S)}\right)$ which immediately implies $\text{Reg}_{\text{Sq}}(T) \leq \mathcal{O}(\log T)$. A similar argument follows for Foster and Krishnamurthy (2021).

# E    INTERPOLATION WITH WIDE NETWORKS (PROOF OF THEOREM E.1)

In this section, we focus on showing that under suitable assumptions, wide networks can interpolate any given data. We assume $\ell$ to be the squared loss throughout this subsection.

**Theorem E.1.** *Under Assumptions 4 and 5, for any $h : \mathcal{X} \mapsto [0, 1]$ and any set of inputs $\mathbf{x}_t \in \mathcal{X}, t \in [T]$, for $f(\theta; \mathbf{x})$ of the form equation 1, if the width $m = \Omega(T^4)$, there exists $\tilde{\theta} \in B_{\rho,\rho_1}^{\mathrm{Frob}}(\theta_0)$ with $\rho = \Theta(\frac{\sqrt{T}}{\lambda_0})$ and $\rho_1 = \Theta(1)$, such that with probability at least $(1 - \frac{2(L+1)}{m})$ we have $f(\tilde{\theta}, \mathbf{x}_t) = h(\mathbf{x}_t), \forall t \in [T]$; further, there exists $\bar{\theta} \in B_{\rho,\rho_1}^{\mathrm{Frob}}(\theta_0)$ such that with probability at least $(1 - \frac{2(L+1)}{m})$ we have $f(\bar{\theta}, \mathbf{x}_t) = y_t$, for any set of $y_t \in [0, 1], t \in [T]$.*

We start with an outline of the overall proof, which has four technical steps, some of which follow from direct observations, assumptions, or existing results (especially on the NTK), and some require new proofs.

1.  It is sufficient to prove the interpolation result showing $f(\bar{\theta}, \mathbf{x}_t) = y_t$ for any $y_t$ as the result for the interpolation $f(\bar{\theta}, \mathbf{x}_t) = h(\mathbf{x}_t)$ follow as a special case with $y_t = h(\mathbf{x}_t)$. Further we consider $\rho_1 = 0$ which immediately implies the result for $\rho_1 = \Theta(1)$.

2.  The interpolation analysis will utilize the fact that the NTK is positive definite at initialization. For simplicity, Assumption 5 (positive definite NTK) takes care of this aspect for Theorem E.1, with $\lambda_0 > 0$ being the lower bound to minimum eigen-value of the NTK.

3.  To show existence of $\bar{\theta}$ which interpolates $f(\bar{\theta}, \mathbf{x}_t) = y_t, t \in [T]$, we in fact show that gradient descent on least squares loss with suitably small step size $\eta$ and suitably large width $m$ will have geometrically decreasing cumulative square loss. The decreasing loss along with the fact that the sequence of iterates $\theta_t \in B_{\rho,\rho_1}^{\mathrm{Frob}}(\theta_0)$ with $\rho = \Theta(\frac{\sqrt{T}}{\lambda_0}), \rho_1 = 0$, i.e., stay within the closed ball, implies existence of $\bar{\theta} \in B_{\rho,\rho_1}^{\mathrm{Frob}}(\theta_0)$ which interpolates the data.

4.  Two key properties need to be maintained as the gradient descent iterations proceed: first, as discussed above, the iterates $\theta_t \in B_{\rho,\rho_1}^{\mathrm{Frob}}(\theta_0)$ with $\rho = \Theta(\frac{\sqrt{T}}{\lambda_0}), \rho_1 = 0$, i.e., the iterates stay within the ball; and second, the NTK corresponding to all $\theta_t$ need to stay positive definite. As we will show, these two properties are coupled, and the geometric decrease of the loss helps in the analysis of both properties.

We define

$$\hat{\ell}(\theta) := \sum_{t=1}^{T} \ell(y_t, f(\theta; \mathbf{x}_t)) , \tag{48}$$

**Lemma 1** (NTK condition per step). *Under Assumptions 5 and 4, for the gradient descent update $\theta_{t+1} = \theta_t - \eta_t \nabla L(\theta_t)$ for the cumulative square loss $\hat{\ell}(\theta) = \sum_{n=1}^{T} (y_n - f(\theta; \mathbf{x}_n))^2$ with $\theta_t, \theta_{t+1} \in B_{\rho,\rho_1}^{\mathrm{Frob}}(\theta_0)$ with $\rho = \Theta(\frac{\sqrt{T}}{\lambda_0}), \rho_1 = 0$, with probability at least $\left(1 - \frac{2(L+1)}{m}\right)$ over the initialization of model,*

$$\lambda_{\min}(K_{\mathrm{NTK}}(\theta_{t+1})) \geq \lambda_{\min}(K_{\mathrm{NTK}}(\theta_t)) - 4c_H \varrho^2 \frac{T}{\sqrt{m}} \eta_t \sqrt{\hat{\ell}(\theta_t)} , \tag{49}$$

*where $c_H$ and $\varrho$ are as in Lemma 1.*

*Proof.* Observe that $K_{\mathrm{NTK}}(\theta) = J(\theta) J(\theta)^\top$, where the Jacobian

$$J(\theta) = \begin{bmatrix} \left(\frac{\partial f(\theta; x_1)}{\partial W^{(1)}}\right)^\top & \cdots & \left(\frac{\partial f(\theta; x_1)}{\partial W^{(L)}}\right)^\top \\ \vdots & \ddots & \vdots \\ \left(\frac{\partial f(\theta; x_n)}{\partial W^{(1)}}\right)^\top & \cdots & \left(\frac{\partial f(\theta; x_n)}{\partial W^{(L)}}\right)^\top \end{bmatrix} \in \mathbb{R}^{T \times (md + Lm^2)} , \tag{50}$$

where the parameter matrices are vectorized and the last layer $W^{(L+1)} = \mathbf{v}_0$ is ignored since we will not be doing gradient descent on the last layer and its kept fixed. Then, the spectral norm of the change in the NTK is given by

$$
\begin{aligned}
\|K_{\mathrm{NTK}}(\theta_{t+1}) - K_{\mathrm{NTK}}(\theta_t)\|_2 &= \left\| J(\theta_{t+1})J(\theta_{t+1})^\top - J(\theta_t)J(\theta_t)^\top \right\|_2 \\
&= \left\| J(\theta_{t+1})(J(\theta_{t+1}) - J(\theta_t))^\top - (J(\theta_{t+1}) - J(\theta_t))J(\theta_t)^\top \right\|_2 \\
&\le (\|J(\theta_{t+1})\|_2 + \|J(\theta_t)\|_2)\,\|J(\theta_{t+1}) - J(\theta_t)\|_2 \ .
\end{aligned}
\tag{51}
$$

Now, for any $\theta \in B^{\mathrm{Frob}}_{\rho,\rho_1}(\theta_0)$,

$$
\|J(\theta)\|_2^2 \le \|J(\theta)\|_F^2 = \sum_{n=1}^{N} \left\| \frac{\partial f(\theta; x_n)}{\partial \theta} \right\|_2^2 \overset{(a)}{\le} T\varrho^2
$$

where (a) follows from by Lemma 1. Assuming $\theta_t, \theta_{t+1} \in B^{\mathrm{Frob}}_{\rho,\rho_1}(\theta_0)$, we have $\|J(\theta_t)\|_2, \|J(\theta_{t+1})\|_2 \le \sqrt{T}\varrho$, so that from equation 51 we get

$$
\|K_{\mathrm{NTK}}(\theta_{t+1}) - K_{\mathrm{NTK}}(\theta_t)\|_2 \le 2\sqrt{T}\varrho\,\|J(\theta_{t+1}) - J(\theta_t)\|_2 \ .
\tag{52}
$$

Now, note that

$$
\|J(\theta_{t+1}) - J(\theta_t)\|_2 \le \|J(\theta_{t+1}) - J(\theta_t)\|_F
\tag{53}
$$

$$
\le \sqrt{\sum_{n=1}^{T} \left\| \frac{\partial f(\theta_{t+1}; x_n)}{\partial \theta} - \frac{\partial f(\theta_t; x_n)}{\partial \theta} \right\|_2^2}
\tag{54}
$$

$$
\overset{(a)}{\le} \sqrt{T}\sup_{\tilde{\theta}_t, i} \left\| \frac{\partial^2 f(\tilde{\theta}_t; x_i)}{\partial \theta^2} \right\|_2 \|\theta_{t+1} - \theta_t\|_2
$$

$$
\overset{(b)}{\le} \frac{c_H\sqrt{T}}{\sqrt{m}} \|\theta_{t+1} - \theta_t\|_2
\tag{55}
$$

$$
\overset{(c)}{=} \frac{c_H\sqrt{T}}{\sqrt{m}} \eta_t \left\| \nabla\hat{\ell}(\theta_t) \right\|_2
$$

$$
\overset{(d)}{\le} \frac{2c_H\sqrt{T}\varrho}{\sqrt{m}} \eta_t \sqrt{\hat{\ell}(\theta_t)} \ ,
$$

where (a) follows from the mean-value theorem with $\tilde{\theta}_t \in \{(1-\xi)\theta_t + \xi\theta_{t+1}$ for some $\xi \in [0,1]\}$, (b) follows from Lemma 1 since $\tilde{\theta} \in B^{\mathrm{Frob}}_{\rho,\rho_1}(\theta_0)$, (c) follows from the gradient descent update, and (d) follows from Lemma 3. Then, using equation 52, we have

$$
\|K_{\mathrm{NTK}}(\theta_{t+1}) - K_{\mathrm{NTK}}(\theta_t)\|_2 \le 4c_H\varrho^2\frac{T}{\sqrt{m}}\eta_t\sqrt{\hat{\ell}(\theta_t)} \ .
\tag{56}
$$

Then, by triangle inequality

$$
\begin{aligned}
\lambda_{\min}(K_{\mathrm{NTK}}(\theta_{t+1})) &\ge \lambda_{\min}(K_{\mathrm{NTK}}(\theta_t)) - \|K_{\mathrm{NTK}}(\theta_{t+1}) - K_{\mathrm{NTK}}(\theta_t)\|_2 \\
&\overset{(a)}{\ge} \lambda_{\min}(K_{\mathrm{NTK}}(\theta_t)) - 4c_H\varrho^2\frac{T}{\sqrt{m}}\eta_t\sqrt{\hat{\ell}(\theta_t)} \ ,
\end{aligned}
$$

where (a) follows from equation 56. That completes the proof. $\square$

**Theorem E.2** (**Geometric convergence: Unknown Desired Loss**). *Under Assumptions 4 and 5, consider the gradient descent update* $\theta_{t+1} = \theta_t - \eta_t\nabla\hat{\ell}(\theta_t)$ *for the cumulative loss* $\hat{\ell}(\theta) = \sum_{n=1}^{T}(y_i - f(\theta; \mathbf{x}_i))^2$ *with step size*

$$
\eta_t = \eta < \min\left( \frac{1}{\beta N}, \frac{1}{\lambda_0} \right) \ ,
$$

with $\beta$ as in Lemma 4. Then, choosing width $m = \Omega\left(\frac{T^3}{\lambda_0^4}\right)$ and depth $L = O(1)$, with probability at least $\left(1 - \frac{2(L+1)}{m}\right)$, we have $\{\theta_t\}_t \subset B_{\rho,\rho_1}^{\mathrm{Frob}}(\theta_0)$ with $\rho = \Theta(\frac{\sqrt{T}}{\lambda_0})$ $\rho_1 = 0$, and for every $t$,

$$\hat{\ell}(\theta_{t+1}) \leq (1 - \eta\lambda_0)^t \hat{\ell}(\theta_0) . \tag{57}$$

*Proof.* First note that with probability at least $\left(1 - \frac{2(L+1)}{m}\right)$, all the bounds in Theorem 1, Lemma 2, and Corollary 3 hold, and so does Lemma 1, since its proof uses these bounds.

Further, we note that for $L = O(1)$ we obtain the constant $c_H = O((1+\rho_1)(1+(4\nu_0 + \frac{\rho}{\sqrt{m}})^{O(1)})) = O(1)$ following Lemma 1 since $\rho_1 = 0$, where the last equality follows from the fact that $m = \Omega(\frac{T^4}{\lambda_0^2}) > \Omega(\frac{T}{\lambda_0^2})$ and $\rho = \Theta(\frac{\sqrt{T}}{\lambda_0})$. We will use $c_H \leq c_2$ for some suitable constant $c_2 > 0$.

We also note that $\varrho^2 = O((1 + \frac{1}{m}(1+\rho_1)^2(1+\gamma^{2(L+1)}))$. Then, using the fact that, $L = O(1)$, $\rho = \Theta(\frac{\sqrt{T}}{\lambda_0})$, $\rho_1 = 0$ and $m = \Omega(\frac{T^4}{\lambda_0^2}) > \Omega(\frac{T}{\lambda_0^2})$, we obtain that $\varrho^2 = O(1)$. We will use $\varrho^2 \leq c_3$ for some suitable constant $c_3 > 0$.

Finally, we observe that $\bar{c}_{\rho_1,\gamma} = O((1 + \rho_1^2)(1 + \gamma^L))$ (see definition in Lemma 2). Then, taking the definition of $\beta$ (as in Lemma 4), we have that $\beta = b\varrho^2 + \frac{1}{\sqrt{m}}O(\mathrm{poly}(L)(1 + \gamma^{3L})(1 + \rho_1^2))$. Again, in a similar fashion as in the analysis of the expressions $c_H$ and $\varrho$, we have that in our problem setting $\beta = O(1)$ since $\rho_1 = 0$. We will use $\beta \leq c_4$ for some suitable constant $c_4 > 0$.

We now proceed with the proof by induction. First, for $t = 1$, we show that, based on the choice of the step size, $\theta_1 \in B_{\rho,\rho_1}^{\mathrm{Frob}}(\theta_0)$ for $\rho_1 = 0$. To see this, note that

$$\|\theta_1 - \theta_0\|_2 = \eta\|\nabla\hat{\ell}(\theta_0)\|_2 \overset{(a)}{\leq} 2\varrho\eta\sqrt{\hat{\ell}(\theta_0)}$$
$$\overset{(b)}{\leq} 2\varrho\eta\sqrt{T\bar{c}_{0,4\nu_0}} = 2\varrho\eta\lambda_0\frac{\sqrt{T\bar{c}_{0,4\nu_0}}}{\lambda_0}$$
$$\leq 2\varrho\sqrt{\bar{c}_{0,4\nu_0}}\frac{\sqrt{T}}{\lambda_0} \overset{(c)}{\leq} \rho ,$$

where (a) follows from Lemma 3, (b) from Lemma 2, (c) follows since $\rho = \Theta(\frac{\sqrt{T}}{\lambda_0})$ and $\rho_1 = 0$ so the last layer is not getting updated. Hence, $\theta_1 \in B_{\rho,\rho_1}^{\mathrm{Frob}}(\theta_0)$. We now take the smoothness property from Lemma 4, and further obtain

$$\hat{\ell}(\theta_1) - \hat{\ell}(\theta_0) \leq \langle\theta_1 - \theta_0, \nabla_\theta\hat{\ell}(\theta_0)\rangle + \frac{\beta T}{2}\|\theta_1 - \theta_0\|_2^2$$
$$\overset{(a)}{\leq} -\eta\left\|\nabla_\theta\hat{\ell}(\theta_0)\right\|_2^2 + \frac{\beta\eta^2 T}{2}\left\|\nabla_\theta\hat{\ell}(\theta_0)\right\|_2^2$$
$$= -\eta\left(1 - \frac{\beta\eta T}{2}\right)\left\|\nabla_\theta\hat{\ell}(\theta_t)\right\|_2^2$$
$$\overset{(b)}{\leq} -\frac{\eta}{2}\left\|\nabla_\theta\hat{\ell}(\theta_0)\right\|_2^2 \overset{(c)}{\leq} -\frac{\eta}{2}\ell_0'^\top K_{\mathrm{NTK}}(\theta_0)\ell_0' \tag{58}$$
$$\overset{(d)}{\leq} -\frac{\eta}{2}\lambda_{\min}(K_{\mathrm{NTK}}(\theta_0))\|\ell_0'\|_2^2 \overset{(e)}{\leq} -\frac{\eta}{2}\lambda_0 4\hat{\ell}(\theta_0)$$
$$\leq -\eta\lambda_0\hat{\ell}(\theta_0)$$
$$\implies \quad \hat{\ell}(\theta_1) \leq (1 - \eta\lambda_0)\hat{\ell}(\theta_0),$$

where (a) follows from the gradient descent update; (b) follows from our choice of step-size $\eta \leq \frac{1}{\beta N}$ so that $-(1 - \frac{\beta\eta T}{2}) \leq -\frac{1}{2}$; (c) follows from the following property valid for any iterate $\theta_t \in \mathbb{R}^p$,

$$\left\|\nabla_\theta\hat{\ell}(\theta_t)\right\|_2^2 = \left\|\sum_{n=1}^N \ell_{t,n}'\nabla_\theta f(\theta_t; \mathbf{x}_n)\right\|_2^2 = \sum_{n=1}^N\sum_{n'=1}^N \ell_{t,n}'\ell_{t,n'}'\langle\nabla_\theta f(\theta_t; \mathbf{x}_n), \nabla_\theta f(\theta_t; \mathbf{x}_{n'})\rangle = \ell_t'^T K_{\mathrm{NTK}}(\theta_t)\ell_t' ,$$

where $\ell'_t := [\ell'_{t,n}] \in \mathbb{R}^N$, with $\ell'_{t,n} = -2(y_i - f(\theta_t; \mathbf{x}_n))$ and $\ell_{t,n} = (y_n - f(\theta_t; \mathbf{x}_n))^2$; (d) follows from the definition of minimum eigenvalue; and (e) follows from the following property valid for any iterate $\theta_t \in \mathbb{R}^p$,

$$\|\ell'_t\|_2^2 = \sum_{n=1}^{N} \ell'^2_{t,n} = 4 \sum_{n=1}^{N} (y_n - f(\theta_t; \mathbf{x}_n))^2 = 4\hat{\ell}(\theta_t) . \tag{59}$$

Notice that, from our choice of step-size $\eta < \frac{1}{\lambda_0}$, we have that $1 - \eta\lambda_0 \in (0,1)$.

Continuing with our proof by induction, we take the following induction hypothesis: we assume that

$$\hat{\ell}(\theta_t) \le (1 - \eta\lambda_0)^{t-1} \hat{\ell}(\theta_0) \tag{60}$$

and that $\theta_\tau \in B_{\rho,\rho_1}^{\text{Frob}}(\theta_0)$ with $\rho_1 = 0$ for $\tau \le t$.

First, based on the choice of the step sizes, we show that $\theta_{t+1} \in B_{\rho,\rho_1}^{\text{Frob}}(\theta_0)$ with $\rho_1 = 0$. To see this, note that, using similar inequalities as in our analysis for the case $t = 1$,

$$\|\theta_{t+1} - \theta_0\|_2 \le \sum_{\tau=0}^{t} \|\theta_{\tau+1} - \theta_\tau\|_2 = \sum_{\tau=0}^{t} \eta \|\nabla_\theta \hat{\ell}(\theta_\tau)\|_2 \le 2\varrho\eta \sum_{\tau=0}^{t} \sqrt{\hat{\ell}(\theta_\tau)}$$

$$\overset{(a)}{\le} 2\varrho\eta \left( \sum_{\tau=0}^{t} (1 - \eta\lambda_0)^{\tau/2} \right) \sqrt{\hat{\ell}(\theta_0)} \le 2\varrho\eta \frac{\sqrt{\hat{\ell}(\theta_0)}}{1 - \sqrt{1 - \eta\lambda_0}}$$

$$\overset{(b)}{\le} \frac{4\varrho\sqrt{\hat{\ell}(\theta_0)}}{\lambda_0} \le 4\varrho\sqrt{c_{0,4\nu_0}} \frac{\sqrt{T}}{\lambda_0} \overset{(c)}{\le} \rho ,$$

where (a) follows from our induction hypothesis, (b) follows from $\frac{x}{1 - \sqrt{1 - x\lambda_0}} \le \frac{2}{\lambda_0}$ for $x < \frac{1}{\lambda_0}$, and (c) follows since $\rho = \Theta(\frac{\sqrt{T}}{\lambda_0})$.

Now, we have

$$\lambda_{\min}(K_{\text{NTK}}(\theta_t)) \overset{(a)}{\ge} \lambda_{\min}(K_{\text{NTK}}(\theta_{t-1})) - 4c_H \varrho^2 \frac{T}{\sqrt{m}} \eta \sqrt{\hat{\ell}(\theta_{t-1})}$$

$$\ge \lambda_{\min}(K_{\text{NTK}})(\theta_0) - 4c_H \varrho^2 \eta \frac{T}{\sqrt{m}} \sum_{\tau=0}^{t-1} \sqrt{\hat{\ell}(\theta_\tau)}$$

$$\overset{(b)}{\ge} \lambda_0 - 4c_2 \varrho^2 \eta \frac{T}{\sqrt{m}} \left( \sum_{\tau=0}^{t} (1 - \eta\lambda_0)^{\tau/2} \right) \sqrt{\hat{\ell}(\theta_0)}$$

$$\ge \lambda_0 - 8c_2 \varrho^2 \frac{T\sqrt{\hat{\ell}(\theta_0)}}{\sqrt{m}} \frac{\eta}{1 - \sqrt{1 - \eta\lambda_0}}$$

$$\overset{(c)}{\ge} \lambda_0 - \frac{T^{3/2}}{\sqrt{m}} \frac{\bar{c}\eta}{1 - \sqrt{1 - \eta\lambda_0}}$$

$$\ge \lambda_0 - \frac{2\bar{c}T^{3/2}}{\lambda_0\sqrt{m}}$$

where (a) follows from Lemma 1, (b) follows by the induction hypothesis, and (c) follows with $\bar{c} = 8\sqrt{c_{0,\sigma_1}} c_3$. Then, with $m \ge 16\bar{c}^2 \frac{T^3}{\lambda_0^4}$ we have

$$\lambda_{\min}(K_{\text{NTK}}(\theta_t)) \ge \lambda_0/2 . \tag{61}$$

Since $\theta_t, \theta_{t+1} \in B_{\rho,\rho_1}^{\mathrm{Frob}}(\theta_0)$ with $\rho_1 = 0$, we now take the smoothness property and further obtain, using similar inequalities as in our analysis for the case $t = 1$,

$$
\begin{aligned}
\hat{\ell}(\theta_{t+1}) - \hat{\ell}(\theta_t) &\leq \langle \theta_{t+1} - \theta_t, \nabla_\theta \hat{\ell}(\theta_t) \rangle + \frac{\beta T}{2} \|\theta_{t+1} - \theta_t\|_2^2 \\
&\overset{(a)}{\leq} -\eta \left\| \nabla_\theta \hat{\ell}(\theta_t) \right\|_2^2 + \frac{\beta \eta^2 T}{2} \left\| \nabla_\theta \hat{\ell}(\theta_t) \right\|_2^2 \\
&= -\eta \left( 1 - \frac{\beta \eta T}{2} \right) \left\| \nabla_\theta \hat{\ell}(\theta_t) \right\|_2^2 \\
&\leq -\frac{\eta}{2} \ell_t'^\top K_{\mathrm{NTK}}(\theta_t) \ell_t' \\
&\leq -\frac{\eta}{2} \lambda_{\min}(K_{\mathrm{NTK}}(\theta_t)) \|\ell_t'\|_2^2 \\
&\overset{(b)}{\leq} -\frac{\eta}{2} \frac{\lambda_0}{2} 4\hat{\ell}(\theta_t) \\
\implies \quad \hat{\ell}(\theta_{t+1}) &\leq (1 - \eta\lambda_0) \hat{\ell}(\theta_t),
\end{aligned}
\tag{62}
$$

where (a) follows from the gradient descent update and (b) from our recently derived result. That establishes the induction step and completes the proof. $\square$

## F    RELATED WORK

**Contextual Bandits:** The contextual bandit setting with linear losses has received extensive attention (see for eg. Abe et al., 2003; Chu et al., 2011; Abbasi-Yadkori et al., 2011; Agrawal and Goyal, 2013; Ban and He, 2020; 2021). Owing to its remarkable success, Lu and Van Roy (2017); Zahavy and Mannor (2020); Riquelme et al. (2018) adapted neural models to the contextual bandit setting. In these initial works all but the last layers of a DNN were utilized as a feature map to transform contexts from the raw input space to a low-dimensional space and a linear exploration policy was then learned on top of the last hidden layer of the DNN. Although these attempts have yielded promising empirical results, no regret guarantees were provided. Subsequently (Zhou et al., 2020) introduced the first neural bandit algorithm with provable regret guarantees that uses a UCB based exploration and Zhang et al. (2021) further extended it to the Thompson sampling approach. Both these approaches rely on Kernel bandits (Valko et al., 2013) and have a linear dependence on effective dimension $\tilde{d}$ of the (NTK) Neural Tangent Kernel (see Allen-Zhu et al., 2019b; Jacot et al., 2018; Cao and Gu, 2019b). Moreover both these algorithms require the inversion of a matrix of size equal to the number of parameters in the model at each step of the algorithm. Recently, Ban et al. (2022b) attained a regret bound independent of $\tilde{d}$, but makes distributional assumptions on the context. (Qi et al., 2022; 2023; Ban et al., 2021; 2022a) shows the successful application of neural bandits on the recommender systems.

**Overparameterized Models:** Considerable progress has been made in understanding the expressive power of Deep Neural Networks in the overparameterized regime (Du et al., 2019; Allen-Zhu et al., 2019b;a; Cao and Gu, 2019b; Arora et al., 2019a). It has been shown that the dynamics of the Neural Tangent Kernel always stays close to random initialization when the network is wide enough (Jacot et al., 2018; Arora et al., 2019b). Further Cao and Gu (2019b) demonstrate that the loss function of neural network has the almost convexity in the overparameterized regime while Liu et al. (2020; 2022); Frei and Gu (2021); Charles and Papailiopoulos (2018) study neural models under Polyak-Lojasiewicz (PL) Type conditions (Polyak, 1963; Lojasiewicz, 1963; Karimi et al., 2016). Recently, Banerjee et al. (2023) provided a bound on the spectral norm of the Hessian of the netowrk over a larger layerwise spectral norm radius ball (in comparison to Liu et al. (2020)) and show geometric convergence in deep learning optimization using restricted strong convexity. Our regret analysis makes use of these recent advances in deep learning.

## G    DETAILS OF EXPERIMENTS

**Baselines.** We choose four neural bandit algorithms: (1) NeuralUCB (Zhou et al., 2020) maintains a confidence bound at every step using the gradients of the network and selects the most optimistic arm. (2) NeuralTS (Neural Thompson Sampling) (Zhang et al., 2021) estimates the rewards by drawing them from a normal distribution whose mean is the output of the neural network and the variance is a quadratic form of the gradients of the network. The arm with the maximum sampled reward from this distribution is selected. (3) EE-Net (Ban et al., 2022b): In addition to employing an Exploitation network for learning the output function, it uses another Exploration network to learn the potential gain of exploring in relation to the current estimated reward. (4) NeuralEpsilon employs the $\epsilon$-greedy strategy: with probability $1 - \epsilon$ it chooses the arm with the maximum estimated reward generated by the network and with probability $\epsilon$ it chooses a random arm.

**Datasets.** We consider a collection of 6 multiclass classification based datasets from the `openml.org` platform: covertype, fashion, MagicTelescope, mushroom, Plants and shuttle. Following the evaluation setting of existing works (Zhou et al., 2020; Ban et al., 2022b), given an input $\mathbf{x}_t \in \mathbb{R}^d$ for a $K$-class classification problem, we transform it into $dK$ dimensional context vectors for each arm: $\mathbf{x}_{t,1} = (\mathbf{x}_t, \mathbf{0}, \mathbf{0}, \ldots, \mathbf{0})^T, \mathbf{x}_{t,2} = (\mathbf{0}, \mathbf{x}_t, \mathbf{0}, \ldots, \mathbf{0})^T), \ldots, \mathbf{x}_{t,K} = (\mathbf{0}, \mathbf{0}, \ldots, \mathbf{0}, \mathbf{x}_t)^T$. The reward is defined as $1$ if the index of selected arm equals $\mathbf{x}$'s ground-truth class; otherwise, the reward is $0$.

**Architecture:** Both NeuSquareCB and NeuFastCB use a 2-layered ReLu network with 100 hidden neurons. The last layer in NeuRIG uses a linear activation while NeuFastCB uses a sigmoid. Following the scheme in Zhou et al. (2020) and Zhang et al. (2021) we use a diagonal matrix approximation in both NeuralUCB and NeuralTS to save computation cost in matrix inversion. Both use a 2-layered ReLu network with 100 hidden neurons and the last layer uses a linear activation. We perform a grid-search over the regularization parameter $\lambda$ over $(1, 0.1, 0.01)$ and the exploration parameter $\nu$ over $(0.1, 0.01, 0.001)$. NeuralEpsilon uses the same neural architecture and the exploration parameter $\epsilon$ is searched over $(0.1, 0.05, 0.01)$. For EE-Net we use the architecture from

. For all the algorithms we also do a grid-search for the step-size over $(0.01, 0.005, 0.001)$.

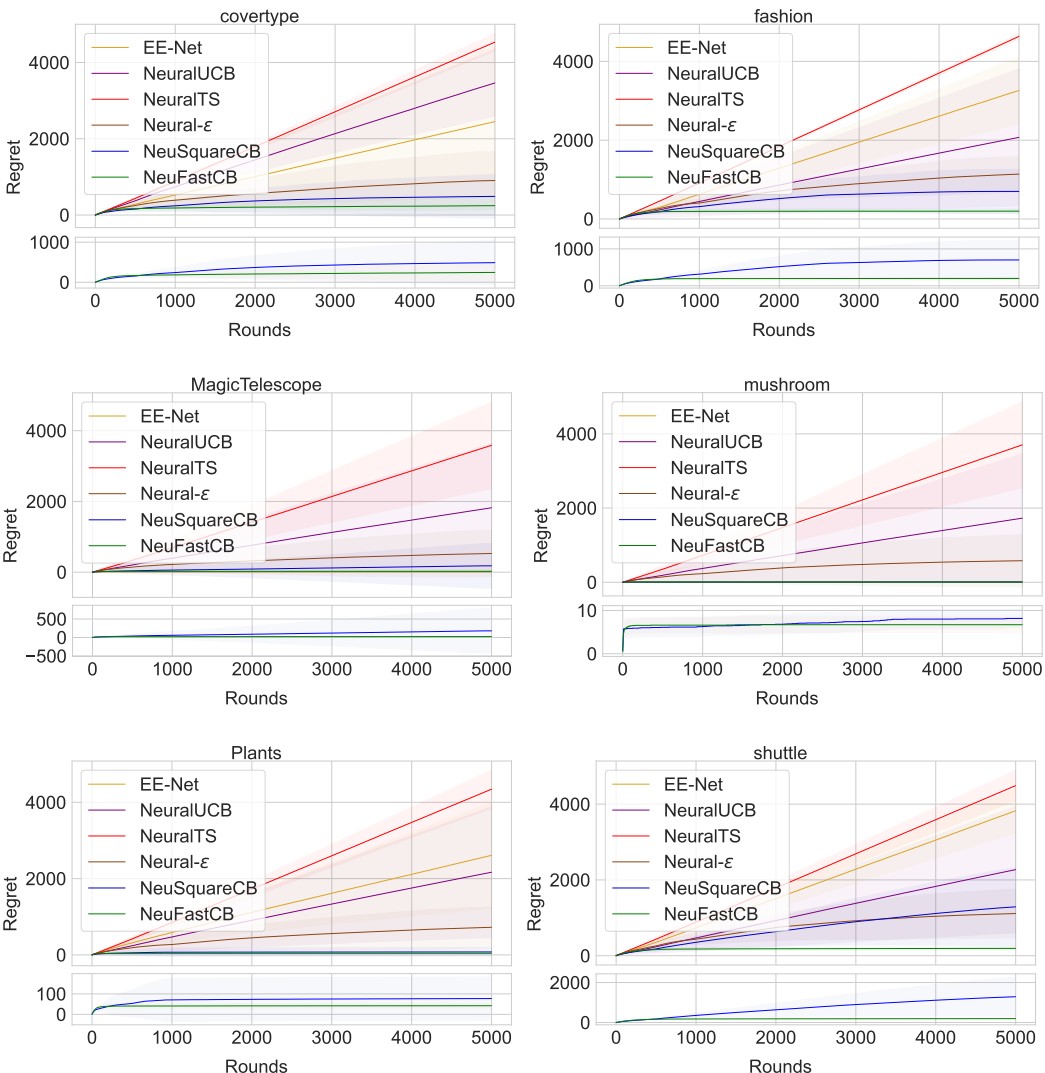

Figure 2: Figure 1 re-plotted for better visualization. Comparison of cumulative regret of NeuSquareCB and NeuFastCB with baselines on real-world datasets (averaged over 20 runs). The subplot below each figure plots the regret curve for NeuSquareCB and NeuFastCB again.

## G.1 NeuSquareCB AND NeuFastCB

Although our regret bounds are for the regularized network as defined in (10), the results presented in Section 5 are for the un-perturbed network. In this section we compare the un-perturbed network with the regularized one for different choices of perturbation constant $c = c_{\text{reg}}/m^{1/4}$.

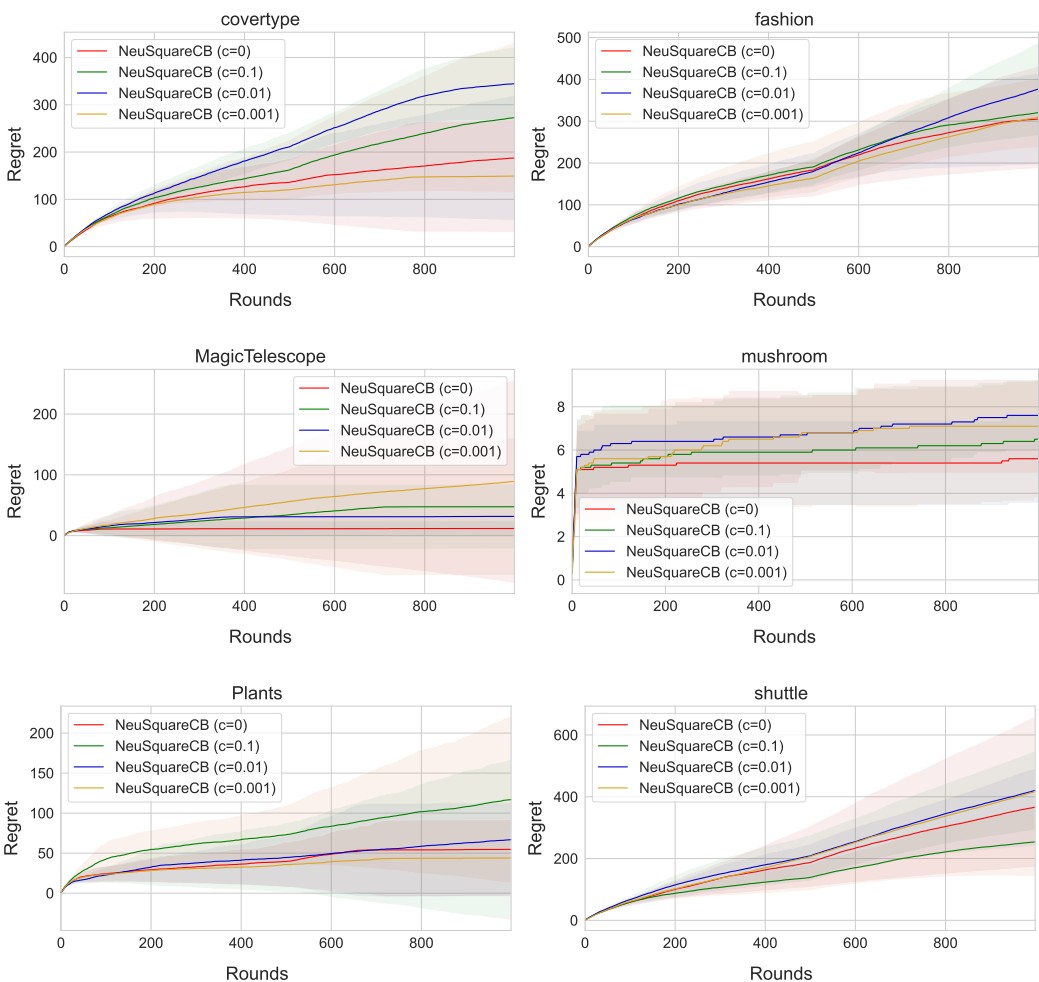

Figure 3: Comparison of cumulative regret of `NeuSquareCB` for different choices of the regularization parameter $c = c_{\text{reg}}/m^{1/4}$ for the model defined in (10) (averaged over 10 runs).

As is evident from Figure 3 and 4, the cumulative regrets attained by the un-perturbed networks ($c = 0$) are more or less similar to the perturbed one on these data-sets. However, the output perturbation ensured a provable $\mathcal{O}(\sqrt{KT})$ and $\mathcal{O}(\sqrt{KL^*} + K)$ regret bound for NeuSquareCB and NeuFastCB respectively. For the set of problems in our experiments we observe that the non-perturbed versions of NeuSquareCB and NeuFastCB behave similar to the perturbed versions, but do not come with a provable regret bound. In particular, one may be able to construct a problem where the non-perturbed version performs poorly.

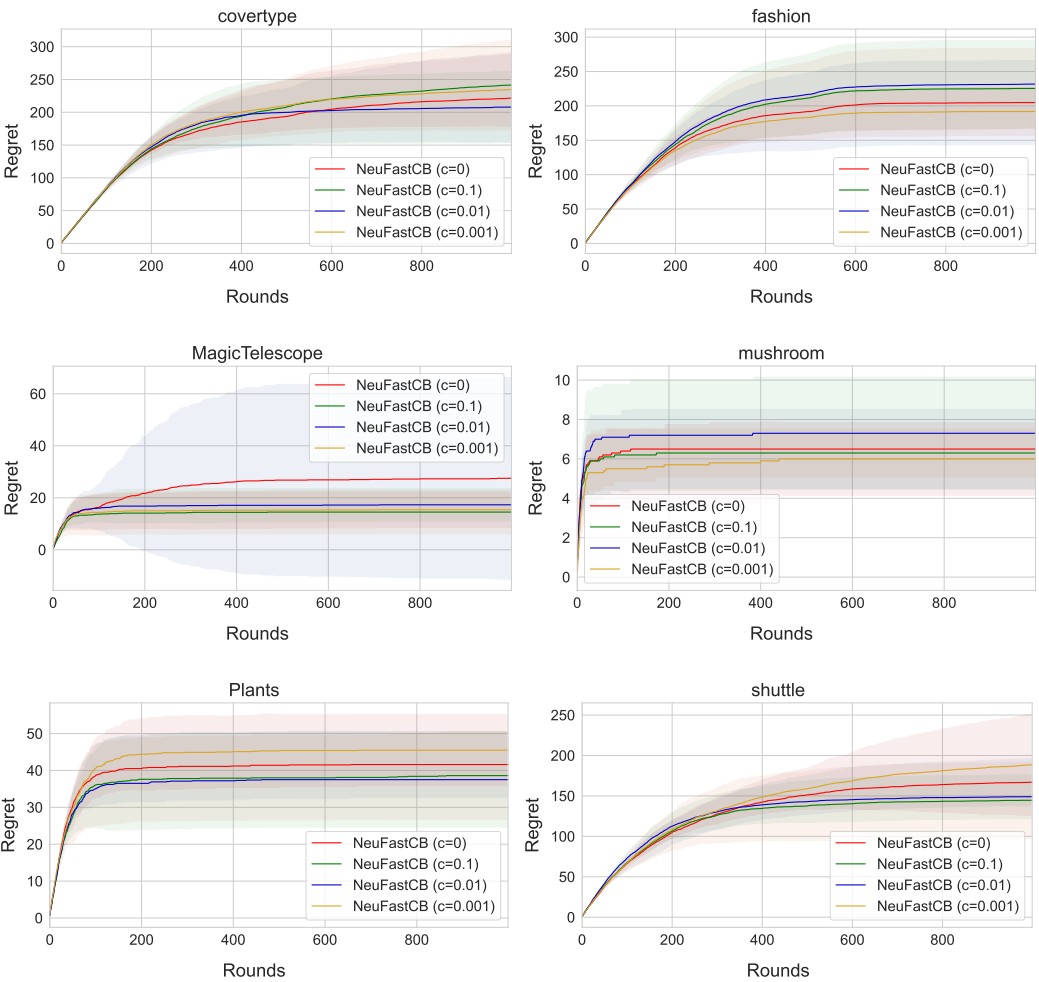

Figure 4: Comparison of cumulative regret of `NeuFastCB` for different choices of the regularization parameter $c = c_{\mathrm{reg}}/m^{1/4}$ for the model defined in (10) (averaged over 10 runs).

