# OpenReview forum: "Contextual Bandits with Online Neural Regression"
_ICLR.cc/2024/Conference — ICLR 2024 poster_

### Official Review · Reviewer_Gbgm · 2023-10-28

**Soundness:** 3 good
**Presentation:** 4 excellent
**Contribution:** 3 good
**Rating:** 8
**Confidence:** 4

**Summary:**

This paper studies neural contextual bandits (CB) with the reduction to online regression under the realizability assumption.
Using existing results for wide neural networks, it immediately gets a regret of $O(T^{3/4})$ for neuCB.

When the loss function of the online regression satisfies $\epsilon$-almost convexity, Quadratic Growth condition and uniqueness of minima, this paper provides a $O(\log T) + \epsilon T$ regret for the online regression.

Then, since the above result does not directly apply to wide NN due to the violation of uniqueness of minima, this paper proposes a novel trick of adding a small perturbation to the NN prediction. This seemingly simple trick makes the loss satisfy the Quadratic Growth condition with a unique minimum. Consequently, an $O(\log T)$ regret for the online regression with KL loss and squared loss is established.

Furthermore, this paper provides a lower bound of $\Omega(T)$ for NeuralUCB and NeuralTS with an oblivious adversary.

Finally, Experimental results show that the proposed algorithms outperform baselines on 6 classification datasets.

**Strengths:**

**Significance**: this paper is a quite extensive theoretical work on NeuralCB. The theoretical results as well as the techniques are of independent interest.

**Quality**: the quality of the paper is good. Definitions are introduced without ambiguity; the theoretical analysis seems solid to me; experimental details are given.

**Originality**: this paper contains some novel idea. The perturbation trick is a very interesting trick for bypassing the restriction on the uniqueness of minima.

**Clarity**: this paper is very well written. The theoretical proof is divided into steps. A sketch of proof like this make it easier for the readers to follow and understand.

**Weaknesses:**

I did not detect any major technical flaw or major weakness in this paper.

**Questions:**

1. Is it possible to extend the current results to a broader choice of loss functions? It would be great if the authors can elaborate on this.

---

> ### Author Response · Authors · 2023-11-21
> **Response to Reviewer Gbgm**
>
> We thank the reviewer for the encouraging comments. The summary provided by the reviewer indeed captures the main story-line of this work and we have incorporated the structure in our general response as well. We adress the question raised by the reviewer below.
> ***
> **[Q1] Extension to other loss functions:**
> Our QG regret bound (Theorem 3.1) is a general result that holds for any loss function as longs as it satisfies the following conditions: (i) almost convexity, (ii) QG and (iii) unique minima. In Theorem 3.2 and 3.3 we show that these conditions hold for square loss and KL loss respectively with the perturbed network. These results do not immediately extend to other loss functions and one needs to carry out an independent analysis for each such loss function.

---

> > ### Author Response · Authors · 2023-11-23
> >
> > We hope you had a chance to go over our responses to your queries. Since the discussion period is ending soon, we were wondering if you needed any further clarifications.

---

> ### Comment · Reviewer_Gbgm · 2023-12-04
> **Response**
>
> I would like to thank the authors for the rebuttal. It addressed my question.
>
> Best,
> Reviewer Gbgm

---

### Official Review · Reviewer_uhN1 · 2023-10-31

**Soundness:** 3 good
**Presentation:** 3 good
**Contribution:** 3 good
**Rating:** 6
**Confidence:** 4

**Summary:**

This work provides a solid investigation on neural regression in contextual bandits. One interesting point is to use perturbation to construct a surrogate network output with unique minimizer. In that analysis, the work manage to get a approximate loss to satisfy strong-convexity-like condition (Quadratic Growth condition) to get analysis through. The article is full of useful theoretical insights and enjoyable to read.

**Strengths:**

The work provides solid investigation on improving online neural regression for better online decision making. The paper also is well-written, easy to read for audience with relative expertise.

**Weaknesses:**

Maybe not so readable for people without theoretical background.

**Questions:**

1. One particular point that draws my attention is to adopt KL loss in the online gradient descent step. Could you make comments on when should we use NeuFastCB and when should we use NeuSquareCB in practice? In particular, how the dimension or datatype of the real dataset impact this decision?

2. Could you provide your comments on why NeuFastCB and NeuSquareCB outperform NeuralTS?

---

> ### Author Response · Authors · 2023-11-21
> **Response to Reviewer uhN1**
>
> We thank the reviewer for the encouraging comments. We are especially glad that the reviewer found the paper full of useful theoretical insights and enjoyable to read. Despite being a theory heavy paper, we had put in a lot of efforts to ensure that the results are accessible to wide range of people. We have further updated the draft to incorporate suggestions from this and other reviewers. Below we respond to all the questions posed by the reviewer.
> ***
> **[Q1] NeuSquareCB vs NeuFastCB:**
> This is an interesting question and we thank the reviewer for bringing this up.
> In practice, we recommend using NeuFastCB, both because it seems to perform better empirically in our experiments and theory results are much stronger (see below).
>
> Next recall that the regret bound for NeuFastCB is $\mathcal{O}(\sqrt{KL^*} + K)$ and that of NeuSquareCB is $\mathcal{O}(\sqrt{KT})$. Note that the regret bound for NeuFastCB is a first order bound, i.e., it depends on $L^*$, the loss of the best policy and not the time horizon $T$ as in NeuSquareCB. Since $L^* \leq T$ then $\mathcal{O}(\sqrt{KL^*}) \leq \mathcal{O}(\sqrt{KT})$. Therefore NeuFastCB is expected to perform better in most settings ``in practice'', especially when $L^*$ is small, i.e., the best policy has low regret.
>
> Also note that going by the upper bounds on the regret, especially the dependence on $K$, the number of arms, NeuSquareCB could outperform NeuFastCB only if $L^* = \Theta(T)$ and $K>>T$.
>
> Since this discussion would indeed help users choose between the two algorithms, we have included a brief discussion about this in Section 4 of the new draft (Remark 4.1 in the updated draft).
> ***
> **[Q2] Why NeuFastCB and NeuSquareCB outperform NeuralTS:**
>
> The improved empirical performance of NeuFastCB and NeuSquareCB in comparison to NeuralTS and NeuralUCB can be explained by the following.
>
> 1. Both NeuralUCB and NeuralTS need to invert a matrix given by $\sum_{i=1}^t g_i g_i^T + \lambda I$, where $g_i$ is the vectorized gradient of the neural network at time $i$ and $\lambda$ is the regularization parameter. However, since the matrix is of very high dimensions, both the works implement their respective algorithms with a diagonal approximation to the matrix to make it computationally feasible. The regret guarantee does not hold with the approximation and {possibly} leads to sub-par performance.
>
> 2. In this work we also show that the regret bounds for NeuralUCB and NeuralTS (even without the diagonal approximation) is $\Omega(T\sqrt{K})$ which could explain why their performance is not entirely satisfactory. Further note that the regret of our algorithm NeuFastCB is sub-linear in $L^*$, the loss of the best policy and not the time horizon $T$. Therefore, in situations where the best policy has low regret, our algorithm NeuFastCB will also have low regret.
> ***
> **[Q3] Readability for people without theoretical background.**
>
> We thank the reviewer for reminding us to be mindful that the contributions should be accessible to a broader set of audience. We describe some of the measures we had adopted along with some changes made to the new draft to ensure that.
>
> 1. We give a gentle introduction to NeuCBs in Section 1 and specifically discuss the research gaps in the current NeuCB literature in Section 1.1 to ensure that the reader can appreciate the contributions of this work in the context of available works.
> 2. All the algorithms and clearly stated and should be easy to implement.
> 3. In Section 1.2 we enumerate the contributions of this work that clearly describe the main story-line:
> >**(A)** QG condition along with additional standard assumptions gives logarithmic regret if the loss has unique minima,
> \
> >**(B)** Neural networks do not have unique minima
> \
> >**(C)** Small perturbation to the output of neural networks yield unique minima as well as all other desirable properties.
> 4. The relative merits of the different approaches in NeuCBs are summarized in Table 1.
> 5. In the experiments section we clearly describe the baselines, datasets and how contexts are chosen adversarially along with a description of the results.
> ***
> If the reviewer has further suggestions to help make the paper more accessible, we would be glad to include them in the subsequent draft.

---

> > ### Author Response · Authors · 2023-11-23
> >
> > We hope you had a chance to go over our responses to your queries. Since the discussion period is ending soon, we were wondering if you needed any further clarifications.

---

### Official Review · Reviewer_EQUF · 2023-11-01

**Soundness:** 2 fair
**Presentation:** 2 fair
**Contribution:** 3 good
**Rating:** 5
**Confidence:** 3

**Summary:**

This paper studies the contextual bandits problem. The paper proposed NeuCB, an algorithm based on applying SquareCB to neural networks with some assumptions. The paper proved a novel online regression guarantee for neural networks satisfying QG condition and some other assumptions. Based on this, the paper converted this into a $\sqrt T$-type bandit regret bound for NeuCB. Furthermore, the paper presented experiments on various real-world classification datasets to show its advantage over existing neural contextual bandits algorithms.

**Strengths:**

1. The online regression bound is novel and could further benefit the community.
2. The paper presents experiments to corroborate their theoretical findings.

**Weaknesses:**

1. The paper seems not to mention about whether the reward function $h$ in Assumption 1 is realizable by a neural network. Is it the case?
2. Similar to most previous NTK bandit paper, the paper made the assumption on both gaussian initialization and lower-bounded eigenvalue of the NTK. This limits the scope of this and all the NTK bandits papers.
3. The paper is written in a nested way so that I have to chase after the links between theorems, lemmas, and assumptions to figure out the true assumption set of a theorem: for each theorem, I need to consider its assumption and the assumptions in all lemmas/theorems it used.
4. The bounds in various theorems used $\widetilde{\mathcal O}$ without saying what's exactly being omitted by it. I suggest the author complete this part of information.

**Questions:**

1. In regret bounds Theorems 4.1 and 4.2, what's the dependency of the regret bound w.r.t. width $m$? If there's dependency, could you please write it out? If neither the Theorem nor the algorithm used $m$, then we could omit it.

---

> ### Author Response · Authors · 2023-11-21
> **Response to Reviewer EQUF**
>
> We thank that the reviewer for the encouraging comments. We are glad that the reviewer is of the opinion that the online regression bounds are novel and could further benefit the community and that experiments do corroborate the theoretical findings. We are further thankful for the interesting questions and suggestions brought by the reviewer. Below we respond to all of them.
>
> ***
> **[Q1] Realizability of the reward function:** The reward function $h$ could be arbitrary, and we do not assume any specific form. The interpolation result (Theorem E1) shows that the neural network defined in (1) of the paper with Assumption 2 (network initialization) and Assumption 5 (NTK full rank) can realize any function $h$ on a finite set of $T$ points. To make the discussion clearer we have added a short remark (Remark 2.1) in the new draft.
> ***
> **[Q2] Gaussian Initialization and lower bounded eigen value of the NTK**
> 1. **Gaussian Initialization:** Pytorch supports both "Xavier Normal'' and "Kaiming Normal'' (also called He initialization) both of which use a gaussian distribution for initialization (see pytorch/torch/nn/init.py in the pytorch repository).
> Further many works in deep learning assume that the parameters are initialized using a gaussian distribution (eg. [Du et al., 2019],[Du et al., 2018],[Chen et al. 2015],[Allen-Zhu et al., 2019a],[Allen-Zhu et al., 2019b],[Cao and Gu, 2019]).
> \
> \
> Further, we expect the analysis to go through (however, this will need additional work) for any sub-Gaussian distribution based initialization, with suitable changes in (distribution dependent) constants in the results. Initialization from the uniform distribution is also supported in PyTorch (default initialization for linear layer) and is an example of a sub-Gaussian distribution.
>
> 2. **NTK full rank:**  The assumption that the NTK is full rank is used in many deep learning works ([Du et al., 2019],[Liu et al., 2022],[Du et al., 2018]). We use the assumption to show that PL (and hence the QG condition holds in our analysis). We share the following points regarding the assumption.
> \
> \
> **(1)** In practice experiments have also shown that the assumption of NTK being full rank holds (see [Fort et al. 2020]). Moreover, as long as there are no two contexts that are identical or in parallel, the assumption holds. We can ensure this is satisfied by removing the duplicate contexts.
> \
> \
> **(2)** It should be noted that NeuralUCB and NeuralTS also assume that NTK is full rank. Our lower bound results show that their regret bounds are $\Omega(T)$ (see Appendix A of our paper). Therefore even under Assumption 5 our regret bounds are the first sub-linear bounds for NeuCBs when contexts are otherwise chosen adversarially.
> ***
> **[Q3] Writing Style:**
>
> To ensure that our work is accessible to a broader audience and is easy to follow we have incorporated suggestions from this and other reviewers. We highlight some of the aspects that should make the presentation better.
> 1. In the proof of the main theorems (Theorem 3.1 and 3.2), we included a proof sketch to ensure that the reader has an overall idea of the way the analysis proceeds.
> 2. We have also ensured in the Appendix that all the Lemma statements explicitly state the assumptions being used. To enhance the readability we have also included a description every time we invoke an assumption in the proof in the new draft. We hope this would eliminate moving back and forth between the assumption statements in the main paper and the proof details in the appendix, thus enhancing the proof reading experience.
> 3. For a non-theoretical audience we give a gentle introduction to NeuCBs in Section 1 and specifically discuss the research gaps in the current NeuCB literature in Section 1.1 to ensure that the reader can appreciate the contributions of this work in the context of available works.
> 4. Further, in Section 1.2 we enumerate the contributions of this work that clearly describe the main story-line: (A) QG condition gives logarithmic regret if the loss has unique minima, (B) neural networks do not have unique minima (C)  small perturbation to the output of neural networks yield unique minima as well as all other desirable properties.
>
> If the reviewer has further suggestions to help make the paper more accessible, we would be glad to include them in the subsequent draft.
> ***
> **[Q4] Definition of $\tilde{\mathcal{O}}$**
>
> We thank the reviewer for bringing this to our attention. Indeed a precise definition would help improve the exposition and we have included the definition in Section 2.
> ***

---

> > ### Author Response · Authors · 2023-11-21
> > **Response to Reviewer EQUF**
> >
> > [Fort et al. 2020] Fort S, Dziugaite GK, Paul M, Kharaghani S, Roy DM, Ganguli S. Deep learning versus kernel learning: an empirical study of loss landscape geometry and the time evolution of the neural tangent kernel. Advances in Neural Information Processing Systems. 2020;33:5850-61.
> > [Du et al., 2019]: S. Du, J. Lee, H. Li, L. Wang, and X. Zhai. Gradient descent finds global minima of deep neural networks. In International conference on machine learning, pages 1675–1685. PMLR, 2019.
> >
> > [Du et al., 2018]: Simon S Du, Xiyu Zhai, Barnabas Poczos, and Aarti Singh. Gradient descent provably optimizes over-parameterized neural networks. In: arXiv preprint arXiv:1810.02054 (2018).
> >
> > [Chen et al. 2015] Chen X, Minasyan E, Lee JD, Hazan E. Provable regret bounds for deep online learning and control. arXiv preprint arXiv:2110.07807. 2021 Oct 15.
> >
> > [Allen-Zhu et al., 2019a]: Z. Allen-Zhu, Y. Li, and Y. Liang. Learning and generalization in overparameterized neural networks, going beyond two layers. Advances in neural information processing systems, 32, 2019a
> >
> > [Allen-Zhu et al., 2019b]: Z. Allen-Zhu, Y. Li, and Z. Song. A convergence theory for deep learning via over-parameterization. In International Conference on Machine Learning, pages 242–252. PMLR, 2019b.
> >
> > [Cao and Gu, 2019]: Y. Cao and Q. Gu. Generalization bounds of stochastic gradient descent for wide and deep neural networks. Advances in neural information processing systems, 32, 2019.

---

> > > ### Author Response · Authors · 2023-11-23
> > >
> > > We hope you had a chance to go over our responses to your queries. Since the discussion period is ending soon, we were wondering if you needed any further clarifications.

---

### Official Review · Reviewer_eHnV · 2023-11-04

**Soundness:** 3 good
**Presentation:** 2 fair
**Contribution:** 2 fair
**Rating:** 5
**Confidence:** 3

**Summary:**

The paper provides $O(\log T )+\epsilon T$ regret for online regression when the loss function satisfies $\epsilon$-almost convexity, QG condition, and has unique minima. The authors show that even though QG with unique minima may not be realistic, adding a suitably small random perturbation to the predictions, makes the losses satisfy QG with unique minima. Using a reduction from contextual bandits to online regression oracles, the authors use the results to show regret bounds for contextual bandits.

**Strengths:**

- The paper shows nearly-constant regret for online regression under certain assumptions.
- Some of the assumptions are valid for wide range of networks or can be enforced by perturbing the loss function.
- The regret bounds for contextual bandits are tight in some cases.

**Weaknesses:**

- The results require strong assumptions (2.a, 3, 5).
- The failure probability is large (grows with T) and cannot be controlled (see for example Thm 3.2).
- The regret bounds for contextual bandits grow with the number of arms rather than a complexity measure of the underlying class of functions. A typical scenario for CBs is when the number of arms is very large or even infinite. This does not recover even the simple result of contextual linear bandits where the regret grows as $\tilde{O}(\sqrt{dT\log K})$.

**Questions:**

Please see weaknesses.

---

> ### Author Response · Authors · 2023-11-21
> **Response to Reviewer eHnV**
>
> We are thankful to the reviewer for their insightful questions and suggestions. Below we address all of them.
> ***
> **[Q1] Restrictive Assumptions**:
>
> >**Assumptions 2 and 3:** We clarify that Assumption 2 and 3 were used only in the preliminary QG regret result (Theorem 3.1) and not by the regret bounds for NeuSquareCB and NeuFastCB (Theorem 3.1 and 3.2 respectively). For our algorithms, **_we rigorously proved that these assumptions hold_** for the class of models being used and the choice of loss functions with output perturbation. For example, in the proof sketch of Theorem 3.2 Assumption 3 is shown to be satisfied in Step-1, Assumption 2 (a) and (b) are shown to be satisfied in Step-2, and Assumption 2 (c) in Step-3. To make sure that this is clearer, we have added the following sentence at the beginning of the proof sketch: "Note that we do not use Assumptions 2 and 3 and, but rather explicitly prove that they hold''.
>
> >**Assumptions 5:** The assumption that the NTK is full rank is used in many deep learning works ([Du et al., 2019],[Liu et al., 2022],[Du et al., 2018]). We use the assumption to show that PL (and hence the QG condition holds in our analysis). We share the following points regarding the assumption.
> \
> \
> **(1)** In practice experiments have also shown that the assumption of NTK being full rank holds (see [Fort et al. 2020]). Moreover, as long as there are no two contexts that are identical or in parallel, the assumption holds. We can ensure this is satisfied by removing the duplicate contexts.
> \
> \
> **(2)** It should be noted that NeuralUCB and NeuralTS also assume that NTK is full rank. Our lower bound results show that their regret bounds are $\Omega(T)$ (see Appendix A of our paper). Therefore even under Assumption 5 our regret bounds are the first sub-linear bounds for NeuCBs when contexts are otherwise chosen adversarially.
>
> ***
> **[Q2] Failure probability grows with $T$**
>
> We clarify that the expression was not completely simplified to only show the dependence on $T$. Like NeuralUCB, NeuralTS and EE-Net our results are for wide neural networks (i.e., width, $m \gg T$, the number of samples/steps). We have plugged in the choice of $m$ and simplified to obtain the final expression that is given by $\left(1 - \frac{C}{T^4}\right)$, where $C > 0$ is some constant. Note that this decreases with $T$ as expected. We thank the reviewer to bring this up. We have updated the new draft with the simplified expression to avoid this confusion. Also note that for any $\delta > 0$ we can choose $m \geq \frac{2CTL}{\delta}$, to ensure that the theorem holds with probability at least $1 - \delta$.
> ***
> **[Q3] Dependence on number of arms.**
>
> Our regret bounds indeed have a $\sqrt{K}$ dependence. We provide the following discussions about this aspect of the bounds.
>
> 1. NeuralUCB and NeuralTS provide regret bounds that depend on $\tilde{d}$ the effective dimension of the NTK matrix. As we show in Appendix-A both these bounds are in fact $\Omega(\sqrt{K}T)$ with the same dependence on $K$ as in this paper and a damaging linear dependence on the horizon $T$. Therefore under the same setting as in this paper there does not exist any NeuCB algorithm that even gives a sub-linear regret in the time horizon $T$ and as such our results do significantly improve the state of art in NeuCBs.
>
> 2. We inherit the dependence on the number of arms from the online regression to bandit reduction from [Foster and Rakhlin, 2020] and [Foster and Krishnamurthy, 2021] and are not artifacts of our analysis. Improving the dependence on $K$ is an important direction for future work.
> ***
> [Fort et al. 2020]: Fort S, Dziugaite GK, Paul M, Kharaghani S, Roy DM, Ganguli S. Deep learning versus kernel learning: an empirical study of loss landscape geometry and the time evolution of the neural tangent kernel. Advances in Neural Information Processing Systems. 2020;33:5850-61.
>
> [Foster and Rakhlin, 2020]: D. Foster and A. Rakhlin. Beyond ucb: Optimal and efficient contextual bandits with regression oracles. In International Conference on Machine Learning, pages 3199–3210. PMLR, 2020.
>
> [Foster and Krishnamurthy, 2021]: D. J. Foster and A. Krishnamurthy. Efficient first-order contextual bandits: Prediction, allocation, and triangular discrimination. Advances in Neural Information Processing Systems, 34, 2021.
>
> [Du et al., 2019]: S. Du, J. Lee, H. Li, L. Wang, and X. Zhai. Gradient descent finds global minima of deep neural networks. In International conference on machine learning, pages 1675–1685. PMLR, 2019.
>
> [Liu et al., 2022]: Chaoyue Liu, Libin Zhu, and Mikhail Belkin. Loss landscapes and optimization in over-parameterized non-linear systems and neural networks. Applied and Computational Harmonic Analysis, 2022.
>
> [Du et al., 2018]: Simon S Du, Xiyu Zhai, Barnabas Poczos, and Aarti Singh. Gradient descent provably optimizes over-parameterized neural networks. In: arXiv preprint arXiv:1810.02054 (2018).

---

> > ### Comment · Reviewer_eHnV · 2023-11-21
> >
> > Thank you for your reply.
> >
> > Could you please expand on what you mean by "the class of model being used"? What are the assumptions on that class of models?
> >
> > For assumption 5, how can you remove duplicate contexts? What if all contexts are the same? That should be an easy case to handle.
> >
> > I am not convinced by Q2 answer. I understand that the failure probability decreases with the width m. However, for a fixed width m, as time increases, we should be able to learn and achieve a small regret with high probability. The results in the paper do not show this.

---

> ### Author Response · Authors · 2023-11-22
> **Response to Comment by Reviewer eHnV**
>
> Thank you for replying to our response. Below we clarify the points raised by you.
>
> ***
>
> 1. **Class of model:** By the class of models being used we mean the feed forward network defined in (1) of the paper that satisfies Assumption 4 (gaussian initialization scheme of the network) and Assumption 5 (positive definite NTK).
>
> ***
>
> 2. **NTK Assumption:** Assumption 5 implies the $T$ contexts are unique, i.e., not duplicate, and duplicate contexts will imply Assumption 5 has been violated.
> \
> \
> Note that **all** existing work on neural contextual bandits ([Zhang et al., 2021],[Zhou et al., 2020],[Ban et al., 2022]) make this assumption. In fact, this is also the standard assumption is most work in optimization and generalization with neural networks. While we agree with you that relaxing the assumption will be a big advance, that is beyond the scope of the current work.
> \
> \
> Our work presents the first theoretically correct algorithm for NeuCBs with sub-linear regret bound under Assumption 5 (see Table 1 in the current paper) in the standard setting. In particular, no prior work has managed to do this even under Assumption 5. We have already specifically discussed that NeuralUCB and NeuralTS have $\Omega(T)$ regret and EE-Net needs i.i.d. contexts and needs to store all previous networks.
> \
> \
> On a more pragmatic side, one can remove duplicate contexts to ensure that the training data for neural networks is non-degenerate.
> From this perspective, the results are in terms of $T$, the number of unique contexts. More broadly, consider any standard learning setting, say classification or regression, with data $(x_t,y_t), t \in [T]$ but $x_t = x, \forall t \in [T]$. We effectively have one training point. We do not expect a learning algorithm to successfully learn anything in this setting.
>
> ***
> 3. **Width requirements:** All existing results in neural contextual bandits need width $m = \Omega(poly(T))$, i.e., $m$ has to grow with $T$. Keeping m fixed limits the representation power of the neural network, and such networks cannot be expected to model (including interpolate) an arbitrary function h on T points, where $T$ is arbitrarily large and the points are chosen by an adversary. An interesting variant of your question is: can we get similar results with a smaller $m$, perhaps $m = \mathcal{O}(T)$ or even $m = \mathcal{O}(\log T)$? No prior work has managed to accomplish this without additional assumptions, but this does constitute an important direction for future work.
> \
> \
> For a fair comparison with existing NeuCB algorithm, below we specifically discuss the width requirements for existing algorithms:
>
>     >**(i)** Neural Thompson Sampling (TS) [Zhang et al., 2021]:  $m (\log m)^{-3} \geq T^{13}$ (see Condition 4.1 in their paper).
>     \
>     \
>     >**(ii)** Neural UCB [Zhou et al., 2020]: $m \geq T^{16}$.  The main regret bound in Theorem 4.5 requires the width to be $\tilde{\Omega}(\text{poly}(T))$ but does not specify the exact dependence on $T$. However we can infer the exact dependence of $m$ on $T$ from the proof as follows. Consider the last term in the regret bound in the \emph{Proof of Theorem 4.5}: $m^{-1/6}\sqrt{\log m} T^{8/3} \lambda^{-2/3}L^3$. At the end of the proof the authors conclude that the above term is $\leq 1$ by choosing sufficiently large $m$. To ensure this one has to choose $m^{-1/6} \geq T^{8/3}$ or $m \geq T^{16}$.
>     \
>     \
>     >**(iii)** EE-Net [Ban et al., 2022]: $m \geq T^{30}$.  Theorem 1 requires the width to be $\tilde{\Omega}(\text{poly}(T))$ and does not specify the exact dependence on $T$. Again we can infer it from the proof as follows. Consider the term $\xi_1 = \Theta(\frac{t^4}{m^{1/6}})$ in equation (C.4). The final regret expression sums this term from $t=1$ to $t=T$ (see the display below C.5). The sum is therefore $\Theta(\frac{T^5}{m^{1/6}})$ and the authors make it $\mathcal{O}(1)$ by choosing sufficiently large $m$, which leads to choosing $m^{1/6} \geq (T^5)$ or $m \geq T^{30}$ (see the text below C.6).
>
> \
> Our results need $m = \Omega(T^5)$. Note that the width requirements of EE-Net is prohibitively large in comparison to our requirements. Further the other two algorithms NeuralUCB and NeuralTS have $\Omega(T)$ regret in worst case (we show this in Appendix A of the current paper). Therefore for NeuCB even in terms of width requirements our results are better than EE-Net, the only NeuCB algorithm with provably sub-linear regret.
> \
> \
> Also, following your line of thinking, another interesting question to consider is: How would the algorithm look like if T is not known upfront? Note that then we cannot set $m = \Omega(T^5)$. In this more general setting, the **double trick** would work. In essence, we work with $\mathcal{O}(\log T) $ episodes, and for $t \in (2^{k-1}, 2^k]$, we use a neural network with width $m = \mathcal{O}(2^{5k})$. Once $t$ crosses $2^k$, a new episode starts, and we initialize a new neural network with width $m = \mathcal{O}(2^{5(k+1)})$.

---

> > ### Comment · Reviewer_eHnV · 2023-11-22
> >
> > Thank you for your reply.
> >
> > 3. I am confused about the representation power part. I thought by the realizability assumption (assumption 1), you mean that the true function can be represented within the considered class of neural networks with width m? If that is the case then we do not need to worry about the representation power as the considered class of functions contain the true function?

---

> > > ### Author Response · Authors · 2023-11-22
> > >
> > > Thank you for replying. We appreciate the clarification questions.
> > >
> > > Assumption 1 assumes the expected reward can be represented by some function $h$.
> > > For $T$ steps, we rigorously prove that the realized values of the function can be interpolated by a neural network with width $m = poly(T)$, thus making it realizable by a neural network as defined in (1) of our paper.

---

> > > > ### Author Response · Authors · 2023-11-23
> > > >
> > > > We hope our discussions have helped clarify the doubts you had. Since the discussion period is ending soon, we were wondering if you needed any further clarifications.

---

### Official Review · Reviewer_PXpv · 2023-11-09

**Soundness:** 3 good
**Presentation:** 3 good
**Contribution:** 3 good
**Rating:** 6
**Confidence:** 4

**Summary:**

This paper studies the contextual bandit problem with the use of a neural network for online regression. The authors build upon recent progress in our understanding of neural networks, providing an online regression algorithm that for the KL and squared loss acheives O(log T) regret a significant improvement over previous O(sqrt(T)) results. Using this, they employ the reduction of contextual bandits to online regression presented in the SquareCB algorithm of Foster/Rakhlin.

**Strengths:**

- I found the paper to be easy to read, well-presented and interesting.
- The results are interesting and seem like a potentially important contribution.
- It was harder for me to gauge the novelty of the approach used to prove the given results.

**Weaknesses:**

- The main result of this paper is given in section 3, where the authors show that a noise perturbed version of the neural network can be used to obtain a O(log T) regret bound. From what I understood, compared to the past results of Chen which just used Assumption 2a, this result exploited 2b and 2c to gain an improvement. This is akin to the fast rates that one gets for gradient descent run on strongly convex functions vs convex functions. Though the result is natural and intuitive based on Chen, I failed to understand what technical contributions the authors made to achieve the result in Theorem 3.1. Is this a straightforward application of existing results, or did it require a new perspective? Is the noise perturbation the main novelty?  Further discussion would have helped couch this work in the existing literature.

- I was a bit confused about why Section 4 was presented in the context of KL-loss. Presumably it applies to squared loss as well given the results of the previous sections?

- I have some concerns about the experiments. Building on the previous comment, it seems that part of the novelty of this paper is the use of a perturbed neural network. In the experiments you compare your algorithm using KL loss to NeuSquareCB and show an improvement. However, it feels like the natural comparison would be NeuFastCB against a variant using the KL loss where you don’t perturb. Similarly for the square loss. This “ablation”-esque study would highlight the important of the algorithmic contributions you have proposed.

Minor Typos

- theta_t in equation 4 should be theta

**Questions:**

See above

---

> ### Author Response · Authors · 2023-11-21
> **Response to Reviewer PXpv: Novelty of Results and Analysis**
>
> We are encouraged by the reviewer's comments. As requested we provide a thorough discussion of the novel aspects of the results and analysis. Further we also clarify the concerns raised in the review.
> ***
> **[Q1] Novelty of our results and analysis**
> 1. **QG Result (Theorem 3.1):** The regret bound under QG condition (Theorem 3.1) is a new contribution. Unlike [Chen et al., 2021] that uses existing analysis for online regression with convex functions, we develop a new analysis using the following assumptions on the loss function (i) almost convexity, (ii) QG and (iii) unique minima.
> \
> \
> While Theorem 3.1 is new, (iii) is not satisfied by (wide) neural networks. So, the central challenge was: how to ensure unique minima for losses on neural networks while holding on to QG and almost convexity. The following two points describe this for square loss and KL loss respectively.
> ***
> 2. **Regret with square loss (Theorem 3.2):** Square loss for (wide) neural networks satisfies (i) almost convexity and (ii) QG. To ensure (iii) unique minima, our novel idea of combining predictions from perturbed networks and a very delicate analysis achieved the following (a) ensured Strong Convexity (SC) with SC constant $\mathcal{O}(1/\sqrt{m})$ and therefore unique minima, (b) QG condition with QG constant $\Theta(1)$, and (c) maintained almost convexity as before. There are two delicate aspects to this approach:
>     >**(i)**  Our analysis ensured that the QG constant is $\Theta(1)$. This was necessary since the regret bound in (8) was given by
>     $$\tilde{R}(T)\leq \mathcal{O}\left( \frac{\lambda^2}{\mu}\log T\right) + \epsilon T + 2 \underset{\theta \in B}{\inf}\sum_{t=1}^T \ell(y_t,g(\theta;x_t))$$ and the QG constant $\mu$ needed to be $\Theta(1)$ to ensure that the first term is indeed $\mathcal{O}(\log T)$. Also, $\lambda$ is the lipschitz constant of the loss and we also show that it is $\Theta(1)$ (see Lemma 5 for square loss and Lemma 9 for KL loss in the current paper)
>
>     >**(ii)** We added a small $\left(\Theta(\frac{1}{m^{1/4}})\right)$ perturbation to ensure strong convexity (SC) of the loss with SC constant $\Theta(\frac{1}{\sqrt{m}})$, but used this only to ensure uniqueness of the minima. Adding "small'' perturbation ensured that the perturbed network is not far away from the un-perturbed one and we used this to bound the regret for the original problem.
>
>     **Why ridge regularization does not work**: If one uses the traditional route of adding ridge ($L_2$ squared) regularization to the loss function, then although the loss now has a unique minima, the original analysis with $\Theta(1)$ PL/QG constant does not work.
>
>      **Why directly using SC does not work:** Naively using the $\mathcal{O}(\log T)$ regret for SC functions also does not work. This is because the constant hidden by $\mathcal{O}$ scales as $\frac{1}{\nu} = \sqrt{m}$, where $\nu$ is the SC constant. For large width models, $m \gg T$, and the bound is $O(\sqrt{m}\log T)$, which does not yield a $\mathcal{O}(\log T)$ bound (also see Remark 3.5 in the current paper).
> ***
> 3. **Regret with KL loss (Theorem 3.3):** We showed that a similar set of results hold for KL loss, and thereafter obtained a "first order" regret bound, a bound that is data-dependent in the sense that it scales sub-linearly with the loss of the best policy $L^*$ instead of $T$.
> \
> A noteworthy point about the analysis is the following - although for square loss (without the perturbed network) it was shown that (wide) neural networks satisfy PL and therefore QG, we are the first to show that KL-loss satisfies PL and QG condition which would be of independent interest.
> ***
> 4. **Regret Lower bounds for existing NeuCB methods:** Further note that our lower bound results which show that NeuralUCB and NeuralTS incur an $\Omega(T)$ are novel contributions and further highlight the need for provable sub-linear regret guarantees for NeuCBs. To give some details about the lower bounds, for both algorithms we show two kinds of results:
>     \
>     \
>     **(i)** An instance where an oblivious adversary selects a set of contexts (at the beginning; doesn't even need to know the set of arms played by the algorithm) and a reward function such that the regret of the algorithm is $\Omega(T)$.
>
>     **(ii)** We show that for any choice of reward function $h$ and context vectors, the regret of the algorithm is $\Omega(T)$ as long as $\frac{||\mathbf{h}||_2}{\kappa}$ is $\mathcal{O}(1)$. Here $\mathbf{h}$ is the vector of rewards for all contexts and $\kappa$ is the condition number of the NTK matrix.
>     \
>     \
>     These lower bounds make our regret bound contributions far more important. Note that the only other NeuCB algorithm that provides provable sub-linear regret is EE-Net. However EE-Net makes restrictive assumptions (i) EE-Net assumes all contexts are drawn iid from the same distribution, (ii) EE-Net stores all past networks which does not scale to real word deployment.

---

> ### Author Response · Authors · 2023-11-21
> **Response to Reviewer PXpv**
>
> We hope the above discussion helps further clarify all the technical efforts we have put into this paper and better position the paper in the context of current neural contextual bandits literature.
> ***
> **[Q2] Section 4 only for KL losses**
>
> We clarify that Section-4 applies to **both square loss and KL loss**. Algorithm 1 (NeuSquareCB) uses square loss and the corresponding regret bound is given by Theorem 4.1. Algorithm 2 (NeuFastCB) uses KL loss and the corresponding regret bound is given by Theorem 4.2. To further emphasize this we have added a description in the captions of Algorithm-1 and Algorithm-2 in the new draft that explicitly states the loss function being used.
> ***
> **[Q3] Concerns about Experiments**
>
> We did include an ablation study of the perturbed network in Appendix G.1. We also clarify that both NeuSquareCB and NeuFastCB are NeuCB algorithms proposed in the current paper. We have also stated this in Table-1 for quick access. NeuSquareCB uses the novel results developed by us using the perturbed network for square loss in Section 3.2 while NeuFastCB uses a similar set of results for KL loss that we develop in Section 3.3.
> Therefore in our experiments we compare both our NeuSquareCB and NeuFastCB against existing baselines: NeuralUCB, NeuralTS, and Neural $\epsilon$ greedy.
> \
> \
> The output perturbation ensured a provable $\mathcal{O}(\sqrt{KT})$ and $\mathcal{O}(\sqrt{KL^*} + K)$ regret bound for NeuSquareCB and NeuFastCB respectively. For the set of problems in our experiments we observe that the non-perturbed versions of NeuSquareCB and NeuFastCB empirically behave similarly to the perturbed versions. However, since the non-perturbed algorithms do not come with a provable regret bound, one may be able to construct a problem where the non-perturbed version performs poorly.
> Because of space constraints, these comparisons and discussions were included in Appendix G.1. We have included a brief discussion in the main paper with a forward reference to the appendix in the new draft.
> ***
> **[Q4] Minor typo:**
>
> Thanks for pointing this out. We have updated it in the new version.
> ***
> [Chen et al., 2021]: X. Chen, E. Minasyan, J. D. Lee, and E. Hazan. Provable regret bounds for deep online learning and control. arXiv preprint arXiv:2110.07807, 2021.

---

> > ### Author Response · Authors · 2023-11-23
> >
> > We hope you had a chance to go over our responses to your queries. Since the discussion period is ending soon, we were wondering if you needed any further clarifications.

---

### Author Response · Authors · 2023-11-21
**General Response**

***
We thank the reviewers for the efforts they have put into providing thoughtful feedback. We are encouraged that they found the paper **easy to read**, **well-presented**, **interesting** (R-uhN1) and results are a **potentially important contribution** which would further **benefit the community** (R-PXpv, R-EQUF). We are glad that they found the **investigation and analysis solid** (R-uhN1, R-Gbgm) with **assumptions that are valid for wide range of networks** (R-eHnV). Most importantly we are pleased that the reviewers found the **core perturbation trick** and the corresponding online regression bounds **interesting and novel** (R-Gbgm, R-EQUF) with the **article full of useful theoretical insights** and **enjoyable to read** (R-uhN1). Lastly we are also encouraged that the reviewers are pleased by the experimental details and that they **corroborate the theoretical findings** (R-EQUF,R-Gbgm).
***
The reviewers have asked several insightful questions and made various suggestions to further enhance the current draft. We address these questions and comments in individual responses. Below we provide a brief description of the contributions and the novel aspects of our work to better position the paper in the context of current literature. For a more detailed version please refer to the response "**Novelty of our results and analysis**'' to reviewer PXpv ([Link](https://openreview.net/forum?id=5ep85sakT3&noteId=enBW2sU9Ow)).

1. We introduce a novel $\mathcal{O}(\log T)$ regret bound for online regression under the QG condition (Theorem 3.1), which is different compared to the approach in [Chen et al., 2021]. Unlike their reliance on existing analyses for online regression with convex functions with regret $\mathcal{O}(\sqrt{T})$, we present a new analysis using three assumptions about the loss function: (i) almost convexity, (ii) QG condition, and (iii) unique minima.
2. However since losses with neural networks do not have unique minima, one cannot readily use the above result. Our novel idea of small random perturbation of the neural network output ensures unique minima as well as all other desirable properties and thereby leads to a sharp $\mathcal{O}(\log T)$ regret for online regression with neural networks.
3. With a delicate analysis we show that loss function with the perturbed network satisfies strong convexity and therefore have unique minima. Note that we use the strong convexity only to ensure unique minima and combine that with the new QG regret analysis. We do not use existing regret bounds for strongly convex functions as this does not lead to $\mathcal{O}({\log T})$ regret (also see Point 2. in ([Link](https://openreview.net/forum?id=5ep85sakT3&noteId=enBW2sU9Ow)) for more details).
4. Using the above strategy we show $\mathcal{O}(\log T)$ regret bound with both square loss and KL loss and subsequently use the bandit to online regression reduction to respectively get $\mathcal{O}(\sqrt{KT})$ and $\mathcal{O}(\sqrt{KL^*} + K)$ regret bounds. Here $K$ is number of arms and the second regret is a ``first order'' bound which scales with $L^*$, the loss of the best policy.
5. Our $\Omega(T)$ regret bounds for NeuralUCB ([Zhou et al., 2020]) and NeuralTS ([Zhang et al., 2021]) in Appendix A establishes our proposed algorithms as the first neural contextual bandit algorithms with sub-linear regret under standard assumptions. This is because the only other NeuCB algorithm with sub-linear regret (EE-Net), assumes i.i.d. contexts and stores all past networks to make predictions.
***
[Chen et al., 2021]: X. Chen, E. Minasyan, J. D. Lee, and E. Hazan. Provable regret bounds for deep online learning and control. arXiv preprint arXiv:2110.07807, 2021.

[Zhang et al., 2021]: W. Zhang, D. Zhou, L. Li, and Q. Gu. Neural thompson sampling. In International Conference on Learning Representation (ICLR), 2021.

[Zhou et al., 2020] D. Zhou, L. Li, and Q. Gu. Neural contextual bandits with ucb-based exploration. In International Conference on Machine Learning, pages 11492–11502. PMLR, 2020.

---

### Meta-Review · Area_Chair_xLxL · 2023-12-06

**Metareview:**

Recent work by Foster & Rakhlin (2020) shows a reduction from online contextual bandits (CBs) to online regression, thus enabling sublinear regret CB algorithms via any online regression algorithm (when a realizability assumption holds). The submitted paper leverages and extends this work to study _neural CBs_ (i.e., CBs with neural network-based policies) via online neural regression. The paper's main contributions are:

1. Using off-the-shelf results for wide networks, applied to online regression with the squared error loss, one can immediately show $O(\sqrt{T})$ regret for online neural regression, which yields $O(\sqrt{K} T^{3/4})$ (sublinear) regret for neural CBs, where $K$ is the number of arms and $T$ the number of rounds.
2. For losses that satisfy a quadratic growth (QG) condition, they prove $O(\log T)$ regret for online neural regression, a substantial improvement over the above result.
3. In order to satisfy QG for neural nets, they propose a perturbation approach. They then show that this perturbation method yields $\tilde{O}(\sqrt{K T})$ regret for neural CB with the squared loss, and $\tilde{O}(\sqrt{K L^*} + K)$ regret (where $L^*$ is the loss of the best policy) for neural CB with the KL loss.
4. They compare to existing regret bounds for neural CBs, which either assume i.i.d. contexts (they do not) or are $\Omega(T)$ with an oblivious adversary.
5. Experiments confirm that the proposed neural CB algorithms outperform prior work.

The reviewers more or less agreed that the work was interesting, relevant to the ICLR community, sound and well written.

The main critiques were:
1. Reviewer `PXpv ` said that the technical novelty was not immediately clear. The authors clarified their technical contributions in their response, and `PXpv ` says that this resolved the issue for them.
2. Reviewer `EQUF` had some doubts about whether the paper's assumptions are reasonable; namely, the NTK assumption that the kernel is positive definite. The authors rebutted that this assumption is standard in the NTK literature, and reviewer `Gbgm` agreed with them. `EQUF`, however, remains unconvinced.
3. Reviewer `eHnV ` was concerned that the bounds' failure probability ($\delta$) does not seem to decrease with the number of rounds ($T$), and in fact appears to increase with $T$. The authors responded that this was just a presentation issue; that when one reduces the expression for $\delta$, with certain assumptions on the width ($m$) of the neural network, the failure probability is inversely proportional with $T$. However, this assumes that $m = \Omega(\text{poly}(T))$, meaning the width of the network must grow with the time horizon. This seemed counterintuitive to `eHnV` (and myself), because one would think that, if we keep $m$ fixed, observing more data should result in a better learned policy. Nonetheless, the authors claim that this assumption has precedence in the literature of neural regression, and that prior work has made the same assumption. Not being familiar with this work myself, I will take their word for it. They also propose to use the doubling trick to adapt $m$ to growing values of $T$, and this seems like a reasonable workaround to me.

Though the reviewers still have some unresolved doubts (#2 and #3 above), I nonetheless believe this is a good paper, and it deserves to be accepted at the conference.

**Justification For Why Not Higher Score:**

The overall scores don't seem high enough to justify a spotlight talk. That said, I would be fine with bumping this up to a spotlight.

**Justification For Why Not Lower Score:**

This is solid work, and I feel that its strengths far outweigh its weaknesses.

---

### Decision · Program_Chairs · 2024-01-16

Accept (poster)